biomechanics

biomechanics, footwear inclination, grade, metabolic rate, stairs, walking

**Author for correspondence:**
Prokopios Antonellis
e-mail: pantonellis@unomaha.edu

# Modular footwear that partially offsets downhill or uphill grades minimizes the metabolic cost of human walking

Prokopios Antonellis, Cory M. Frederick, Arash Mohammadzadeh Gonabadi and Philippe Malcolm

Department of Biomechanics and Center for Research in Human Movement Variability, University of Nebraska at Omaha, 6160 University Drive South, Omaha, NE 68182, USA

 PA, 0000-0002-8927-1274; AMG, 0000-0002-4535-0325; PM, 0000-0003-4110-4167

Walking on different grades becomes challenging on energetic and muscular levels compared to level walking. While it is not possible to eliminate the cost of raising or lowering the centre of mass (COM), it could be possible to minimize the cost of distal joints with shoes that offset downhill or uphill grades. We investigated the effects of shoe outsole geometry in 10 participants walking at $1 \, m \, s^{-1}$ on downhill, level and uphill grades. Level shoes minimized metabolic rate during level walking ($P_{\text{second-order effect}} < 0.001$). However, shoes that entirely offset the (overall) treadmill grade did not minimize the metabolic rate of walking on grades: shoes with a +3° (upward) inclination minimized metabolic rate during downhill walking on a −6° grade, and shoes with a −3° (downward) inclination minimized metabolic rate during uphill walking on a +6° grade ($P_{\text{interaction effect}} = 0.023$). Shoe inclination influenced (distal) ankle joint parameters, including soleus muscle activity, ankle moment and work rate, whereas treadmill grade influenced (whole-body) ground reaction force and COM work rate as well as (distal) ankle joint parameters including tibialis anterior and plantarflexor muscle activity, ankle moment and work rate. Similar modular footwear could be used to minimize joint loads or assist with walking on rolling terrain.

## 1. Introduction

The optimal way to locomote depends on the situation, for example, when walking up a grade, walking on uneven terrain

or walking with new shoes. Humans are able to find the metabolically optimal locomotor pattern during novel situations [1] when exposed to the broader energetic landscape, rather than only being able to be metabolically economic during previously known gaits. Walking on level ground demands little effort, but walking on slopes quickly becomes challenging on a metabolic [2–6] and muscular level [7–10]. While it might not be possible to entirely eliminate the cost of raising or lowering the centre of mass (COM) against or with gravity during downhill or uphill walking, investigating the effects of whole-body and distal-limb mechanics on metabolic energy cost could inform new ways of minimizing metabolic rate. For example, stairs are one possible way to reduce metabolic cost compared to climbing an equivalent slope on a ramp surface [11]. Walking on stairs differs from walking on a ramp since stairs allow horizontal foot placement but also require placing the feet in specific positions.

Changes in metabolic rate during uphill and downhill walking come from changes in whole-body mechanics [2–6,12]. Uphill walking leads to an increase in metabolic rate since the legs have to perform more positive work [13] to move the COM upward against gravity. Downhill walking causes a decrease in metabolic rate, but only up to approximately a −6° grade [2,4,6]. During downhill walking, the muscles produce more eccentric work than during level or uphill walking [13] to prevent the COM from accelerating downward. Negative mechanical work is less metabolically costly than positive mechanical work [5] indicating that downhill walking is less metabolically costly than uphill walking. This mechanical work is delivered by the muscles: when walking uphill, hip, knee and ankle extensor muscle activities are increased during the stance phase [7,8] and serve to perform positive work. By contrast, downhill walking causes an increase in the activity of knee extensors [7,9], which is required for limiting forward velocity.

Distal manipulations, such as changes in shoe outsole geometry, can alter ankle biomechanics and metabolic rate during human walking. Shoes with high heels have been shown to increase the metabolic rate of walking by an amount that is proportional to heel height [14]. Walking with shoes that are pointed upward, also called negative-heel shoes, increased the metabolic rate compared to walking with normal shoes [15]. The influence of other geometric parameters of shoes has also been investigated. For example, adding a curved surface with a radius of 30% of the leg length to the bottom of a rigid boot minimized the metabolic rate of walking compared to a range of tested radii [16].

A model of muscle energy expenditure [17] based on the Hill-type muscle model [18] suggests that (i) metabolic cost is near maximal when contractile element lengths are at or below the optimal length on the force–length curve, and (ii) metabolic cost decreases with longer contractile element lengths [19,20]. Optimal muscle fascicle lengths and tendon stiffness values appear to be important for providing the required power output from a muscle [21] and for minimizing metabolic cost during human walking [22,23]. During level walking, altering shoe inclination could influence tendon elongation and change the contractile element length (in lower leg muscles) away from the region that minimizes metabolic rate. Conversely, altering the shoe inclination to offset uphill or downhill walking grades could bring the plantarflexor muscle fascicle lengths back to the optimal region that minimizes metabolic rate.

Using modular footwear as a new tool for altering treadmill grade while keeping the foot segment angle constant (by offsetting the treadmill grade) could provide new insights into the relationships between biomechanical changes and the resulting changes in metabolic rate. Studies on negative-heel shoes [15,24] and shoes with different heel heights [25–27] have used footwear that is commonly available but differs in more than one property (outsole geometry, hardness, etc.). Using commonly available footwear has the advantage of being more ecologically valid but makes it difficult to attribute changes in metabolic rate and biomechanics to a specific shoe parameter. Hence, the aim of this study was to investigate the interaction effects of varying footwear outsole geometry and treadmill grade on metabolic rate, joint mechanics, muscle activity, ground reaction forces (GRF) and COM mechanics in a controlled experiment (all footwear and treadmill parameters were kept constant except the outsole geometry and treadmill grade). We hypothesized that a shoe inclination of 0° (level shoes) would minimize the metabolic energy rate during level walking. We also hypothesized that a shoe inclination that exactly offsets the treadmill grade would minimize the metabolic rate during downhill and uphill walking because it would mimic walking on stairs [11]. Since both experimental manipulations (footwear and treadmill) occurred in the sagittal plane, we sought potential explanations in the sagittal plane biomechanical parameters. Therefore, we focused mostly on ankle kinetics and muscle activation to explain the effects of footwear changes [14,15,24–27] and on whole-body parameters (GRF and COM work rate) to explain the effects of treadmill grade changes.

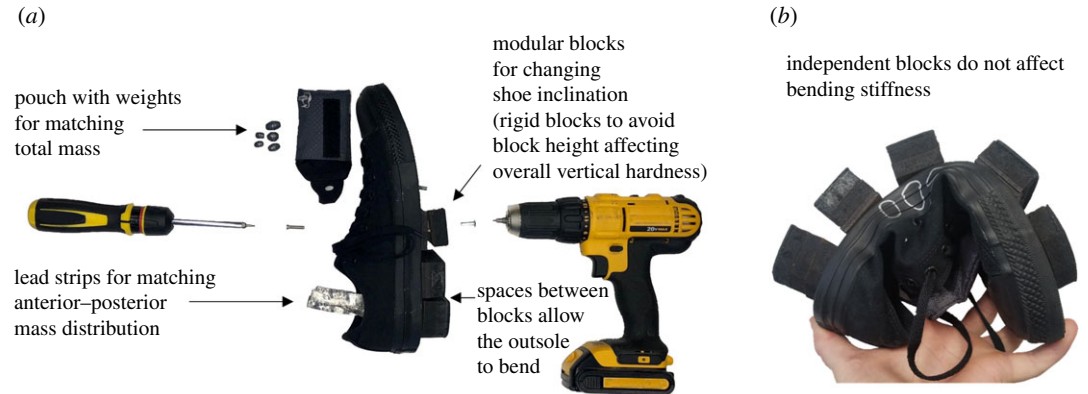

**Figure 1.** Modular outsole geometry shoe assembly. (*a*) By attaching wooden blocks of different heights on the sole, we were able to change outsole geometry without altering bending stiffness or damping (blocks are only shown in the rear part of the shoe). We placed thin lead strips under the forefoot or heel of the shoe to keep the anterior–posterior weight distribution constant across shoes with different geometries. Finally, we attached a pouch with lead weights on top of the centre of the shoe to match the total weight between shoe configurations. (*b*) The attachment of the blocks does not change the bending stiffness.

## 2. Methods

### 2.1. Participants

We recruited 10 healthy male participants (age 24.9 ± 2.7 years, mass 75.7 ± 13.4 kg, height 173.2 ± 6.4 cm; mean ± s.d.). Participants were free from injuries and neuromotor disorders. We tested a homogeneous sample of male participants in order to limit the chances that differences in previous experience with wearing high heels among participants would affect the results and to limit the number of shoe sizes that we had to design for the testing population. The University of Nebraska Medical Center Institutional Review Board approved the study. All participants provided written informed consent.

### 2.2. Modular footwear

We developed a modular shoe that allows for altering the inclination of the foot relative to the ground without affecting other parameters, such as the amount of damping under different parts of the foot, and the bending stiffness of different parts of the outsole (figure 1). We based our experimental shoes on a conventional shoe that does not provide support and has an entirely level outsole (Chuck Taylor All-Star Low Top, Converse, Boston, MA, USA). We used three shoe sizes to accommodate participants (9.5, 10 and 10.5 men's US sizes). Outsole geometry was altered by attaching blocks with different heights to the sole. In studies with high-heel shoes or negative-heel shoes in which the outsole is designed as a single block [15,24,25,27], it is possible that the thicker parts of the outsoles have a greater bending stiffness and lower vertical hardness, which can both affect walking biomechanics [28,29]. We chose a design with separate blocks as opposed to single wedge-shaped outsoles to avoid differences in outsole stiffness due to the outsole inclination. The blocks were made from rigid material (1.27 and 0.32 cm medium-density fibreboard; compression tests with a clamp and digital caliper did not show deformation) to avoid having different block heights result in differences in vertical hardness. Tests of the bending stiffness when dorsiflexing the toe region showed an average stiffness of 0.0011 N m deg$^{-1}$. Grip rubber was glued to the bottom of each block to prevent slipping. To keep the anterior–posterior mass distribution constant, lead weight strips were added under the toe or heel region of the insole depending on where heavier blocks were placed. We also added lead weights in a pouch on top of the centre of the shoe to match the total weight between all shoe configurations within each participant, and avoid that differences in total mass would affect metabolic rate [30–32]. The pouch was tightened down with rubber bands to prevent wobbling. The modular shoes, including the added weights, amounted to a mass of 1.1 kg per side.

### 2.3. Experimental conditions

We tested 15 combinations of three different treadmill grades (−6° downhill, level and +6° uphill) and five shoe inclinations per treadmill condition (figure 2). The number of combinations was selected based

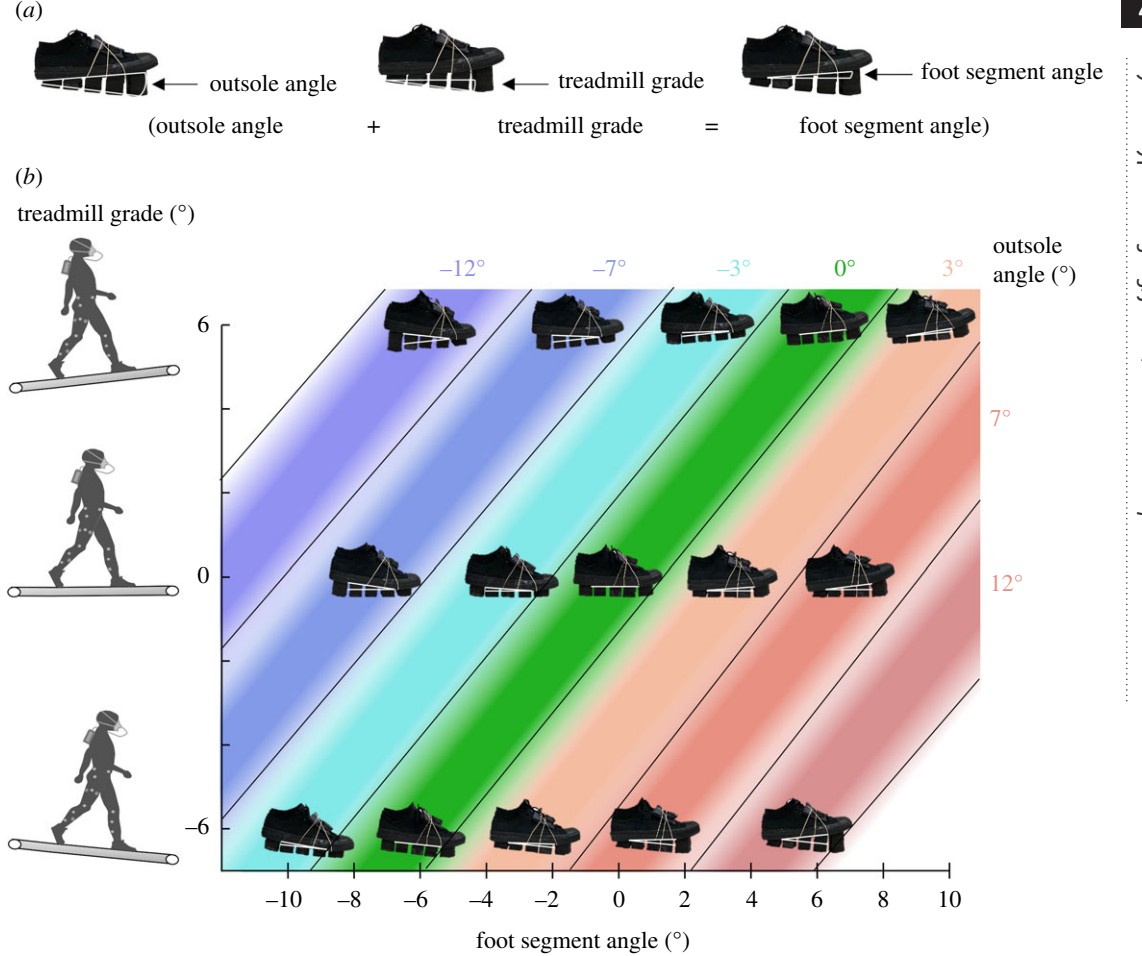

**Figure 2.** Experimental conditions. (*a*) Definition of shoe and treadmill grades. The outsole angle was defined as the angle of the shoe versus the bottom of blocks. Treadmill grade was defined as the grade of the treadmill versus the horizontal plane. The foot segment angle was defined as the angle of the foot versus the horizontal plane. (*b*) We evaluated five different shoe inclinations at three treadmill grades (shown on the vertical axis): downhill (−6°), level and uphill (6°). During downhill walking, we tested shoe inclinations (shown as white angle indicators) ranging from: −3° to +12° (downward shoe inclinations are labelled negative, upward shoe inclinations are labelled positive). During level walking, we tested shoe inclinations ranging from −7° to +7°. During uphill walking, we tested shoe inclinations ranging from −12° to +3°. Coloured bands connect shoe inclinations that were tested at multiple treadmill grades. The combination of the shoe inclinations and treadmill grades resulted in foot segment angles (shown on the horizontal axis) ranging from −9° to 9°.

on an analysis of the results from pilot tests to detect interaction effects. A larger number of footwear conditions than treadmill grades was used because we expected that the effects of treadmill grade would be larger than the effects of shoe inclination, which would be smaller and would therefore require a greater number of data points. The shoe inclinations for each treadmill condition were approximately centred around the inclination that offset the treadmill grade. Specifically, five shoe inclinations (−3°, 0°, 3°, 7° and 12°) were tested during (−6°) downhill walking, five shoe inclinations (−7°, −3°, 0°, 3° and 7°) were tested during level walking and five shoe inclinations (−12°, −7°, −3°, 0° and 3°) were tested during (+6°) uphill walking. Only the −3°, 0° and 3° shoe inclinations were tested at all treadmill grades. Shoe angles that result in the toes pointing down on a level treadmill are given a negative sign and labelled as 'downward'. Shoe angles that result in the toes pointing up on a level treadmill are given a positive sign and labelled 'upward'. One participant did not complete one of the conditions because one of the blocks of the outsole detached.

## 2.4. Protocol

Participants were instructed to fast for at least 6 h prior to the experiment and to abstain from caffeine overnight [33]. Before the experiment, we measured resting metabolic rate during a 5 min standing trial. Participants then

performed a 10 min warm-up [34] to become habituated with the shoe construction and treadmill before data collection. The treadmill speed was set at 1 m s$^{-1}$ [35,36]. All testing conditions lasted 5 minutes and were completed in a single session to avoid day-to-day variability and marker repositioning errors that would affect metabolic rate and biomechanical measurements. The conditions (shoe inclination angles and treadmill grades) were semi-randomized for each participant using a random order generating software, and to minimize the number of shoe inclination angle changes, we tested each shoe inclination on all applicable treadmill grades before changing to the next shoe inclination. The order of the conditions for every second participant was the inverse of the order of the previous participant. As such, all conditions were symmetrically distributed over the first and second halves of the protocol to avoid the effects of metabolic drift on the results. We also allowed the participants at least 2 min of rest between conditions.

## 2.5. Measurements

We recorded oxygen consumption and carbon dioxide production using an indirect calorimetry system (Cosmed, K5, Rome, Italy) during the entirety of each condition. We recorded three-dimensional kinematics at a rate of 200 Hz using a 14-camera motion capture system (VICON Vero, Oxford Metrics, Yarnton, UK). A total of 41 reflective markers were placed on anatomical landmarks and the shoes according to a modified Helen Hayes marker set [37]. The foot segment was defined by using markers placed at the calcaneus (lateral on the heel of the shoe, at furthest point from heel marker on rigid portion of counter), heel (on heel counter at the same height as the toe, centrally when viewing from a posterior position along the long axis of the shoe), first metatarsal (medially on shoe at a point approximating position of first metatarsal head), fifth metatarsal (laterally on shoe at a point approximating position of fifth metatarsal head) and toe (at a point approximating position of second metatarsal head on the dorsum of shoe). We recorded GRF at a frequency of 2000 Hz using an instrumented split-belt treadmill (Bertec, Columbus, OH, USA). We recorded muscle activation using a wireless electromyography (EMG) system (Trigno TM, Delsys, USA; 2000 Hz). Since the footwear conditions were expected to primarily influence the ankle muscles [24–27] and the treadmill grades were expected to influence all lower limb joints [7,8,10,38] and since both manipulations happen mostly in the sagittal plane (i.e. we did not use shoe wedges that were tilted in the frontal plane), we chose to record the muscle activation of the most accessible (major) flexor and extensor muscles of the ankle, knee and hip joints: soleus, gastrocnemius medialis, tibialis anterior, vastus medialis, rectus femoris, biceps femoris and gluteus maximus. Electrodes were positioned according to SENIAM guidelines [39]. Skin preparation prior to attaching the electrodes included shaving hair from the recording site and wiping with an alcohol swab. During the last minute of each condition, we recorded motion capture, GRF and EMG data for 30 s.

## 2.6. Data processing

We used the Brockway equation to calculate metabolic rate based on the rates of oxygen consumption and carbon dioxide production [40]. We averaged the metabolic rate of the last 2 min of each condition to reflect the steady-state metabolic rate. The net metabolic rate of walking was calculated by subtracting the metabolic rate of the standing trial from that of each walking condition.

We filtered marker positions and GRF with a fourth-order low-pass Butterworth filter with a 6 Hz cut-off frequency [35,41]. We calculated the foot segment angle, joint angles, joint moments and powers of the right leg using three-dimensional kinematic analyses and inverse dynamic analyses (Visual3D, C-Motion, Germantown, MD, USA). We estimated body segment mass distribution based on Dempster [42], whereby foot segment mass was adjusted to account for shoe mass. We calculated whole-body COM acceleration based on the body mass and the total GRF of both legs. We calculated COM velocity by integrating the COM acceleration over time [43]. Next, we calculated the product of COM velocity and the individual leg GRF to obtain the right leg COM power [44]. We filtered EMG signals with a 50–400 Hz band-pass filter [45–47]. Then, we rectified EMG signals and applied a moving root mean square with a centred window of 100 ms. We visually inspected EMG signals and removed 95 signals from individual muscles on specific trials that had artefacts (9% of all EMG data).

We segmented each time series based on heel strike detection (using the GRF) and calculated the representative profile per stride by taking the median of all strides. Metabolic rate, GRF, joint moments and powers were normalized to body mass. We normalized each muscle activity to the maximum value across the stride during walking in the condition with level shoes on level treadmill. We calculated the mean angle of the foot segment with respect to the horizontal plane (figure 2$a$) during the foot flat phase approximately from 20 to 35% of the stride cycle [48]. This angle was then subtracted from the condition with level shoes at each treadmill grade such that the shoe inclination

for level shoes is reported as zero degrees. This foot segment angle that was measured during realistic loading conditions was used as an independent parameter in further analyses together with our second independent parameter, treadmill grade. We reported the stride averages of positive or negative forces and moments. We reported joint and COM mechanics by integrating joint and COM powers over stride time and dividing by stride time. We chose to report joint and COM mechanics as work rates (W) instead of work (J) so that they had comparable units to those of metabolic rate and to avoid the potential for changes in step frequency to confound the mechanical work results. We also calculated the stride average of each EMG signal.

## 2.7. Statistical analyses

We reported all dependent variables using means and standard error across participants for each condition. To examine whether shoes that offset treadmill grade optimize metabolic rate and other dependent variables during downhill, level and uphill walking we analysed the changes in all dependent variables versus 'foot segment angle' and 'treadmill grade' as independent variables. Since shoe inclinations are measured as deviations in shoe segment angles from the conditions with level shoes, a foot segment-angle of zero indicates a shoe inclination that offsets the treadmill grade. To determine the effects and interaction effects of foot segment angle and treadmill grade on metabolic rate and biomechanical variables, we used linear mixed-effects model analyses with participant number as random effect [49,50]. For each parameter, we started by evaluating the following model:

$$\text{Dependent variable} = c_0 + c_1 \cdot \text{foot segment angle} + c_2 \cdot \text{foot segment angle}^2$$
$$+ c_3 \cdot \text{treadmill grade} + c_4 \cdot \text{treadmill grade}^2$$
$$+ c_5 \text{foot segment angle} \cdot \text{treadmill grade} \tag{2.1}$$

Where $c_0$ is the constant intercept term and terms $c_1$ to $c_5$ are coefficients for the independent variables. Since we hypothesized that shoes that offset the treadmill grade could minimize metabolic rate and this could be related to minimizing other biomechanical variables, the initial model includes a first- and second-order term of foot segment angle to fit U-shaped trends. We also included a first- and second-order treadmill grade term since metabolic rate follows a U-shaped trend versus treadmill grade [2]. We included an interaction term to evaluate whether an optimal foot segment angle changes for different treadmill grades.

Starting from this model, we could test a number of specific hypotheses regarding the landscapes of metabolic rate and other biomechanical variables in one single analysis. The $p$-value associated with $c_0$ will inform us whether a variable has an intercept that is consistently different (or not) from zero across participants. For example, if the $p$-value for $c_0$ for metabolic rate is different from zero, it indicates that the best fitting statistical model has a metabolic rate that is different from zero for walking with level shoes on a level treadmill grade. The $p$-value associated with $c_1$ will inform us whether a variable follows a non-horizontal trend versus foot segment angle, or in the case that the trend is consistently parabolic, it will inform us whether the location of the minimum is different from a foot segment angle of zero. The $p$-value associated with $c_2$ will inform us whether a variable follows a parabolic trend versus foot segment angle. The $p$-value associated with $c_3$ will inform us whether a variable follows a non-horizontal trend versus treadmill grade and whether the location of the minimum is different from a level treadmill grade. The $p$-value associated with $c_4$ will inform us whether a variable follows a parabolic trend versus treadmill grade. The $p$-value associated with $c_5$ will inform us whether the trend of a variable versus foot segment angle is different at treadmill grades. The landscape shape of the dependent variables would be different depending on which coefficients in the statistical model are not significantly different from zero (figure 3). To avoid overfitting and to adapt the model for dependent variables that have linear trends, we removed terms that did not significantly contribute using backward stepwise elimination similar to other studies [51,52]. If the resulting trend showed a minimum versus the foot segment angle and/or treadmill grade, the location of the minima was obtained by calculating the minimum of the equation from the linear mixed-effects model (with coefficients shown in electronic supplementary material, table S1) at the different treadmill grades and shoe angles. To obtain a sense of the inter-subject variability of the location of the minima, we fitted the terms of the linear mixed-effects model that were statistically significant on each individual participant and calculated the location of the individual minima. To facilitate the interpretation of the meaning of the significant coefficients (positive or negative interaction coefficients), we evaluated the formula resulting from the linear mixed-effects model analysis over the tested range of foot segment angles at each treadmill grade and plotted the results as lines in the scatter

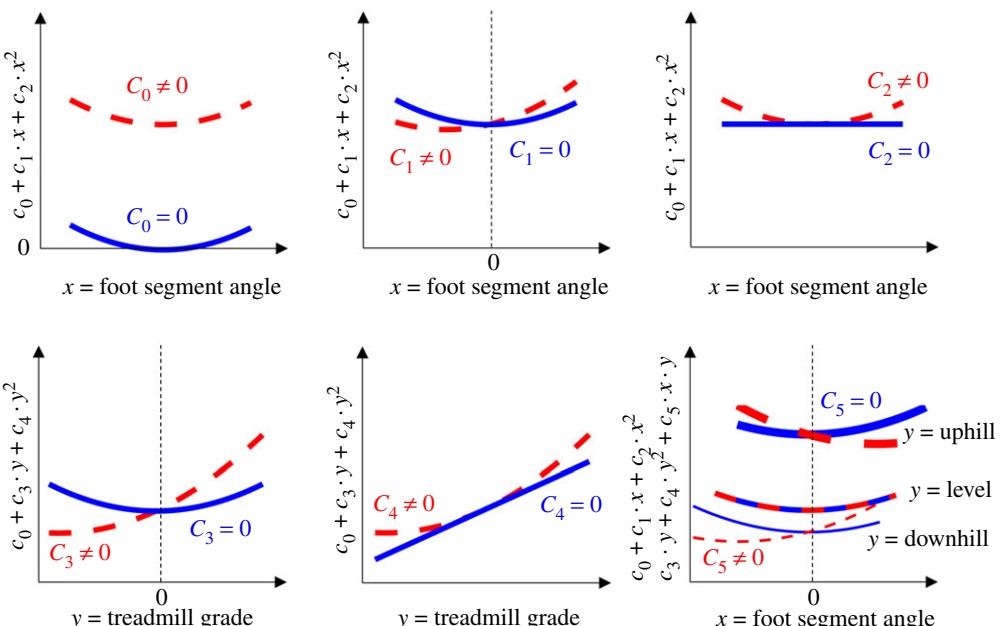

**Figure 3.** Role of significance testing of each coefficient in the statistical model. Subplots show the landscape shape of a variable when certain coefficients are not significantly different from zero.

plots. As such, the results of the statistical analysis can be understood from the figures; for example, if the coefficient for a second-order term was not significant, then the results were plotted as straight lines, or when there was a significant interaction effect, this effect could be understood by analysing the differences in the slope or the location of the minima. All statistical analyses were performed in Matlab (MathWorks, Natick, MA, USA) and the significance threshold was set at 0.05.

# 3. Results

## 3.1. Metabolic rate

We found significant effects of $c_2$ (square of foot segment angle, $p = 0.013$; electronic supplementary material, table S1; figure 4$a$) on metabolic rate indicating that the effect of the foot segment angle on metabolic rate at each treadmill grade followed a parabolic trend. There was a significant effect of $c_3$ (treadmill grade, $p < 0.001$; electronic supplementary material, table S1; figure 4$b$) and of $c_4$ (the square of treadmill grade, $p < 0.001$; electronic supplementary material, table S1; figure 4$b$). This shows that the effect of the treadmill grade on metabolic rate followed a U-shaped trend that was not centred at a 0° treadmill grade. The first order of foot segment angle did not show a significant effect in the first iteration of the linear mixed-effects model analysis ($p = 0.318$) and was therefore removed. Supplementary residual analyses confirmed that the inclusion of the square of foot segment angle improved the fits of the individual participant data, whereas the first order of foot segment angle did not improve the fits. The resulting trends show that during walking on a level treadmill, shoes that result in a level foot segment angle minimized metabolic rate. We found a significant interaction effect of $c_5$ (foot segment angle and treadmill grade, $p = 0.023$; electronic supplementary material, table S1) on metabolic rate. As a result of this interaction effect, shoes that result in a level foot segment angle did not minimize metabolic rate at all treadmill grades. A +3° (upward) outsole minimized metabolic rate during walking on a −6° (downhill) treadmill. A −3° (downward) outsole minimized metabolic rate during walking on a +6° (uphill) treadmill. The combinations of outsole angle and the resulting foot segment angle that minimized metabolic rate at each treadmill grade are shown with pictograms in figure 4$a$.

## 3.2. GRF and COM mechanics

Positive COM work rate (figure 5$b$) and average propulsive GRF (electronic supplementary material, figure S4B) follow a rising parabolic trend versus $c_3$ (treadmill grade, $p < 0.001$; electronic supplementary

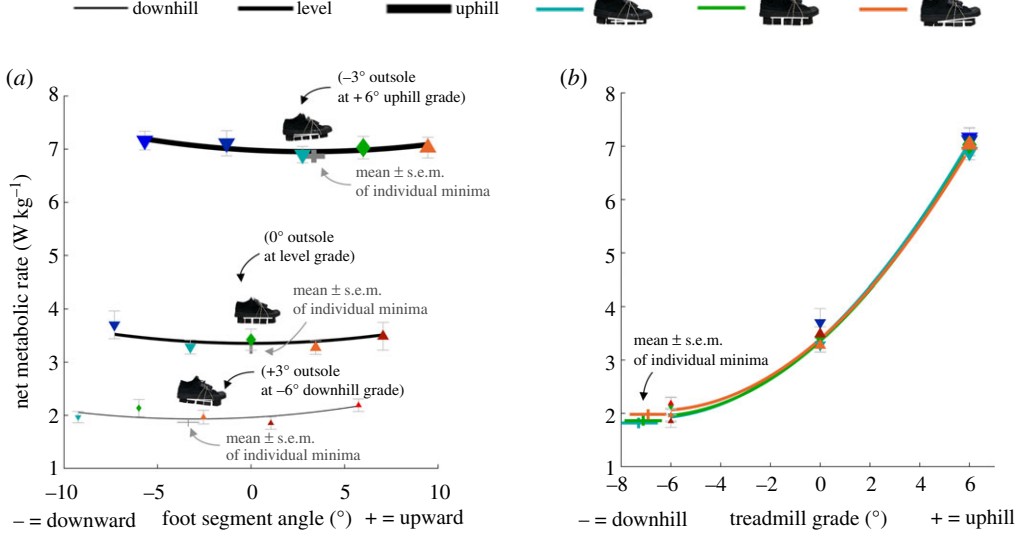

**Figure 4.** Net metabolic rate. (a) Metabolic rate versus foot segment angle during downhill, level and uphill walking. Black lines represent the formula from the linear mixed-effects model analysis evaluated over the tested range of foot segment angles at each treadmill grade. The lines are slightly U-shaped because the coefficient for the second-order of foot segment angle was significantly different from zero and positive. The optimal foot segment angle at each treadmill grade was different because there was a significant interaction effect between the foot segment angle and treadmill grade. Grey crosses represent the mean ± s.e.m. of the interpolated minima. (b) Metabolic rate versus treadmill grade during walking for different shoe inclinations. Coloured lines represent the formula from the linear mixed-effects model analysis evaluated over the range of treadmill grades. Although five shoe inclinations were tested at each treadmill grade and were included in the evaluation of the statistical model, we only plotted the lines representing the evaluation of the linear mixed-effects model analysis for the three shoe inclinations that were tested on all three treadmill grades. The lines are parabolic because the effects of first- and second-order of treadmill grade were significantly different from zero. Coloured crosses represent the mean ± s.e.m. of the extrapolated minima. Blue and cyan downward-pointing triangles represent mean values of conditions with downward shoe inclinations. Green diamonds represent mean values of conditions with level shoes. Orange and red upward-pointing triangles represent mean values of conditions with upward shoe inclinations. Error bars are s.e.m.

material, table S1). Increases in $c_3$ (treadmill grade) from downhill (negative) to level or from level to uphill (positive) led to decreasing magnitudes of average braking GRF ($p < 0.001$; electronic supplementary material, table S1 and figure S4D) and decreasing magnitudes of negative COM work rate ($p < 0.001$; electronic supplementary material, table S1 and figure S4F). In summary, greater uphill treadmill grades resulted to more propulsion and less braking.

We found no isolated effects of $c_1$ (foot segment angle) on average positive and negative GRFs, and COM work rates, but we did find significant interaction effects of $c_5$ (foot segment angle and treadmill grade) on average braking GRF ($p = 0.032$; electronic supplementary material, table S1) and on positive COM work rate ($p = 0.015$; electronic supplementary material, table S1). During downhill walking, smaller negative (downward) and greater positive (upward) foot segment angles led to decreasing positive COM work rate (figure 5a) and increasing magnitudes of average braking GRF (electronic supplementary material, figure S4C). During uphill walking, smaller negative (downward) and greater positive (upward) foot segment angles led to decreasing positive COM work rate (figure 5a) and decreasing magnitudes of average braking GRF (electronic supplementary material, figure S4C). In summary, we found interaction effects of $c_5$ (foot segment angle and treadmill grade) on braking GRF and positive COM work rate that were similar to the interaction effect on metabolic rate.

## 3.3. Ankle mechanics and muscle activation

We found a U-shaped trend in average soleus EMG versus $c_2$ (square of foot segment angle, $p = 0.002$; electronic supplementary material, table S1; figure 6a). The $c_1$ (foot segment angle) and the $c_5$ (foot segment angle and treadmill grade) did not have significant effects on soleus EMG, which indicates that a foot segment angle of 0° minimized soleus EMG for downhill, level and uphill walking. Evaluating the statistical model without eliminating the non-significant terms showed that the individual minima varied

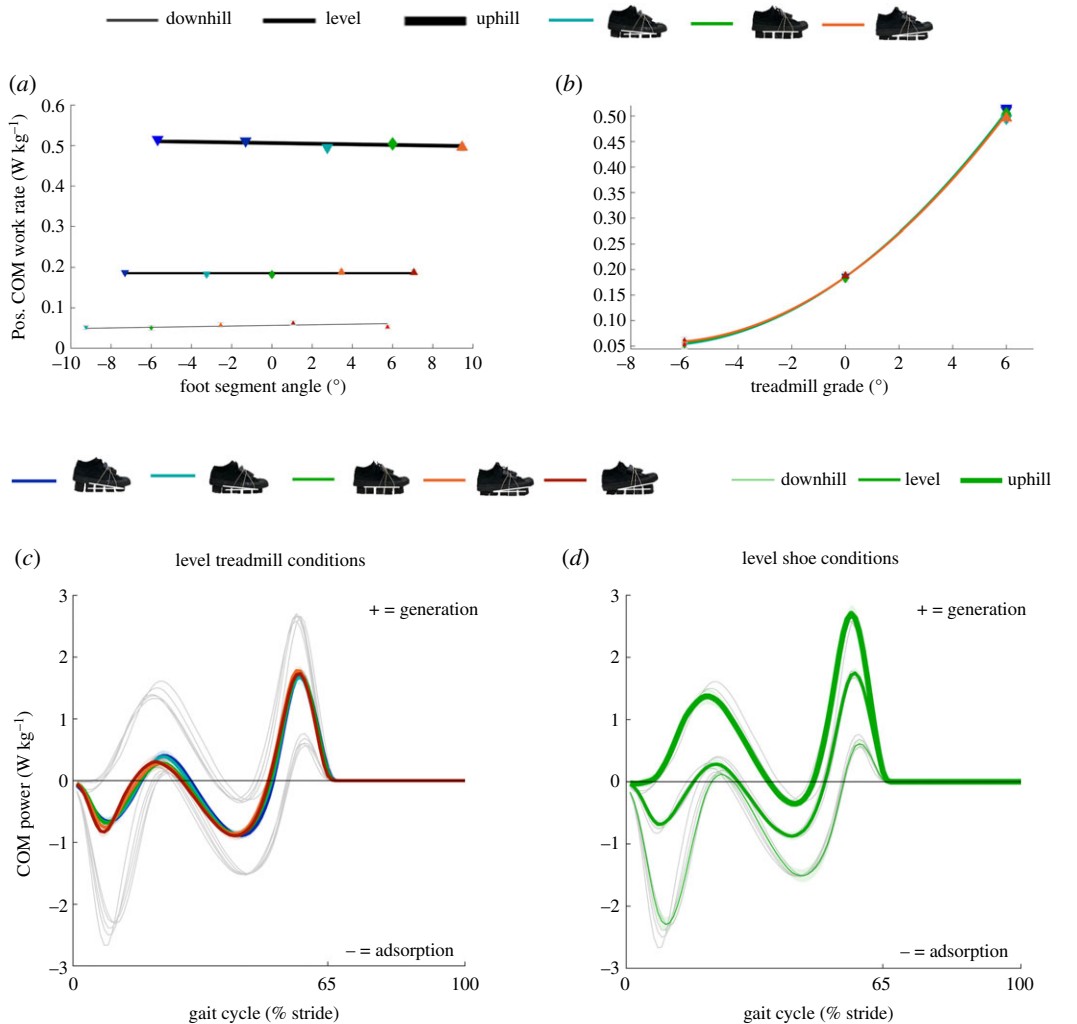

**Figure 5.** COM mechanics. (*a*) Change in positive (pos.) COM work rate versus foot segment angle during downhill, level and uphill walking. Black lines represent the formula from the linear mixed-effects model analysis evaluated over the tested range of foot segment angle at each treadmill grade. (*b*) Change in positive (pos.) COM work rate versus treadmill grade during walking in different shoe inclinations. Coloured lines represent the formula from the linear mixed-effects model analysis evaluated over the range of treadmill grades. Although five shoe inclinations were tested at each treadmill grade and were included in the evaluation of the statistical model, we only plotted the lines representing the evaluation of the linear mixed-effects model analysis for the three shoe inclinations that were tested on all three treadmill grades. The lines are parabolic because the effects of first- and second-order of treadmill grade were significantly different from zero. Blue and cyan downward-pointing triangles represent mean values of conditions with downward shoe inclinations. Green diamonds represent mean values of conditions with level shoes. Orange and red upward-pointing triangles represent mean values of conditions with upward shoe inclinations. Error bars are s.e.m. (*c*) Individual limb COM power differences between shoe inclinations during walking on level treadmill represented by different coloured lines. Grey lines represent other than level treadmill grades. (*d*) Individual limb COM power differences between treadmill grades during walking with level shoes represented by different line thicknesses. Grey lines represent other shoe inclinations. All lines are means of all participants plotted versus stride time. Transparent bands are s.e.m.

with an s.e.m. of 1.1, 0.9 and 0.7 for downhill, level and uphill walking, respectively. Smaller negative (downward) and greater positive (upward) foot segment angles led to increasing average plantarflexion moment ($p < 0.001$; electronic supplementary material, table S1 and figure S1A), increasing positive ankle work rate ($p = 0.001$; electronic supplementary material, table S1 and figure S1E), increasing magnitudes of negative ankle work rate ($p = 0.049$; electronic supplementary material, table S1 and figure S1G) and increasing average tibialis anterior EMG ($p < 0.001$; electronic supplementary material, table S1 and figure S1 K). Smaller negative (downward) and greater positive (upward) foot segment angles led to decreasing magnitudes of average dorsiflexion moment ($p < 0.001$; electronic supplementary material, table S1 and figure S1C). In summary, a level foot segment angle minimized soleus EMG and more upward foot segment angles led to increases in most ankle parameters.

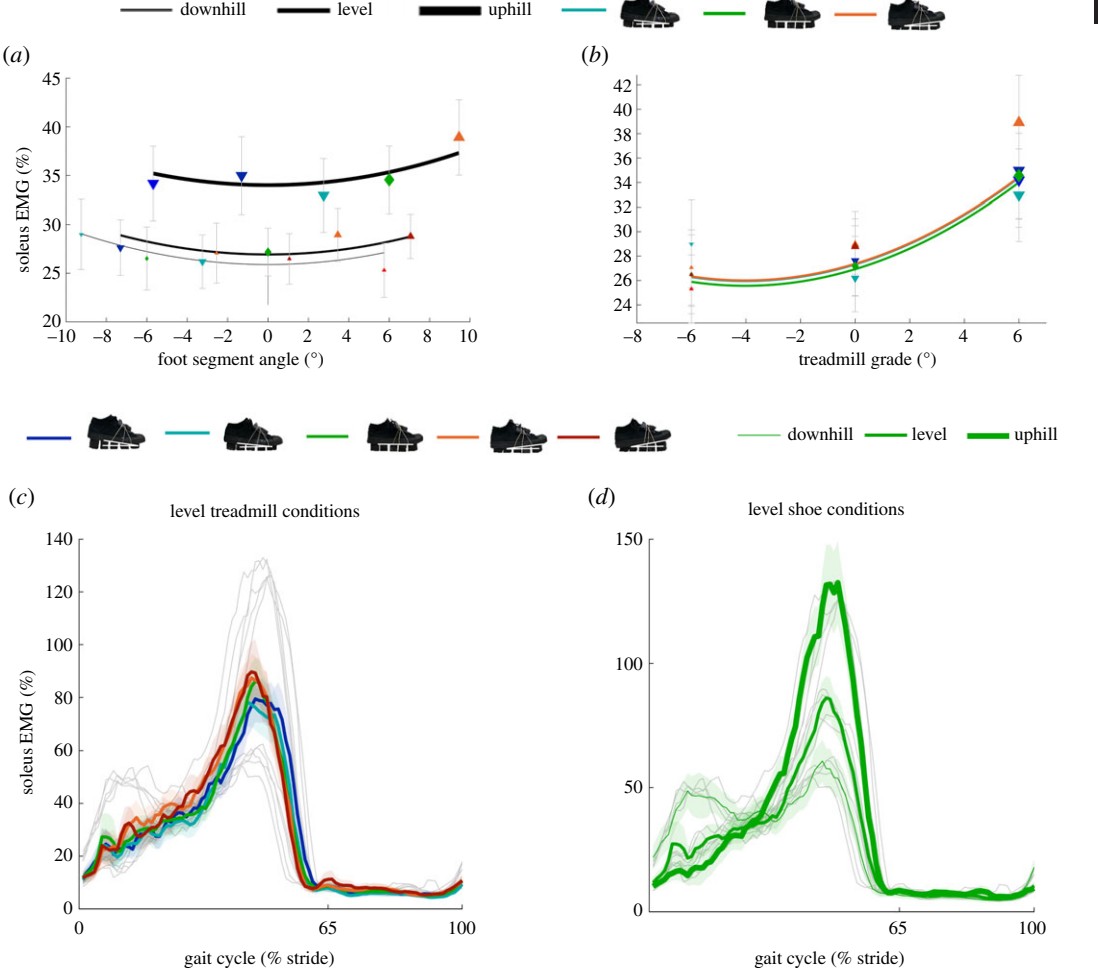

**Figure 6.** Soleus muscle activity. (*a*) Change in average soleus EMG versus foot segment angle during downhill, level and uphill walking. Black lines represent the formula from the linear mixed-effects model analysis evaluated over the tested range of foot segment angle at each treadmill grade. (*b*) Change in average soleus EMG versus treadmill grade during walking in different shoe inclinations. Coloured lines represent the formula from the linear mixed-effects model analysis evaluated over the range of treadmill grades. Although five shoe inclinations were tested at each treadmill grade and were included in the evaluation of the statistical model, we only plotted the lines representing the evaluation of the linear mixed-effects model analysis for the three shoe inclinations that were tested on all three treadmill grades. The lines are parabolic because the effects of first- and second-order of treadmill grade were significantly different from zero. Blue and cyan downward-pointing triangles represent mean values of conditions with downward shoe inclinations. Green diamonds represent mean values of conditions with level shoes. Orange and red upward-pointing triangles represent mean values of conditions with upward shoe inclinations. Error bars are s.e.m. (*c*) Soleus EMG differences between shoe inclinations during walking on level treadmill represented by different coloured lines. Grey lines represent other than level treadmill grades. (*d*) Soleus EMG differences between treadmill grades during walking with level shoes represented by different line thicknesses. Grey lines represent other shoe inclinations. All lines are means of all participants plotted versus stride time. Transparent bands are s.e.m.

Increases in $c_3$ (treadmill grade) from downhill (negative) to level or from level to uphill (positive) led to increasing positive ankle work rate ($p = 0.001$; electronic supplementary material, table S1 and figure S1F) and increasing average gastrocnemius medialis EMG ($p < 0.001$; electronic supplementary material, table S1 and figure S1 J). Increases in $c_3$ (treadmill grade) from downhill (negative) to level or from level to uphill (positive) led to decreasing average plantarflexion moment ($p < 0.001$; electronic supplementary material, table S1 and figure S1B) and decreasing magnitudes of negative ankle work rate ($p < 0.001$; electronic supplementary material, table S1 and figure S1H). We found U-shaped trends in average soleus EMG ($p < 0.001$; electronic supplementary material, table S1; figure 6*b*) and average tibialis anterior EMG ($p < 0.001$; electronic supplementary material, table S1 and figure S1 L) versus $c_3$ (treadmill grade) with respective minima at downhill treadmill grades of $-3°$ and $-2°$.

# 4. Discussion

The aim of the present study was to investigate the interaction effects of varying footwear outsole geometry and treadmill grade on metabolic rate, joint mechanics, muscle activity, GRF and COM mechanics. During level walking, we found that a shoe inclination of 0° minimized metabolic rate (figure 4a). Remarkably, we found that a −3° foot segment angle (obtained with a 3° upward shoe) minimized metabolic rate during walking on a −6° downhill grade. A +3° foot segment angle (obtained with a −3° downward shoe) minimized metabolic rate during walking on a +6° uphill grade. Thus, a shoe inclination that partially offsets uphill or downhill grades can minimize metabolic rate during walking.

The minima for the positive COM work rate and propulsive GRF were very close to the treadmill grade (−6° downhill) that minimized metabolic rate (figure 5b; electronic supplementary material, figure S4B). Both the mechanical work rate and force variables could indeed be responsible for the changes in metabolic rate because energy consumption at the muscle fibre level is dependent on the fibre work force [17]. Concentric muscle work is four times more expensive than eccentric muscle work [5], which could explain why the positive COM work rate and propulsive GRF are closely related to the metabolically optimal treadmill grade. The metabolically optimal treadmill grade was similar to the optimal grade from other studies [5,6], which supports the validity of our experiment.

The reductions in negative COM work rate and braking GRF due to increases in treadmill grade from downhill (negative) to level or from level to uphill (positive) for all shoe inclinations that were tested on all three treadmill grades are consistent with those from other studies [13,53–57]. The increases in positive COM work rate are in line with a study from Franz et al. [13] and the increases in propulsion GRF are in agreement with a number of studies [13,53–57]. We found that increases in treadmill grade from downhill (negative) to level or from level to uphill (positive) also led to increasing dorsiflexion moments, positive ankle work rate, tibialis anterior and plantarflexor muscle activations. These results appear to be consistent with studies that found increased positive ankle work rate [58–60], tibialis anterior [10,38] and plantarflexor muscle activations [7,8,10,38] during uphill walking. Additionally, we found that uphill walking decreased magnitudes of negative ankle work rate similar to other studies [58–60].

To investigate why level shoes minimized metabolic rate during walking on a level treadmill, we started by analysing data from the ankle muscles and the joint kinetics since we expected the footwear manipulation to mostly effect the ankle joint. The energetics model from Umberger et al. [17], which is based on muscle studies [19,20], predicts that when contractile elements are longer than the optimal length, the estimated metabolic rate is multiplied by a force–length-dependent coefficient that is lower than one resulting in a lower metabolic rate. If we assume that shoes that result in upward foot segment angles would lead to longer plantarflexor contractile element lengths, then if all other parameters remain constant, based on the energetics model from Umberger et al. [17], we would expect lower metabolic rate values in conditions with upward foot segment angles. In contrast with this hypothesis, we found increasing metabolic rates in conditions with greater upward foot segment angles. Knowing whether upward shoe conditions led to longer contractile elements lengths would require additional measurements (e.g. ultrasonography). It is also likely that other parameters that affect metabolic rate of muscles were induced, such as contractile element force, velocity and muscle activation. As a matter of fact, we found that soleus muscle activity showed a U-shaped trend with a minimum that aligned with a foot segment angle of 0°, which minimized metabolic rate (figure 6a). A musculoskeletal simulation from Dorn et al. [61] suggests that the soleus consumes the greatest amount of metabolic energy from all muscles during level walking, which could explain the similar U-shaped relationship in metabolic rate found in our study between soleus muscle activity and foot segment angles. It could also be that the U-shaped landscape in metabolic rate is the net result of the summation of metabolic energy consumptions in muscles that have increasing trends with smaller negative (downward) and greater positive (upward) foot segment angles, and other muscles that have decreasing trends with smaller negative (downward) and greater positive (upward) foot segment angles. In our results, we did indeed find variables that increased with smaller negative (downward) and greater positive (upward) foot segment angles, such as average plantarflexion moment (electronic supplementary material, figure S1A), as well as variables that decrease with smaller negative (downward) and greater positive (upward) foot segment angles, such as average dorsiflexion moment (electronic supplementary material, figure S1C) and a number of knee variables (electronic supplementary material, figure S3A,G and I).

Most changes in ankle moments, work rates and muscle activations (figure 6; electronic supplementary material, figure S1) appear to be consistent with prior studies. Panizzolo et al. [41] examined changes in kinematic and kinetic parameters in response to stepping with different parts of the foot on a small bump. Similar to our study, they found increasing plantarflexion moment, increasing plantarflexion work

rate and decreasing magnitudes of dorsiflexion moment when stepping on a bump with the forefoot, which results in a foot angle similar to our upward shoe conditions. Simonsen *et al.* [27] and Stefanyshyn *et al.* [25] also found higher plantarflexion moments in level shoes compared to high-heeled (i.e. downward) shoes. It is hypothesized from a previous study [25] that this could be due to a longer moment arm of the Achilles tendon in level shoes compared to high-heeled shoes allowing the plantarflexors to apply higher moments. We found increasing tibialis anterior activation with smaller negative (downward) and greater positive (upward) foot segment angles. Inspection of time-series plots shows that this occurred during the swing phase and could have been due to an increased need for toe clearance. Curiously, we found no significant effect of foot segment angle on gastrocnemius medialis muscle activity. Similarly, Stefanyshyn *et al.* [25] did not find an effect of heel height on gastrocnemius medialis muscle activity, whereas other studies did find differences in gastrocnemius medialis muscle activity in high-heel and negative-heel shoes compared to level shoes [24,26]. Additionally, more downward foot segment angles led to a decrease in the plantarflexion moment (electronic supplementary material, figure S1A) but also to a decrease in knee flexion moment (electronic supplementary material, figure S3C). As such, this does not explain the lack of reduction in the activation of the gastrocnemius medialis because this muscle is a biarticular muscle that performs plantarflexion and knee flexion. The absence of a significant effect of foot segment angle in our study could be due to the fact that the modular shoes were designed not to affect bending stiffness, which is different from most high-heel or low-heel shoes.

Contrary to our expectations, we did not find a shoe that exactly offsets the treadmill grade minimizes the metabolic rate for uphill and downhill human walking. A −3° foot segment angle minimized metabolic rate during downhill walking and a 3° foot segment angle minimized metabolic rate during uphill walking. We found interaction effects on the biomechanical parameters that followed a similar direction to that of the interaction effect on metabolic rate. More downward foot segment angles reduced the positive COM work rate during downhill walking, whereas more upward foot segment angles reduced the positive COM work during uphill walking (figure 5a). Analyses of the time series that were used for the calculation of COM power indicated that this interaction effect is a result of changes in the perpendicular GRF and perpendicular COM velocity during the rebound phase. Slightly downward foot segment angles could help avoid excessive rebounding of the COM during downhill walking and vice versa for uphill walking. The second interaction effect involves a positive knee work rate with more downward foot segment angles, leading to a reduced positive knee work rate during downhill walking, whereas more upward foot segment angles reduced the positive knee work rate during uphill walking (figure 7). The minimum positive knee work rate during downhill walking appears to be related to a reduction in the extension velocity (possibly related to the COM power effect) and the minimum positive knee work rate during uphill walking seems to be due to a reduction in the extension moment. Taken together, these interaction effects between foot segment angle and treadmill grade on the positive COM work rate and positive knee work rate could explain why a more downward foot segment angle minimized metabolic rate during downhill walking, and why a more upward foot segment angle minimized metabolic rate during uphill walking.

A possible limitation of our study is that the weight of our modular shoes (approx. 1.1 kg) is higher than normal shoes, which typically range from 0.3 to 0.5 kg. Although we matched the total weight and the front–back weight distribution, there might have been differences in moment of inertia due to the mass being either closer or further from the centre of the shoe in different footwear conditions. The additional shoe material could also have reduced sensory perception and impaired stability during downhill or uphill walking. We designed outsoles that were segmented in separate blocks to avoid altering sole bending stiffness. This construction is different from normal outsoles. However, none of the participants commented that it was difficult to walk or to maintain balance. Although the participants walked comfortably with different shoes, lower and perhaps different metabolic rate landscapes could have resulted if participants were given a longer habituation time (e.g. multiple days). While we avoided fatigue confounding the results by randomizing the conditions such that every condition was situated in the beginning and the end of the protocol, it is also possible that different results could have been found if the protocol was divided into shorter sessions with a lower number of conditions. The maximum downward shoe inclination that we tested appears to be within the range of moderate high-heeled shoes. The range of shoe inclinations that were tested per treadmill grade (15° from most downward to most upward) was higher than the range of treadmill grades (12°). Despite the considerable range of tested shoe inclinations, the effects of shoe inclination were relatively shallow and outweighed by the effects of the treadmill grade. There was also high variability in the individual trends (electronic supplementary material, figure S7) similar to studies

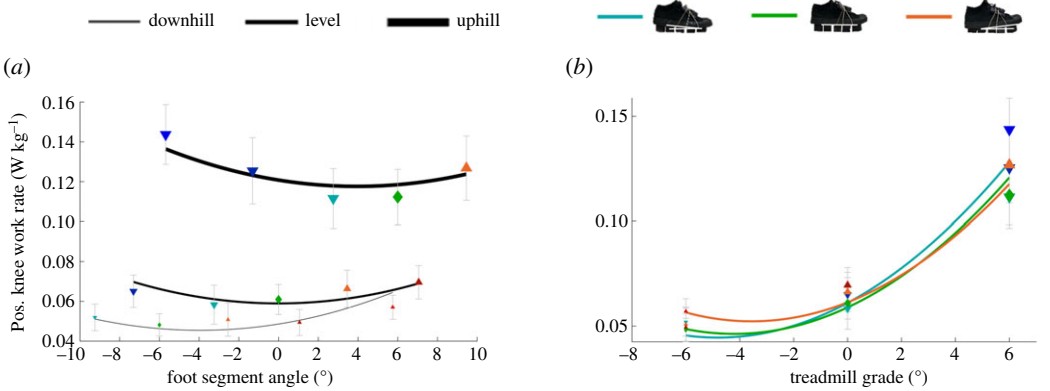

**Figure 7.** Positive knee work rate. (*a*) Positive (pos.) knee work rate changes versus foot segment angle during downhill, level and uphill walking. Black lines represent the formula from the linear mixed-effects model analysis evaluated over the tested range of foot segment angle at each treadmill grade. (*b*) Positive (pos.) knee work rate changes versus treadmill grade during walking in different shoe inclinations. Coloured lines represent the formula from the linear mixed-effects model analysis evaluated over the range of treadmill grades. Although five shoe inclinations were tested at each treadmill grade and were included in the evaluation of the statistical model, we only plotted the lines representing the evaluation of the linear mixed-effects model analysis for the three shoe inclinations that were tested on all three treadmill grades. The lines are parabolic because the effects of first- and second-order of treadmill grade were significantly different from zero. Blue and cyan downward-pointing triangles represent mean values of conditions with downward shoe inclinations. Green diamonds represent mean values of conditions with level shoes. Orange and red upward-pointing triangles represent mean values of conditions with upward shoe inclinations. Error bars are s.e.m.

with exoskeletons and prostheses [62–64]. This interindividual variability is probably a combination of real differences in optimal footwear for each individual and trial-to-trial variability from noisy metabolic rate measurements. A potential application of modular footwear would be to optimize footwear parameters in individual (impaired or unimpaired) participants using human-in-the-loop protocols [64,65] in which the footwear changes are prescribed by a gradient descent algorithm. It is possible that different results might have been found if we had used less sensitive statistical analyses (e.g. repeated-measures ANOVA) or if we had tested a larger number of participants or a larger range of shoe inclinations. While the number of participants and the range of shoe inclinations have been adequate to reject the null hypothesis for five of the six coefficients in our statistical model for metabolic rate ($p$-values $\leq 0.023$), we have no certainty as to whether the non-significant effect of the first order of foot segment angle on metabolic rate ($p = 0.318$) is related to our sample size ($n = 10$). On the other hand, the finding that metabolic rate remains relatively constant over a range of downward and upward shoe inclinations appears to demonstrate how the foot and ankle are capable to adapt to uneven terrain without a large metabolic penalty [66].

Because of the large number of conditions being tested and the changes in metabolic rate and muscle activity that were found, we expect that the dataset provided by this study can be useful for testing and validating musculoskeletal simulations to estimate the metabolic rate of locomotion. The results of our study could inform designs of shoes for walking on rolling terrain (i.e. natural slopes gently rising above and falling below the terrain grade). Since humans typically require more time and have a higher metabolic rate during the uphill portions of terrain with uphill and downhill grades, it could be advantageous to use shoes with a slight downward shoe inclination that is closer to optimal during these uphill portions. This could partially explain why most existing shoes have a small heel to toe drop (a typical offset of 1 cm over a shoe length of 28 cm corresponds to a downward shoe inclination of −2°). However, most existing shoes are probably not designed for only uphill walking and there are other considerations in the design of shoes than minimizing metabolic rate (e.g. injury prevention, comfort and traction). It could also be possible to design shoes that allow changing the inclination depending on the grade of the terrain to avoid repetitive overstretching of the calf muscles, similar to the heel lift on snow shoes or alpine touring skis. Finally, a future experiment could benefit from developing a more lightweight outsole (e.g. through 3D printing) as this would not require as much added mass to keep the total mass and mass distribution constant. These modular shoes could be further used as a research instrument to optimize parameters other than metabolic rate, such as minimizing joint loading to prevent injuries.

# 5. Conclusion

Shoes that are in between level shoes and shoes that entirely offset the treadmill grade minimized the metabolic rate of downhill and uphill human walking. We found that optimal shoe inclinations are primarily related to trends in soleus muscle activity, and interaction effects in positive COM work rate and positive knee work rate could explain why different foot segment angles were optimal at different treadmill grades. Finally, we found that the optimal treadmill grade was about the same as the treadmill grade that minimizes positive COM work rate and propulsive GRFs. The results from this study could be used to tailor modular footwear to optimize metabolic rate and different parameters such as minimizing joint loads for preventing injuries or assisting with walking on rolling terrain.

Ethics. The study protocol was approved by the University of Nebraska Medical Center Institutional Review Board and conducted in accordance with the Declaration of Helsinki. All participants provided written consent to participate.
Data accessibility. Data used to generate the results are available as electronic supplementary material.
Authors' contributions. P.A., C.M.F. and P.M. performed study design. P.A., C.M.F. and A.M.G. collected data. P.A. and C.M.F. performed data analysis. P.A. drafted the manuscript. P.A., C.M.F., A.M.G. and P.M. edited and revised the manuscript.
Competing interests. We declare we have no competing interests.
Funding. This work was supported by the Center for Research in Human Movement Variability of the University of Nebraska at Omaha and the National Institutes of Health (grant no. P20GM109090).
Acknowledgements. We thank Mr Travis Vanderheyden for help with designing the modular shoes, Mr Benjamin Senderling for help with laboratory instrumentation and Dr Aaron Likens, Dr Adam Rosen, Dr Kota Takahashi and Dr Tim Derrick for their insightful advice.

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
