## [Reviewer comments · Royal Society Open Science]

Review History

RSOS-191527.R0 (Original submission)

Review form: Reviewer 1

Is the manuscript scientifically sound in its present form?

Yes

Are the interpretations and conclusions justified by the results?

Yes

Is the language acceptable?

Yes

Do you have any ethical concerns with this paper?

Yes

Have you any concerns about statistical analyses in this paper?

No

Recommendation?

Accept with minor revision (please list in comments)

Comments to the Author(s)

The comments are included (in red and 2 remarks) in the text part of the PDF file (Appendix A).

Review form: Reviewer 2

Is the manuscript scientifically sound in its present form?

Yes

Are the interpretations and conclusions justified by the results?

Yes

Is the language acceptable?

Yes

Do you have any ethical concerns with this paper?

No

Have you any concerns about statistical analyses in this paper?

No

Recommendation?

Accept as is

Comments to the Author(s)

Thank you to the authors for such a comprehensive response to my questions and suggestions.

Review form: Reviewer 3

Is the manuscript scientifically sound in its present form?

Yes

Are the interpretations and conclusions justified by the results?

Yes

Is the language acceptable?

Yes

Do you have any ethical concerns with this paper?

No

Have you any concerns about statistical analyses in this paper?

Yes

Recommendation?

Major revision is needed (please make suggestions in comments)

Comments to the Author(s)

MAJOR COMMENTS

I thank the authors for the opportunity to review this work. The experimental question is particularly interesting, and the research team should be highly commended on a thoughtful experimental approach. The hypothesis motivating this manuscript is that metabolic rate of walking over various slopes can be minimized if the slope is offset by footwear with complementing outsole angles.

However, after hours of pouring over the statistical analyses and connecting these to the results, I still do not feel I have the strongest grasp on the work. I was overwhelmed by this and have no doubt others in the field will also get caught up here. Furthermore, if accepted for publication, this work will be Open Access. The authors reference its clinical application so it may also be important to make findings palatable to a broader audience.

The complex statistical analyses are unfortunate because the question itself is very interesting. I wonder if there are cleaner ways to present this information. One such example is to focus on individual subject responses. Another might be to run a two way repeated measures ANOVA to determine how outsole angle and treadmill grade affect key dependent variables.

It was also unclear whether foot segment angle was a dependent variable because, according to P8, ~L50, it was computed using individual stride data. Yet, it appears to be referenced in results as an independent variable. Should outsole angle not be consistently referenced instead?

More broadly, it feels like an individual subject analysis/discussion is missing in this manuscript. If, as stated in the response to Reviewer 1, the metabolic landscape isn't visible for individual subjects then this is an important and noteworthy finding and I would encourage devoting a figure to this (similar to the Collins example provided). Presenting individual responses in metabolic rate is particularly crucial because the authors state that this work could be applicable to clinicians prescribing footwear to populations with limited effort capacity, and therefore having an understanding of individual variation is useful. Given the similar U-shaped relationship uncovered for soleus activity, it would also be beneficial to know how individual subjects responded here.

One way to quantify this might be use a residual analysis to assess the quality of a second order polynomial fit to the individual subject data points at each grade.

MINOR COMMENTS

P4 L6: Alongside this reference it may be worth noting that humans find the metabolically optimal locomotor pattern during novel situations when exposed to the broader energetic landscape.

P7 Paragraph 1: This is a long protocol, was metabolic data checked for drift?
Also, it may be easier on the reader to list, in-text, the exact treadmill and outsole angles tested instead of a range.

P7 L32: Can the authors clarify (in-text) that the shoe inclination angles were randomized for each participant and the treadmill grades were randomized for each participant?

P8 L27: 6Hz seems very low for ground reaction force data. What was the rationale for this?

P8 L43: Artefacts

P8 L44: Heel strike detection – via ground reaction force or marker data?

P9 L5: Another option for this would be to consider work as J/kg/m (a mechanical cost of transport incorporating the distance traveled per step in case step frequency changes in different conditions, which I suspect it may), and metabolic data as a cost of transport too.

P9 L11: How did the authors confirm their sample size was large enough?

P10 L11: "The P-value associated with c4 will inform us whether a variable follows a parabolic trend versus treadmill grade." - Some outsole angles did not have three samples. Is this only applicable to the range of shared data points across the three treadmill grades?

P11: Statements in Results that extrapolate beyond the experimental data could be excluded in an attempt to simplify this section.

Examples:

"As for the effect of treadmill angle, the resulting trends indicate that a -7deg (downhill) treadmill grade minimized metabolic rate."

"By extrapolating our data, we found that the positive COM work rate and average propulsive GRF are minimized around a downhill treadmill grade of -8°."

P11 L11: For consistency, perhaps referring to C0-C5 would be easier than now referring to the "square of foot segment angle..." for example.

P16 L21, P17 L17: What is rolling terrain?

P16: Is it possible additional shoe sole material reduces sensory perception and perhaps impairs stability in uneven terrain environments?

P16 L38-47: I would consider removing this text.

Figure 2: When printed in BW this figure is very hard to interpret - consider adding thin lines to the edges of the shaded regions.

Also consider removing 4, 2, -2, -4 from the y axis.

If foot segment angle is to be incorporated in this figure, it is hard to read from the x-axis. For example, the far left shoe with -7 slope and 0 treadmill angle seems to align best with -8 foot segment angle, not -7. Perhaps consider writing above/below each shoe what the resulting foot segment angle is. Or, if foot segment angle is a dependent variable, then perhaps consider presenting this as a separate table in the results with mean+/-standard error.

Figures 4-7: Angle lines on shoes are hard to see, maybe these could be thicker and white?

Review form: Reviewer 4

Is the manuscript scientifically sound in its present form?

No

Are the interpretations and conclusions justified by the results?

No

Is the language acceptable?

No

Do you have any ethical concerns with this paper?

No

Have you any concerns about statistical analyses in this paper?

Yes

Recommendation?

Major revision is needed (please make suggestions in comments)

Comments to the Author(s)

Review of “Modular footwear that partially offsets downhill or uphill grades minimizes the metabolic cost of human walking”

Major Concerns

Much stronger justification is needed for background, hypotheses, and application of the results of the proposed study. Why is it important to keep damping and bending stiffness of the shoe constant? What were the values for damping and bending stiffness and how were these verified across conditions? Why was muscle activity measured from the specified 7 muscles of the leg? Why was individual leg power calculated? Why was only Soleus muscle activity reported? The initial aim is vague: “the aim of this study was to investigate the effects of varying footwear outsole geometry on human mechanics and energetics under different walking grades.” compared to the final aim: “The aim of the present study was to investigate the interaction effects of varying footwear outsole geometry and treadmill grade on metabolic rate, joint mechanics, muscle activity, ground reaction forces and COM mechanics.”. There is no connection made between metabolic power and biomechanics. Why and how would positive COM power and propulsive force contribute to metabolic power? Why would longer plantarflexor contractile element lengths reduce metabolic power?

I am confused by the wording that describes the shoe. I have looked at Figure 2 and looked at the first paragraph of the results section and I can't figure out which shoe and treadmill grade combination minimizes metabolic power. Specifically, this statement and others like it are confusing: “A -3° (downward) foot segment angle (or a $+3^\circ$ upward outsole)”. How was the minimum metabolic rate established for the combination of treadmill grade and shoe inclination? Also Figures 4-6 (B) seem to indicate that only 3 shoe inclinations were tested at each slope, but the methods section implies that five were tested. Which 15 conditions were tested? Were all 15 trials completed in one day? If so, how did you account for fatigue and day-to-day variability?

Abstract

Include p-values and number of subjects and walking speed

Please specify what “ankle joint parameters” and whole-body parameters are

Introduction

More justification is needed for why the authors are studying walking on different grades.

It is not apparent why stairs are compared to grades or why it's important to understand the biomechanical parameters.

P4. Lines 21-35. A link between metabolic cost and the mechanical and muscle activity changes during uphill and downhill walking is needed. As written, it is not clear what the point of the paragraph is.

P4. Line 43. Do shoes with the toes pointed upward have the same correlation between height and metabolic cost?

P4. Line 48. This sentence lacks a comparison.

P4-5. Starting at Line 29. Please switch the first two sentences in this paragraph and/or revise to better link to the previous paragraph.

P5. Lines 19-20. There is no justification for why a shoe that offsets grade would minimize metabolic cost.

Methods

P6. Line 7. Why all males?

P7. Line 5. Why were 15 combinations tested?

P7. Line 29. Why was the speed 1 m/s? And how would this affect the results of the study?

Did subjects fast prior to the experiment?

P8. Line 19. The Brockway equation allows you to calculate metabolic power from the rates of oxygen consumption and carbon dioxide production. Please revise.

Results

Why is a polynomial curve fit used for the three grades in Fig. 4B, Fig. 5B, and Fig. 6B? And the y-axes should be labelled "Change in Metabolic Power"

Were any statistics done to ensure that the minimum metabolic power was significantly different for each foot segment angle/treadmill angle compared to the other angles?

The results are extremely hard to follow.

P12. Line 37. It is unclear what "Smaller downhill and great uphill treadmill grades" refers to or what it is being compared to.

Discussion

P13. Second and third paragraphs. Are these paragraphs referring to only the 0° shoe?

P14. Line 6. What is a "a descending-rising trend"?

P14. Lines 40-42. How would a higher plantarflexion moment result from a longer moment arm of the Achilles tendon in level shoes compared to high-heeled shoes?

P15. Lines 7-9. Were these 2% and 8% metabolic power differences significant?

P16. Lines 13-14. It is unclear how a percentage change in walking speed could relate to a percentage change in metabolic power.

P16. How can a speculation about and shoe angle that is outside of what was tested in the current study, which looked at healthy individuals, be used for potential guidance in a patient population?

P16. The comparison with stairs seems like a stretch. It is not clear why the authors are comparing their data from slopes to stairs. Stairs are flat, level surfaces. Why would a shoe with an inclination angle be beneficial?

Decision letter (RSOS-191527.R0)

01-Nov-2019

Dear Dr Malcolm,

The editors assigned to your paper ("Modular footwear that partially offsets downhill or uphill grades minimizes the metabolic cost of human walking") have now received comments from reviewers. We would like you to revise your paper in accordance with the referee and Associate Editor suggestions which can be found below (not including confidential reports to the Editor). Please note this decision does not guarantee eventual acceptance.

Please submit a copy of your revised paper before 24-Nov-2019. Please note that the revision deadline will expire at 00.00am on this date. If we do not hear from you within this time then it will be assumed that the paper has been withdrawn. In exceptional circumstances, extensions may be possible if agreed with the Editorial Office in advance. We do not allow multiple rounds of revision so we urge you to make every effort to fully address all of the comments at this stage. If deemed necessary by the Editors, your manuscript will be sent back to one or more of the original reviewers for assessment. If the original reviewers are not available, we may invite new reviewers.

To revise your manuscript, log into <http://mc.manuscriptcentral.com/rsos> and enter your Author Centre, where you will find your manuscript title listed under "Manuscripts with Decisions." Under "Actions," click on "Create a Revision." Your manuscript number has been

appended to denote a revision. Revise your manuscript and upload a new version through your Author Centre.

- Data accessibility

If you wish to submit your supporting data or code to Dryad (<http://datadryad.org/>), or modify your current submission to dryad, please use the following link:
<http://datadryad.org/submit?journalID=RSOS&manu=RSOS-191527>

- Competing interests

- Authors' contributions

- Acknowledgements

- Funding statement

Kind regards,

Lianne Parkhouse

on behalf of Dr Monica Daley (Associate Editor) and Kevin Padian (Subject Editor)

Associate Editor's comments (Dr Monica Daley):

We have now received feedback from expert reviewers on your revised (transferred) paper. The reviewers are positive about the interesting scientific question, the quality of the experiments and the overall potential for the work to make an important contribution to the literature. They have raised several concerns about the clarity of the study rationale, the statistical analysis and presentation of the results. I recognize and appreciate the authors efforts so far to address the previous comments. Nonetheless, I think the paper would benefit from further revision to address the additional comments raised in response to the revised version. The authors may want to consider consulting a statistical expert for advice on how to address the specific comments about statistical analysis.

Reviewers' Comments to Author:

Reviewer: 1

Comments to the Author(s)

The comments are included (in red and 2 remarks) in the text part of the PDF file
RSOS-191527_Proof_hi_review.pdf

Reviewer: 2

Comments to the Author(s)

Thank you to the authors for such a comprehensive response to my questions and suggestions.

Reviewer: 3

Comments to the Author(s)

MAJOR COMMENTS

I thank the authors for the opportunity to review this work. The experimental question is particularly interesting, and the research team should be highly commended on a thoughtful experimental approach. The hypothesis motivating this manuscript is that metabolic rate of walking over various slopes can be minimized if the slope is offset by footwear with complementing outsole angles.

However, after hours of pouring over the statistical analyses and connecting these to the results, I still do not feel I have the strongest grasp on the work. I was overwhelmed by this and have no doubt others in the field will also get caught up here. Furthermore, if accepted for publication, this work will be Open Access. The authors reference its clinical application so it may also be important to make findings palatable to a broader audience.

The complex statistical analyses are unfortunate because the question itself is very interesting. I wonder if there are cleaner ways to present this information. One such example is to focus on individual subject responses. Another might be to run a two way repeated measures ANOVA to determine how outsole angle and treadmill grade affect key dependent variables.

It was also unclear whether foot segment angle was a dependent variable because, according to P8, ~L50, it was computed using individual stride data. Yet, it appears to be referenced in results as an independent variable. Should outsole angle not be consistently referenced instead?

More broadly, it feels like an individual subject analysis/discussion is missing in this manuscript. If, as stated in the response to Reviewer 1, the metabolic landscape isn't visible for individual subjects then this is an important and noteworthy finding and I would encourage devoting a figure to this (similar to the Collins example provided). Presenting individual responses in metabolic rate is particularly crucial because the authors state that this work could be applicable to clinicians prescribing footwear to populations with limited effort capacity, and therefore having an understanding of individual variation is useful. Given the similar U-shaped relationship uncovered for soleus activity, it would also be beneficial to know how individual subjects responded here.

One way to quantify this might be use a residual analysis to assess the quality of a second order polynomial fit to the individual subject data points at each grade.

MINOR COMMENTS

P4 L6: Alongside this reference it may be worth noting that humans find the metabolically optimal locomotor pattern during novel situations when exposed to the broader energetic landscape.

P7 Paragraph 1: This is a long protocol, was metabolic data checked for drift? Also, it may be easier on the reader to list, in-text, the exact treadmill and outsole angles tested instead of a range.

P7 L32: Can the authors clarify (in-text) that the shoe inclination angles were randomized for each participant and the treadmill grades were randomized for each participant?

P8 L27: 6Hz seems very low for ground reaction force data. What was the rationale for this?

P8 L43: Artefacts

P8 L44: Heel strike detection – via ground reaction force or marker data?

P9 L5: Another option for this would be to consider work as J/kg/m (a mechanical cost of transport incorporating the distance traveled per step in case step frequency changes in different conditions, which I suspect it may), and metabolic data as a cost of transport too.

P9 L11: How did the authors confirm their sample size was large enough?

P10 L11: “The P-value associated with c4 will inform us whether a variable follows a parabolic trend versus treadmill grade.” – Some outsole angles did not have three samples. Is this only applicable to the range of shared data points across the three treadmill grades?

P11: Statements in Results that extrapolate beyond the experimental data could be excluded in an attempt to simplify this section.

Examples:

“As for the effect of treadmill angle, the resulting trends indicate that a -7deg (downhill) treadmill grade minimized metabolic rate.”

“By extrapolating our data, we found that the positive COM work rate and average propulsive GRF are minimized around a downhill treadmill grade of -8°.”

P11 L11: For consistency, perhaps referring to C0-C5 would be easier than now referring to the “square of foot segment angle...” for example.

P16 L21, P17 L17: What is rolling terrain?

P16: Is it possible additional shoe sole material reduces sensory perception and perhaps impairs stability in uneven terrain environments?

P16 L38-47: I would consider removing this text.

Figure 2: When printed in BW this figure is very hard to interpret - consider adding thin lines to the edges of the shaded regions.

Also consider removing 4, 2, -2, -4 from the y axis.

If foot segment angle is to be incorporated in this figure, it is hard to read from the x-axis. For example, the far left shoe with -7 slope and 0 treadmill angle seems to align best with -8 foot segment angle, not -7. Perhaps consider writing above/below each shoe what the resulting foot segment angle is. Or, if foot segment angle is a dependent variable, then perhaps consider presenting this as a separate table in the results with mean+/-standard error.

Figures 4-7: Angle lines on shoes are hard to see, maybe these could be thicker and white?

Reviewer: 4

Comments to the Author(s)

Review of “Modular footwear that partially offsets downhill or uphill grades minimizes the metabolic cost of human walking”

Major Concerns

Much stronger justification is needed for background, hypotheses, and application of the results of the proposed study. Why is it important to keep damping and bending stiffness of the shoe constant? What were the values for damping and bending stiffness and how were these verified across conditions? Why was muscle activity measured from the specified 7 muscles of the leg? Why was individual leg power calculated? Why was only Soleus muscle activity reported? The initial aim is vague: “the aim of this study was to investigate the effects of varying footwear outsole geometry on human mechanics and energetics under different walking grades.” compared to the final aim: “The aim of the present study was to investigate the interaction effects of varying footwear outsole geometry and treadmill grade on metabolic rate, joint mechanics, muscle activity, ground reaction forces and COM mechanics.”. There is no connection made between metabolic power and biomechanics. Why and how would positive COM power and propulsive force contribute to metabolic power? Why would longer plantarflexor contractile element lengths reduce metabolic power?

I am confused by the wording that describes the shoe. I have looked at Figure 2 and looked at the first paragraph of the results section and I can't figure out which shoe and treadmill grade combination minimizes metabolic power. Specifically, this statement and others like it are confusing: “A -3° (downward) foot segment angle (or a +3° upward outsole)”. How was the minimum metabolic rate established for the combination of treadmill grade and shoe inclination?

Also Figures 4-6 (B) seem to indicate that only 3 shoe inclinations were tested at each slope, but the methods section implies that five were tested. Which 15 conditions were tested? Were all 15 trials completed in one day? If so, how did you account for fatigue and day-to-day variability?

Abstract

Include p-values and number of subjects and walking speed

Please specify what “ankle joint parameters” and whole-body parameters are

Introduction

More justification is needed for why the authors are studying walking on different grades.

It is not apparent why stairs are compared to grades or why it's important to understand the biomechanical parameters.

P4. Lines 21-35. A link between metabolic cost and the mechanical and muscle activity changes during uphill and downhill walking is needed. As written, it is not clear what the point of the paragraph is.

P4. Line 43. Do shoes with the toes pointed upward have the same correlation between height and metabolic cost?

P4. Line 48. This sentence lacks a comparison.

P4-5. Starting at Line 29. Please switch the first two sentences in this paragraph and/or revise to better link to the previous paragraph.

P5. Lines 19-20. There is no justification for why a shoe that offsets grade would minimize metabolic cost.

Methods

P6. Line 7. Why all males?

P7. Line 5. Why were 15 combinations tested?

P7. Line 29. Why was the speed 1 m/s? And how would this affect the results of the study? Did subjects fast prior to the experiment?

P8. Line 19. The Brockway equation allows you to calculate metabolic power from the rates of oxygen consumption and carbon dioxide production. Please revise.

Results

Why is a polynomial curve fit used for the three grades in Fig. 4B, Fig. 5B, and Fig. 6B? And the y-axes should be labelled “Change in Metabolic Power”

Were any statistics done to ensure that the minimum metabolic power was significantly different for each foot segment angle/treadmill angle compared to the other angles?

The results are extremely hard to follow.

P12. Line 37. It is unclear what “Smaller downhill and great uphill treadmill grades” refers to or what it is being compared to.

Discussion

P13. Second and third paragraphs. Are these paragraphs referring to only the 0° shoe?

P14. Line 6. What is a “a descending-rising trend”?

P14. Lines 40-42. How would a higher plantarflexion moment result from a longer moment arm of the Achilles tendon in level shoes compared to high-heeled shoes?

P15. Lines 7-9. Were these 2% and 8% metabolic power differences significant?

P16. Lines 13-14. It is unclear how a percentage change in walking speed could relate to a percentage change in metabolic power.

P16. How can a speculation about and shoe angle that is outside of what was tested in the current study, which looked at healthy individuals, be used for potential guidance in a patient population?

P16. The comparison with stairs seems like a stretch. It is not clear why the authors are comparing their data from slopes to stairs. Stairs are flat, level surfaces. Why would a shoe with an inclination angle be beneficial?

Author's Response to Decision Letter for (RSOS-191527.R0)

See Appendix B.

Decision letter (RSOS-191527.R1)

17-Dec-2019

Dear Dr Malcolm,

It is a pleasure to accept your manuscript entitled "Modular footwear that partially offsets downhill or uphill grades minimizes the metabolic cost of human walking" in its current form for publication in Royal Society Open Science. The comments of the reviewer(s) who reviewed your manuscript are included at the foot of this letter.

on behalf of Dr Monica Daley (Associate Editor) and Kevin Padian (Subject Editor)
openscience@royalsociety.org

Associate Editor Comments to Author (Dr Monica Daley):

I want to thank the authors for so thoroughly and thoughtfully addressing the reviewers comments on the revised submission of your paper on the effects of sloped footwear on the energetics of walking at varying grades.

The current version of paper reads clearly, the statistical approach is well justified, and the results and findings are clearly and soundly reported. Thanks for your patience with the review process. The level of engagement and detailed comments by the reviewers make it clear that this is an interesting experiment with some thought-provoking findings. It will make a nice contribution to the field.

Appendix A**ROYAL SOCIETY
OPEN SCIENCE****Modular footwear that partially offsets downhill or uphill grades minimizes the metabolic cost of human walking**

Journal:	Royal Society Open Science
Manuscript ID	RSOS-191527
Article Type:	Research
Date Submitted by the Author:	03-Sep-2019
Complete List of Authors:	Antonellis, Prokopios ; University of Nebraska at Omaha, Department of Biomechanics and Center for Research in Human Movement Variability Frederick , Cory ; University of Nebraska at Omaha, Department of Biomechanics and Center for Research in Human Movement Variability Mohammadzadeh Gonabadi, Arash; University of Nebraska at Omaha, Department of Biomechanics and Center for Research in Human Movement Variability Malcolm, Philippe; University of Nebraska at Omaha, Department of Biomechanics and Center for Research in Human Movement Variability
Subject:	biomechanics < BIOLOGY, biomechanics < CROSS-DISCIPLINARY SCIENCES, biomechanics < PHYSICS
Keywords:	Biomechanics, Footwear inclination, Grade, Metabolic rate, Stairs, Walking
Subject Category:	Biology (whole organism)

Author-supplied statements

Relevant information will appear here if provided.

Ethics

Does your article include research that required ethical approval or permits?:

Yes

Statement (if applicable):

The study protocol was approved by the University of Nebraska Medical Center Institutional Review Board and conducted in accordance with the Declaration of Helsinki. All participants provided written consent to participate.

Data

It is a condition of publication that data, code and materials supporting your paper are made publicly available. Does your paper present new data?:

Yes

Statement (if applicable):

Data used to generate the results are available as electronic supplementary material.

Conflict of interest

I/We declare we have no competing interests

Statement (if applicable):

CUST_STATE_CONFLICT :No data available.

Authors' contributions

This paper has multiple authors and our individual contributions were as below

Statement (if applicable):

P.A., C.M.F. and P.M. performed study design. P.A., C.M.F. and A.M.G. collected data. P.A. and C.M.F. performed data analysis. P.A. drafted the manuscript. P.A., C.M.F., A.M.G. and P.M. edited and revised the manuscript.

**Title:** Modular footwear that partially offsets downhill or uphill grades
minimizes the metabolic cost of human walking

**Authors:** Prokopios Antonellis, Cory M. Frederick, Arash Mohammadzadeh Gonabadi, Philippe
Malcolm*

**Affiliations:** Department of Biomechanics and Center for Research in Human Movement
Variability, University of Nebraska at Omaha, 6160 University Drive South, Omaha, NE 68182,
USA

*** Corresponding Author:**

Philippe Malcolm

Email: pmalcolm@unomaha.edu

Department of Biomechanics and Center for Research in Human Movement Variability

University of Nebraska at Omaha

6160 University Drive South

Omaha, NE 68182

USA

Abstract

Although we know that uneven terrain or grades influence the mechanics and energetics of human walking, we have an incomplete understanding of the effects of whole-body and distal mechanics (do you mean relation between biomechanical adaptations and metabolic rate ?). Is it possible ("is it possible" = descriptive statement = seems not hypothesis driven; and what is the relevance?) to minimize metabolic rate with shoe outsoles that offset downhill or uphill grades? Therefore, We investigated the interaction effects of outsole geometry and treadmill grade on metabolic rate and biomechanics of walking. The resulting trends (why "trends"; aren't there statistical significant differences or correlations ?) indicate that level shoes minimize metabolic rate during level walking. Contrary to our hypothesis (important !), shoes that entirely offset the (overall) treadmill grade (similar to stairs) did not minimize the metabolic rate of walking on grades. Shoes with a +3° (upward) inclination minimized metabolic rate during downhill walking on a -6° grade, and shoes with a -3° (downward) inclination minimized metabolic rate during uphill walking on a +6° grade. Shoe inclination primarily influenced (distal) ankle joint parameters, whereas treadmill grade influenced (whole-body) ground reaction force and center-of-mass parameters as well as (distal) ankle joint parameters. Similar modular footwear could be used to minimize joint loads (e.g., in patient populations) or assist with walking on rolling terrain.

Keywords: Biomechanics, Footwear inclination, Grade, Metabolic rate, Stairs, Walking

1. Introduction

The optimal way to locomote depends on the situation, for example, when walking up a grade, walking on uneven terrain or walking with new shoes. Humans are able to find the metabolically optimal locomotor pattern during novel situations [1], rather than only being able to be metabolically economic during previously known gaits. While it is clear that metabolic rate and mechanical work change for walking on different grades (e.g., walking uphill [2] or on uneven terrain [3]), we still have an incomplete understanding about the effects of whole-body and limb mechanics on metabolic energy cost. For example, walking on a grade can have different energetics and mechanics compared to walking on stairs. It is generally expected that stairs decrease the metabolic cost compared to walking on a ramp at an equivalent grade [4], but we do not fully understand the biomechanical parameters responsible for this.

Changes in whole-body mechanics due to downhill or uphill walking lead to changes in metabolic rate [2,5–9]. Uphill walking leads to an increase in metabolic rate since the legs have to perform more positive work [10] to move the center of mass (COM) upward against gravity. Downhill walking causes a decrease in metabolic rate, but only up to about a -6° grade [2,6,9]. The muscles during downhill walking produce more eccentric work [10] to prevent the COM from accelerating downward. Negative mechanical work is less metabolically costly than positive mechanical work [7]. Furthermore, muscle activity is different in uphill and downhill walking. When walking uphill, hip, knee, and ankle extensor muscle activities are increased during the stance phase [11,12], and the duration of activity of the thigh muscles is also increased [12]. On the contrary, downhill walking causes an increase in the activity of knee extensors [11,13].

Distal manipulations, such as changes in shoe outsole geometry, can also affect ankle biomechanics and metabolic rate during human walking. The heel-toe height difference can be manipulated by adding a high heel or by raising the forefoot. When walking in shoes with high heels, the metabolic rate has been shown to increase proportionally to heel height [14]. Walking with shoes that are pointed upward, also called negative heel shoes, increased the metabolic rate compared to walking with normal shoes [15]. Other geometric parameters of a shoe have also been manipulated. For example, adding a curved surface with a radius of 30% of the leg length to the bottom of a rigid boot minimized the metabolic rate of walking [16].

A model of muscle energy expenditure [17] based on the Hill-type muscle model [18] suggests that (1) metabolic cost is near maximal when contractile element lengths are at or below the optimal length on the force-length curve, and (2) metabolic cost decreases with longer contractile element lengths [19,20]. Optimal muscle fascicle lengths and tendon stiffness values appear to be important for providing the required power output from a muscle [21] and for

minimizing metabolic cost during human walking [22,23]. During level walking, altering shoe
inclination could influence tendon elongation and change the contractile element length (in
lower leg muscles ?) away from the region that minimizes metabolic rate. Conversely, altering
(formulate more directly that you want to offset the grade) the shoe inclination during uphill or
downhill walking could possibly bring the plantarflexor muscle fascicle lengths back to the
optimal region that minimize metabolic rate.

Studying how varying outsole geometry influences metabolic rate on different grades
(formulate more directly that you want to offset the grade) could help to identify relationships
between biomechanical changes and the resulting changes in metabolic rate. Hence, the aim
of this study was to investigate the effects of varying footwear outsole geometry on human
mechanics and energetics under different walking grades (the former 2 sentences say about
the same). We hypothesized that a shoe inclination of 0° (level shoes) would minimize the
metabolic energy rate during level walking. We also hypothesized that a shoe inclination that
exactly offsets the treadmill grade would minimize the metabolic rate during downhill and uphill
walking.

2. Methods

2.1 Participants

We recruited ten healthy male participants (age 24.9 ± 2.7 years, mass 75.7 ± 13.4 kg, height 173.2 ± 6.4 cm; mean \pm SD). Participants were free from injuries and neuromotor disorders. The University of Nebraska Medical Center Institutional Review Board approved the study. All participants provided written informed consent. (OK)

2.2 Modular footwear

We developed a modular shoe that allows for altering the inclination of the foot relative to the ground without affecting other parameters, such as the amount of damping under different parts of the foot, and the bending stiffness of different parts of the outsole (Figure 1). We based our experimental shoes on a conventional shoe that does not provide support and has an entirely level outsole (Chuck Taylor All-Star Low Top, Converse, Boston, MA). We used three shoe sizes to accommodate participants (9.5, 10, and 10.5 men's US sizes). Outsole geometry was altered by attaching blocks with different heights to the sole. We chose a design with separate blocks as opposed to using single wedge-shaped outsoles to avoid differences in outsole stiffness due to the outsole inclination. The blocks were made from non-deformable material (1.27 and 0.32 cm medium-density fiberboard) to avoid that different block heights would result in differences in damping. Grip rubber was glued to the bottom of each block to prevent slipping. To keep the anterior-posterior mass distribution constant, lead weight strips were added under the toe or heel region of the insole depending on where heavier blocks were placed. We also added lead weights in a pouch on top of the center of the shoe to match the total weight between all shoe configurations within each participant, and avoid that differences in total mass would affect metabolic rate [24–26]. The pouch was tightened down with rubber bands to prevent wobbling. The modular shoes, including the added weights, amounted to a mass of 1.1 kg per side. (OK)

[Insert figure 1 about here]

2.3 Experimental conditions

We tested fifteen combinations of three different treadmill grades (-6° downhill, level and $+6^\circ$ uphill) and five footwear inclinations per treadmill condition (Figure 2). The footwear inclinations for each treadmill condition were approximately centered around the inclination that offset the treadmill grade. Footwear inclinations ranged from -3° to $+12^\circ$ during downhill walking, -7° to $+7^\circ$ during level walking and -12° to $+3^\circ$ during uphill walking. Footwear angles that result in the toes pointing down on a level treadmill are given a negative sign and labeled as “downward.” Footwear angles that result in the toes pointing up on a level treadmill are given a positive sign and labeled “upward”. One participant did not complete one of the conditions because one of the blocks of the outsole detached. (OK)

[Insert figure 2 about here]

2.4 Protocol

Before the experiment, we measured resting metabolic rate during a five-minute standing trial. Participants then performed a 10-minute warm-up [27] to become habituated with the shoe construction and treadmill before data collection. The treadmill speed was set at $1 \text{ m}\cdot\text{s}^{-1}$. All testing conditions lasted five minutes. Between conditions, we allowed the participants at least two minutes of rest. The conditions were semi-randomized where, in order 
[revised manuscript text omitted]

0.05. (OK good explanation /motivation of the multiple variable non linear fit, incl interaction term,
+ stats)

**[Insert figure 3 about here]**

3. Results

[Insert table 1 about here]

3.1 Metabolic rate

We found significant effects on metabolic rate of the square of foot segment angle ($P = 0.013$; Table 1; Figure 4A), treadmill grade ($P < 0.001$; Table 1; Figure 4B) and the square of treadmill grade ($P < 0.001$; Table 1; Figure 4B). The first order of foot segment angle did not show a significant effect in the first iteration of the linear mixed-effects model analysis ($P = 0.318$) and was therefore removed. The resulting trends show that during walking on a level treadmill, shoes that result in a level foot segment angle minimized metabolic rate. As for the effect of treadmill angle, the resulting trends indicate that a -7° (downhill) treadmill grade minimized metabolic rate. We found a significant interaction effect on metabolic rate between foot segment angle and treadmill grade ($P = 0.023$; Table 1). As a result of this interaction effect, shoes that result in a level foot segment angle did not minimize metabolic rate at all treadmill grades. A -3° (downward) foot segment angle (or a $+3^\circ$ upward outsole) minimized metabolic rate during walking on a -6° (downhill) treadmill. A $+3^\circ$ (upward) foot segment angle (or a -3° downward outsole) minimized metabolic rate during walking on a $+6^\circ$ (uphill) treadmill.(OK)

[Insert figure 4 about here]

3.2 GRF and COM mechanics

Positive COM work rate (Figure 5B) and average propulsive GRF (Figure S4, B) follow a rising parabolic trend versus treadmill grade ($P < 0.001$; Table 1). By extrapolating our data, we found that the positive COM work rate and average propulsive GRF are minimized around a downhill treadmill grade of -8° . Smaller downhill and greater uphill treadmill grades led to decreasing magnitudes of average braking GRF ($P < 0.001$; Table 1; Figure S4, D) and decreasing magnitudes of negative COM work rate ($P < 0.001$; Table 1; Figure S4, F). In summary, greater uphill treadmill grades resulted to more propulsion and less braking.

We found no isolated effects of foot segment angle on average positive and negative GRFs, and COM work rates, but we did find significant interaction effects of foot segment angle and treadmill grade on average braking GRF ($P = 0.032$; Table 1) and on positive COM work rate ($P = 0.015$; Table 1). During downhill walking, smaller negative (downward) and greater positive

(upward) foot segment angles led to decreasing positive COM work rate (Figure 5A) and
increasing magnitudes of average braking GRF (Figure S4, C). During uphill walking, smaller
negative (downward) and greater positive (upward) foot segment angles led to decreasing
positive COM work rate (Figure 5A) and decreasing magnitudes of average braking GRF (Figure
S4, C). In summary, we found interaction effects of foot segment angle and treadmill grade on
braking GRF and positive COM work rate that were similar to the interaction effect on metabolic
rate. (OK)

**[Insert figure 5 about here]**

**3.3 Ankle mechanics and muscle activation**

We found a U-shaped trend in average soleus EMG versus foot segment angle with a minimum
at a 0° foot segment angle during downhill, level and uphill walking ($P = 0.002$; Table 1; Figure
6A). Smaller negative (downward) and greater positive (upward) foot segment angles led to
increasing average plantarflexion moment ($P < 0.001$; Table 1; Figure S1, A), increasing positive
ankle work rate ($P = 0.001$; Table 1; Figure S1, E), increasing magnitudes of negative ankle work
rate ($P = 0.049$; Table 1; Figure S1, G) and increasing average tibialis anterior EMG ($P < 0.001$;
Table 1; Figure S1, K). Smaller negative (downward) and greater positive (upward) foot segment
angles led to decreasing magnitudes of average dorsiflexion moment ($P < 0.001$; Table S1; Figure
S1, C). In summary, a level foot segment angle minimized soleus EMG and more upward foot
segment angles led to increases in most ankle parameters.

Smaller downhill and greater uphill treadmill grades led to increasing positive ankle work
rate ($P = 0.001$; Table 1; Figure S1, F) and increasing average gastrocnemius medialis EMG (P
< 0.001 ; Table 1; Figure S1, J). Smaller downhill and greater uphill treadmill grades led to
decreasing average plantarflexion moment ($P < 0.001$; Table 1; Figure S1, B) and decreasing
magnitudes of negative ankle work rate ($P < 0.001$; Table 1; Figure S1, H). We found U-shaped
trends in average soleus EMG ($P < 0.001$; Table 1; Figure 6B) and average tibialis anterior EMG
($P < 0.001$; Table 1; Figure S1, L) versus treadmill grade with respective minima at downhill
treadmill grades of -3° and -2° .

**[Insert figure 6 about here] (2 minor remarks for clarity are included in the figures)**

4. Discussion

The aim of the present study was to investigate the interaction effects of varying footwear outsole geometry and treadmill grade on metabolic rate, joint mechanics, muscle activity, ground reaction forces and COM mechanics. During level walking, we found that a shoe inclination of 0° minimized metabolic rate (Figure 4A). Remarkably, we found that a -3° foot segment angle (obtained with a 3° upward shoe) minimized metabolic rate during walking on a -6° downhill grade. A $+3^\circ$ foot segment angle (obtained with a -3° downward shoe) minimized metabolic rate during walking on a $+6^\circ$ uphill grade. (thus a partial offset of the grade by the shoe inclination minimizes metabolic rate: now one expects why such a partial offset "works" best)

(Here, you start with confirmation of existing results = not innovative; serves only validity of the method; consider to write shorter mentioning purpose validity)

Extrapolating the trend of metabolic rate versus treadmill grade showed that the treadmill grade that minimizes metabolic rate was within 1° to the tested downhill treadmill grade of -6° (Figure 4B). Studies from Margaria, [7] and Minetti et al., [9] also found that the metabolic rate of walking is minimized around a -6° downhill grade. The minima for positive COM work rate and propulsive GRF were very close to the treadmill grade that minimized metabolic rate suggesting that these parameters possibly contribute to metabolic rate being minimal around a -6° downhill treadmill grade (Figs. 5B and S4, B).

The decreases in negative COM work rate and braking GRF with smaller thandownhill and greater than uphill walking grades are consistent with previous studies [10,42–46]. The increases in positive COM work rate are in line with a study from Franz et al., [10] and the increases in propulsion GRF are in agreement with a number of studies [10,42–46]. We found that smaller downhill and greater uphill walking also led to increasing dorsiflexion moments, positive ankle work rate, tibialis anterior, and plantarflexor muscle activations. These results appear to be consistent with studies that found increased positive ankle work rate [47–49], tibialis anterior [50,51] and plantarflexor muscle activations [11,12,50,51] during uphill walking. Additionally, we found that uphill walking decreased magnitudes of negative ankle work rate similar to other studies [47–49].

The fact that level shoes minimized metabolic rate during walking on a level treadmill could be related to reduced effort of the ankle joint. If we assume that shoes that result in upward foot segment angles would lead to longer plantarflexor contractile element lengths, then if all other parameters would stay constant, based on the energetics model from Umberger et al., [17] we would expect lower metabolic rate values in conditions with upward foot segment angles. In contrast to this hypothesis, we found increasing metabolic rates in conditions with greater upward foot segment angles. Knowing whether upward shoe conditions led to longer contractile elements lengths would require additional measurements (e.g.,

parameters that affect metabolic rate of muscles were induced such as contractile element
force, velocity, and muscle activation. As a matter of fact, we found that soleus muscle activity
showed a descending-rising trend with a minimum that aligned with a foot segment angle of
0° which minimized metabolic rate (Figure 6A). A musculoskeletal simulation from Dorn et al., [52]
suggests that the soleus consumes the greatest amount of metabolic energy from all muscles
during level walking which could explain the similar U-shaped relationship in metabolic rate
found in our study between soleus muscle activity and foot segment angles. It could also be
that the U-shaped landscape in metabolic rate is the net result of the summation of metabolic
energy consumptions in muscles that have increasing trends with smaller negative
(downward) and greater positive (upward) foot segment angles, and other muscles that
have decreasing trends with smaller negative (downward) and greater positive (upward) foot
segment angles. In our results, we did indeed find variables that increased with smaller
negative (downward) and greater positive (upward) foot segment angles such as average
plantarflexion moment (Figure S1, A) as well as variables that decrease smaller negative
(downward) and greater positive (upward) foot segment angles, such as average dorsiflexion
moment (Figure S1, C) and a number of knee variables (Figure S3, A, G and I).

Most changes in ankle moments, work rates and muscle activations (Figs. 6 and S1)
appear to be consistent with prior studies. Panizzolo et al., [53] examined changes in kinematic
and kinetic parameters in response to stepping with different parts of the foot on a small bump.
Similar to our study, they found increasing plantarflexion moment, increasing plantarflexion work
rate and decreasing magnitudes of dorsiflexion moment when stepping on a bump with the
forefoot, which results in a foot angle similar to our upward shoe conditions. Simonsen et al.,
[54] and Stefanyshyn et al., [55] also found higher plantarflexion moments in level shoes
compared to high-heeled (i.e., downward) shoes. This could be due to a longer moment arm
of the Achilles tendon in level shoes compared to high-heeled shoes [55]. We found increasing
tibialis anterior activation with smaller negative (downward) and greater positive (upward) foot
segment angles. Inspection of time series plots shows that this occurred during the swing phase
and could have been due to an increased need for toe clearance. Curiously, we found no
significant effect of foot segment angle on gastrocnemius medialis muscle activity. Similarly,
Stefanyshyn et al., [55] did not find an effect of heel height on gastrocnemius medialis muscle
activity whereas other studies did find differences in gastrocnemius medialis muscle activity
in high-heel and negative heel shoes compared to level shoes [56,57]. The absence of a
significant effect of foot segment angle in our study could be due to the fact that the modular
shoes were designed not to affect bending stiffness, which is different from most high-heel or
low heel shoes. (what about the effect of a possible change in knee angle; would affect gastroc)
(more important: is the effect on the metabolic rate mostly due to distal = ankle effect or due to
total body COM effect ?)

Contrary to our expectations, we did not find a shoe that exactly offsets the treadmill grade minimizes the metabolic rate for uphill and downhill human walking. A -3° foot segment angle reduced metabolic rate (by $\approx 8\%$ compared to level shoes) during downhill walking and a 3° foot segment angle reduced metabolic rate (by $\approx 2\%$ compared to level shoes) during uphill walking. We found two similar interaction effects on biomechanical parameters; more downward foot segment angles reduced positive COM work rate and positive knee work rate during downhill walking whereas more upward foot segment angles reduced positive COM work rate (Figure 5A) and positive knee work rate (Figure 7) during uphill walking. These interaction effects between foot segment angle and treadmill grade could possibly explain that a more downward foot segment angle minimized metabolic rate during downhill walking, and a more upward foot segment angle minimized metabolic rate during uphill walking. (last sentence is a repetition of results, not a kinesiological explanation, a functional biomechanical explanation would be appropriate)

[Insert figure 7 about here]

A possible limitation of our study is that the weight of our modular shoes (≈ 1.1 kg) is higher than normal shoes which typically range from 0.3 to 0.5 kg. Although we matched the total weight and the front-back weight distribution, there might have been differences in moment of inertia due to the mass being either closer or further from the center of the shoe in different footwear conditions. We designed outsoles that were segmented in separate blocks to avoid altering sole bending stiffness. This construction is different from normal outsoles. However, none of the participants commented that it was difficult to walk or to maintain balance. Although the participants walked comfortably with different shoes, it is possible that lower and perhaps different metabolic rate landscapes could have resulted if participants were given a longer habituation time (e.g., multiple days). The maximum downward shoe inclination that we tested appears to be within the range of moderate high-heeled shoes. The range of shoe inclinations that were tested per treadmill grade (15° from most downward to most upward) was higher than the range of treadmill grades (12°). Despite the considerable range of tested shoe inclinations, the effects of shoe inclination were relatively shallow and outweighed by the effects of treadmill grade. It is possible that different results might have been found if we had tested a larger number of participants or higher range of shoe inclinations. While the number of participants and range of shoe inclinations have been adequate to reject the null hypothesis for 5 of the 6 coefficients of our statistical model for metabolic rate (P -values ≤ 0.023), we have no certainty whether the non-significant effect of the first order of foot segment angle on metabolic rate ($P = 0.318$) is related to our sample size ($n = 10$). On the other hand, the finding that metabolic rate remains relatively constant over a range

of downward and upward shoe inclinations appears to demonstrate how the foot and ankle are
capable to adapt to uneven terrain without a large metabolic penalty [58].

Because of the large number of conditions being tested and the changes in metabolic
rate and muscle activity that were found, we expect that the dataset provided by this study
can be useful for testing and validating musculoskeletal simulations to estimate the metabolic
rate of locomotion. Even though the effects of foot segment angles on metabolic cost were
relatively small, they could be clinically meaningful. Changes of 7% in walking speed have been
considered clinically meaningful in elderly and stroke patients [59]. By slightly extrapolating the
trend in shoe conditions in the level treadmill condition, we can estimate that a shoe with a 9°
inclination could cause a similar to percentage change in metabolic rate. This could be used
as a potential guideline to inform patient populations with limited effort capacity to use shoes
with a lesser inclination. The results of our study could also inform designs of shoes for walking
on rolling terrain. Since humans typically require more time and have a higher metabolic rate
during uphill portions of terrain with uphill and downhill grades, it could be advantageous to use
shoes with a slight downward shoe inclination that are closer to optimal at these uphill
portions. Perhaps this explains why most existing shoes have a small heel to toe drop (a
typical offset of 1 cm over a shoe length of 28 cm corresponds to a downward shoe inclination
of -2°). (nice suggestion, but aren't there different explanations ? fit and avoiding slip and
friction ? orientation of push off vector and timing of forward COP displacement during fore foot
roll off, certainly with stiffer sole constructions, etc ... It could also be possible to design shoes
that allow changing the inclination depending on the grade of the terrain (e.g., similar to heel
lift on snow shoes or alpine touring skis (nice = mostly to avoid repetitive overstretching of calf
muscles)). Our data could further be related to differences in metabolic rate between walking on
stairs and ramps at an equivalent average grade. At a 6° grade, we found that the shoe
condition that offsets the average grade (similar to a stairway) was 13% more metabolically
economic than walking down the same grade with level (0° shoe inclination) shoes. If we
extrapolate this difference to the average grade for which stairs are recommended (37°
based on California Code of Regulations, <https://www.dir.ca.gov/title8/3231.html>), we can
estimate that a shoe that offsets the average grade could be 82% more metabolically economic.
This extrapolation should be interpreted with caution since the grade that we tested was much
lower than 37°, and walking on stairs can constrain foot placement which is different from our

[revised manuscript text omitted]

59. Perera S, Mody SH, Woodman RC, Studenski SA. 2006 Meaningful Change and
Responsiveness in Common Physical Performance Measures in Older Adults. *J. Am.*
*Geriatr. Soc.* **54**, 743–749. (doi:10.1111/j.1532-5415.2006.00701.x)

Figure legends

Figure 1. Modular outsole geometry shoe assembly. By attaching wooden blocks of different heights on the sole, we were able to change outsole geometry without altering bending stiffness or damping (blocks are only shown in the rear part of the shoe). We placed thin lead strips under the forefoot or heel of the shoe to keep the anterior-posterior weight distribution constant across shoes with different geometries. Finally, we attached a pouch with lead weights on top of the center of the shoe to match the total weight between shoe configurations.

Figure 2. Experimental conditions. A: Definition of shoe and treadmill angles. The outsole angle was defined as the angle of the shoe versus the bottom of blocks. Treadmill angle was defined as the angle of the treadmill versus the horizontal plane. The foot segment angle was defined as the angle of the foot versus the horizontal plane. B: We evaluated five different shoe inclinations at three treadmill grades (shown on the vertical axis): downhill (-6°), level and uphill (6°). During downhill walking, we tested shoe inclinations (shown as blue angle indicators) ranging from: -3° to $+12^\circ$ (downward shoe inclinations are labeled negative, upward shoe inclinations are labeled positive). During level walking, we tested shoe inclinations ranging from -7° to $+7^\circ$. During uphill walking, we tested shoe inclinations ranging from -12° to $+3^\circ$. Colored bands connect shoe inclinations that were tested at multiple treadmill grades. The combination of the shoe inclinations and treadmill grades resulted in foot segment angles (shown on the horizontal axis) ranging from -9° to 9° .

Figure 3. Role of significance testing of each coefficient in the statistical model. Subplots show the landscape shape of a variable when certain coefficients are not significantly different from zero.

Figure 4. Metabolic rate. A: Change in metabolic rate versus foot segment angle during downhill, level, and uphill walking. Black lines represent the formula from the linear mixed-effects model analysis evaluated over the tested range of foot segment angles at each treadmill grade. Grey crosses indicate the interpolated minima \pm s.e.m. B: Change in metabolic rate versus treadmill grade during walking for different shoe inclinations. Colored lines represent the formula from the linear mixed-effects model analysis evaluated over the range of treadmill grades for the three shoe inclinations that were tested on all treadmill grades. Colored crosses represent the extrapolated minima \pm s.e.m. Blue and cyan downward-pointing triangles represent mean values

[revised manuscript text omitted]

University of Nebraska at Omaha

6160 University Drive South

Omaha, NE 68182

USA

Abstract

Although we know that uneven terrain or grades influence the mechanics and energetics of human walking, we have an incomplete understanding of the effects of whole-body and distal mechanics. Is it possible to minimize metabolic rate with shoe outsoles that offset downhill or uphill grades? We investigated the interaction effects of outsole geometry and treadmill grade on metabolic rate and biomechanics of walking. The resulting trends indicate that level shoes minimize metabolic rate during level walking. Contrary to our hypothesis, shoes that offset the overall treadmill grade (similar to stairs) did not minimize the metabolic rate of walking on grades. Shoes with a +3° (upward) inclination minimized metabolic rate during downhill walking on a -6° grade, and shoes with a -3° (downward) inclination minimized metabolic rate during uphill walking on a +6° grade. Shoe inclination primarily influenced (distal) ankle joint parameters, whereas treadmill grade influenced (whole-body) ground reaction force and center-of-mass parameters as well as (distal) ankle joint parameters. ~~Positive center-of-mass power or average propulsive ground reaction force each explained the majority of variance in metabolic rate.~~ Similar modular footwear could be used to minimize joint loads (e.g., in patient populations) or assist with walking on rolling terrain.

Keywords: Biomechanics, Footwear inclination, Grade, Metabolic rate, Stairs, Walking

1. Introduction

The optimal way to locomote depends on the situation, for example, when walking up a grade, walking on uneven terrain or walking with new shoes. Humans are able to find the metabolically optimal locomotor pattern during novel situations [1], rather than only being able to be metabolically economic during previously known gaits. While it is clear that metabolic rate and mechanical work change for walking on different grades (e.g., walking uphill [2] or on uneven terrain [3]), we still have an incomplete understanding about the effects of whole-body and limb mechanics on metabolic energy cost. For example, walking on a grade can have different energetics and mechanics compared to walking on stairs. It is generally expected that stairs decrease the metabolic cost compared to walking on a ramp at an equivalent grade [4], but we do not fully understand the biomechanical parameters responsible for this.

Changes in whole-body mechanics due to downhill or uphill walking lead to changes in metabolic rate [2,5–9]. Uphill walking leads to an increase in metabolic rate since the legs have to perform more positive work [10] to move the center of mass (COM) upward against gravity. Downhill walking causes a decrease in metabolic rate, but only up to about a -6° grade [2,6,9]. The muscles during downhill walking produce more eccentric work [10] to prevent the COM from accelerating downward. Negative mechanical work is less metabolically costly than positive mechanical work [7]. Furthermore, muscle activity is different in uphill and downhill walking. When walking uphill, hip, knee, and ankle extensor muscle activities are increased during the stance phase [11,12], and the duration of activity of the thigh muscles is also increased [12]. On the contrary, downhill walking causes an increase in the activity of knee extensors [11,13].

Distal manipulations, such as changes in shoe outsole geometry, can also affect ankle biomechanics and metabolic rate during human walking. The heel-toe height difference can be manipulated by adding a high heel or by raising the forefoot. When walking in shoes with high heels, the metabolic rate has been shown to increase proportionally to heel height [14]. Walking with shoes that are pointed upward, also called negative heel shoes, increased the metabolic rate compared to walking with normal shoes [15]. Other geometric parameters of a shoe have also been manipulated. For example, adding a curved surface with a radius of 30% of the leg length to the bottom of a rigid boot minimized the metabolic rate of walking [16].

A model of muscle energy expenditure [17] ~~developed for predicting thermal and mechanical energy~~ based on the Hill-type muscle model [18] suggests that (1) metabolic cost is near maximal when contractile element lengths are at or below the optimal length on the force-length curve, and at the optimal fascicle length on the force-length relationship, (2) metabolic cost decreases with longer contractile element lengths [19,20] ~~greater fascicle lengths, and (3)~~

~~metabolic cost does not change with shorter fascicle lengths than the optimal length on the force-length relationship.~~ Optimal muscle fascicle lengths and tendon stiffness values appear to be important for providing the required power output from a muscle [21] and for minimizing metabolic cost during human walking [22,23]. During level walking, altering shoe inclination could influence tendon elongation and change ~~the contractile element muscle-fascicle~~ length away from the region that minimizes metabolic rate. Conversely, altering the shoe inclination during uphill or downhill walking could possibly bring the plantarflexor muscle fascicle lengths back to the optimal region that minimize metabolic rate.

Studying how varying outsole geometry influences metabolic rate on different grades could help to identify relationships between biomechanical changes and the resulting changes in metabolic rate. Hence, the aim of this study was to investigate the effects of varying footwear outsole geometry on human mechanics and energetics under different walking grades. We hypothesized that a shoe inclination of 0° (level shoes) would minimize the metabolic energy rate during level walking. We also hypothesized that a shoe inclination that exactly offsets the treadmill grade would minimize the metabolic rate during downhill and uphill walking. ~~Finally, we conducted biomechanical analyses to explain the changes in metabolic rate, provide validation data for predictive models, and inform footwear designs.~~

2. Methods

2.1 Participants

We recruited ten healthy male participants (age 24.9 ± 2.7 years, mass 75.7 ± 13.4 kg, height 173.2 ± 6.4 cm; mean \pm SD). Participants were free from injuries and neuromotor disorders. The University of Nebraska Medical Center Institutional Review Board approved the study. All participants provided written informed consent.

2.2 Modular footwear

We developed a modular shoe that allows for altering the inclination of the foot relative to the ground without affecting other parameters, such as the amount of damping under different parts of the foot, and the bending stiffness of different parts of the outsole (Figure 1). We based our experimental shoes on a conventional shoe that does not provide support and has an entirely

level outsole (Chuck Taylor All-Star Low Top, Converse, Boston, MA). We used three shoe sizes
to accommodate participants (9.5, 10, and 10.5 men’s US sizes). Outsole geometry was altered
by attaching blocks with different heights to the sole. We chose a design with separate blocks as
opposed to using single wedge-shaped outsoles to avoid differences in outsole stiffness due to
the outsole inclination. The blocks were made from non-deformable material (1.27 and 0.32 cm
medium-density fiberboard) to avoid that different block heights would result in differences in
damping. Grip rubber was glued to the bottom of each block to prevent slipping. To keep the
anterior-posterior mass distribution constant, lead weight strips were added under the toe or heel
region of the insole depending on where heavier blocks were placed. We also added lead weights
in a pouch on top of the center of the shoe to match the total weight between all shoe
configurations within each participant, and avoid that differences in total mass would affect
metabolic rate [24–26]. The pouch was tightened down with rubber bands to prevent wobbling.
The modular shoes, including the added weights, amounted to a mass of 1.1 kg per side.

**[Insert figure 1 about here]**

2.3 Experimental conditions

We tested fifteen combinations of three different treadmill grades (-6° downhill, level and $+6^\circ$ uphill) and five footwear inclinations per treadmill condition (Figure 2). The footwear inclinations for each treadmill condition were approximately centered around the inclination that offset the treadmill grade. Footwear inclinations ranged from -3° to $+12^\circ$ during downhill walking, -7° to $+7^\circ$ during level walking and -12° to $+3^\circ$ during uphill walking. Footwear angles that result in the toes pointing down on a level treadmill are given a negative sign and labeled as “downward.” Footwear angles that result in the toes pointing up on a level treadmill are given a positive sign and labeled “upward”. One participant did not complete one of the conditions because one of the blocks of the outsole detached.

[Insert figure 2 about here]

2.4 Protocol

Before the experiment, we measured resting metabolic rate during a five-minute standing trial. Participants then performed a 10-minute warm-up [27] to become habituated with the shoe construction and treadmill before data collection. The treadmill speed was set at $1 \text{ m}\cdot\text{s}^{-1}$. All testing conditions lasted five minutes. Between conditions, we allowed the participants at least two minutes of rest. The conditions were semi-randomized where, in order to minimize the number of shoe inclination angle changes, we tested each shoe inclination on all applicable treadmill grades before changing to the next shoe inclination.

2.5 Measurements

We recorded oxygen consumption and carbon dioxide production using an indirect calorimetry system (Cosmed, K5, Rome, Italy) during the entirety of each condition. We recorded 3D kinematics at a rate of 200 Hz using a 14-camera motion capture system (VICON Vero, Oxford Metrics, Yarnton, UK). A total of 41 reflective markers were placed on anatomical landmarks and the shoes according to a modified Helen Hayes marker set [28]. The foot segment was defined by using markers placed at the calcaneus (lateral on the heel of the shoe, at furthest point from heel marker on rigid portion of counter), heel (on heel counter at the same height as the toe, centrally when viewing from a posterior position along the long axis of the shoe), first metatarsal (medially on shoe at a point approximating position of first metatarsal head), fifth metatarsal (laterally on shoe at a point approximating position of fifth metatarsal head), and toe (at a point approximating position of second metatarsal head on dorsum of shoe). We recorded ground

reaction forces (GRF) at a frequency of 2000 Hz using an instrumented split-belt treadmill (Bertec,
Columbus, OH, USA). We recorded muscle activation from the soleus, gastrocnemius medialis,
tibialis anterior, vastus medialis, rectus femoris, biceps femoris, and gluteus maximus muscles of
the right leg using a wireless electromyography (EMG) system (Trigno TM, Delsys, USA; 2000
7 Hz). Electrodes were positioned according to SENIAM guidelines [29]. Skin preparation prior to
8 attaching the electrodes included shaving hair from the recording site and wiping with an alcohol
swab. During the last minute of each condition, we recorded motion capture, ground reaction
forces and EMG data for 30 s.

17 **2.6 Data processing**

We used the Brockway equation to calculate the metabolic rate [30]. We averaged the metabolic
rate of the last two minutes of each condition to reflect the steady-state metabolic rate. The
metabolic cost of walking was calculated by subtracting the metabolic rate of the standing trial
from that of each walking condition.

[revised manuscript text omitted]

associated with c_1 will inform us whether a variable follows a non-horizontal trend versus foot
segment angle, or in the case that the trend is consistently parabolic, it will inform us whether the
location of the minimum is different from a foot segment angle of zero. The P-value associated
with c_2 will inform us whether a variable follows a parabolic trend versus foot segment angle. The
P-value associated with c_3 will inform us whether a variable follows a non-horizontal trend versus
treadmill grade and whether the location of the minimum is different from a level treadmill grade.
The P-value associated with c_4 will inform us whether a variable follows a parabolic trend versus
treadmill grade. The P-value associated with c_5 will inform us whether the trend of a variable
versus foot segment angle is different at treadmill grades. The landscape shape of the dependent
variables would be different depending on which coefficients in the statistical model are not
significantly different from zero (Figure 3). To avoid overfitting and adapt the model for dependent
variables that have linear trends, we removed terms that do not significantly contribute using
backward stepwise elimination similar to other studies [40,41].

To determine which parameters explain the variance in metabolic rate, we also evaluated
how different combinations of multiple biomechanical parameters relate to metabolic rate using
mixed-model ANOVAs of the following form:

$$\text{Metabolic rate} = c_0 + \sum_{i=1}^n c_i \cdot \text{biomechanical parameter}_i$$

(2)

We applied this approach to evaluate how changes in the metabolic rate are related to
changes in joint moments, joint powers, EMGs, GRFs, COM powers, and combinations of variable
types that were selected based on previous studies [31,32]. Backward stepwise elimination was
used to remove terms that do not significantly contribute [29,30]. We used the coefficient of
determination to compare how well biomechanical parameters explain the variance in metabolic
rate across all condition and participant combinations. All statistical analyses were performed in
MATLAB (MathWorks, Natick, MA, USA) and the significance threshold was set at 0.05.

**[Insert figure 3 about here]**

3. Results

[Insert table 1 about here]

3.1 Metabolic rate

We found significant effects on metabolic rate of the square of foot segment angle ($P = 0.013$; Table 1; Figure 43A), treadmill grade ($P < 0.001$; Table 1; Figure 43B) and the square of treadmill grade ($P < 0.001$; Table 1; Figure 43B). The first order of foot segment angle did not show a significant effect in the first iteration of the linear mixed-effects model analysis ($P = 0.318$) and was therefore removed. The resulting trends show that during walking on a level treadmill, shoes that result in a level foot segment angle minimized metabolic rate. As for the effect of treadmill angle, the resulting trends indicate that a -7° (downhill) treadmill grade minimized metabolic rate. We found a significant interaction effect on metabolic rate between foot segment angle and treadmill grade ($P = 0.023$; Table 1). As a result of this interaction effect, shoes that result in a level foot segment angle did not minimize metabolic rate at all treadmill grades. A -3° (downward) foot segment angle (or a $+3^\circ$ upward outsole) minimized metabolic rate during walking on a -6° (downhill) treadmill. A $+3^\circ$ (upward) foot segment angle (or a -3° downward outsole) minimized metabolic rate during walking on a $+6^\circ$ (uphill) treadmill.

[Insert figure 43 about here]

3.2 GRF and COM mechanics

Positive COM work rate (Figure 5B) and average propulsive GRF (Figure S4, B) follow a rising parabolic trend versus treadmill grade ($P < 0.001$; Table 1). By extrapolating our data, we found that the positive COM work rate and average propulsive GRF are minimized around a downhill treadmill grade of -8° . Smaller downhill and greater uphill treadmill grades led to decreasing magnitudes of average braking GRF ($P < 0.001$; Table 1; Figure S4, D) and decreasing magnitudes of negative COM work rate ($P < 0.001$; Table 1; Figure S4, F). In summary, greater uphill treadmill grades resulted to more propulsion and less braking.

We found no isolated effects of foot segment angle on average positive and negative GRFs, and COM work rates, but we did find significant interaction effects of foot segment angle and treadmill grade on average braking GRF ($P = 0.032$; Table 1) and on positive COM work rate ($P = 0.015$; Table 1). During downhill walking, smaller negative (downward) and greater positive

(upward) foot segment angles led to decreasing positive COM work rate (Figure 5A) and increasing magnitudes of average braking GRF (Figure S4, C). During uphill walking, smaller negative (downward) and greater positive (upward) foot segment angles led to decreasing positive COM work rate (Figure 5A) and decreasing magnitudes of average braking GRF (Figure S4, C). In summary, we found interaction effects of foot segment angle and treadmill grade on braking GRF and positive COM work rate that were similar to the interaction effect on metabolic rate.

[Insert figure 5 about here]

3.32 Ankle mechanics and muscle activation

We found a U-shaped trend in average soleus EMG versus foot segment angle with a minimum at a 0° foot segment angle during downhill, level and uphill walking ($P = 0.002$; Table 1; Figure 64A). Smaller negative (downward) and greater positive (upward) foot segment angles led to increasing average plantarflexion moment ($P < 0.001$; Table 1; Figure S1, A), increasing average positive ankle ~~power-work rate~~ ($P = 0.001$; Table 1; Figure S1, E), increasing magnitudes of ~~average~~-negative ankle ~~work rate power~~ ($P = 0.049$; Table 1; Figure S1, G) and increasing average tibialis anterior EMG ($P < 0.001$; Table 1; Figure S1, K). Smaller negative (downward) and greater positive (upward) foot segment angles led to decreasing magnitudes of average dorsiflexion moment ($P < 0.001$; Table S1; Figure S1, C). In summary, a level foot segment angle minimized soleus EMG and more upward foot segment angles led to increases in most ankle parameters.

Smaller downhill and greater uphill treadmill grades led to increasing ~~average~~-positive ankle ~~work rate power~~ ($P = 0.001$; Table 1; Figure S1, F) and increasing average gastrocnemius medialis EMG ($P < 0.001$; Table 1; Figure S1, J). Smaller downhill and greater uphill treadmill grades led to decreasing average plantarflexion moment ($P < 0.001$; Table 1; Figure S1, B) and decreasing magnitudes of ~~average~~-negative ankle ~~work rate power~~ ($P < 0.001$; Table 1; Figure S1, H). We found U-shaped trends in average soleus EMG ($P < 0.001$; Table 1; Figure 64B) and average tibialis anterior EMG ($P < 0.001$; Table 1; Figure S1, L) versus treadmill grade with respective minima at downhill treadmill grades of -3° and -2°. ~~In summary, more upward foot segment angles led to increases in certain ankle parameters and decreases in other ankle parameters.~~

[Insert figure 64 about here]

3.3 GRF and COM mechanics

Average positive COM (Figure 5B) and average propulsive GRF (Figure S4, B) follow a rising parabolic trend versus treadmill grade ($P < 0.001$; Table 1). By extrapolating our data, we found that the average positive COM power and average propulsive GRF are minimized around a downhill treadmill grade of -8° . Smaller downhill and greater uphill treadmill grades led to decreasing magnitudes of average braking GRF ($P < 0.001$; Table 1; Figure S4, D) and decreasing magnitudes of average negative COM power ($P < 0.001$; Table 1; Figure S4, F). In summary, greater upward uphill treadmill grades foot segment angles resulted to more propulsion and less braking.

We found no isolated effects of foot segment angle on average positive and negative GRFs, and average COM powers, but we did find significant interaction effects of foot segment angle and treadmill grade on average braking GRF ($P = 0.015$; Table 1) and on average positive COM power ($P = 0.032$; Table 1). During downhill walking, smaller negative (downward) and greater positive (upward) foot segment angles led to decreasing average positive COM power and increasing magnitudes of average braking GRF. During uphill walking, smaller negative (downward) and greater positive (upward) foot segment angles led to decreasing average positive COM power and decreasing magnitudes of average braking GRF. In summary, we found interaction effects of foot segment angle and treadmill grade on braking GRF and positive COM power that were similar to the interaction effect on metabolic rate.

[Insert figure 5 about here]

3.4 Explained metabolic rate variance based on different biomechanical variables

We found the highest explained variances in metabolic rate using a statistical model with only average positive COM power ($R^2 = 0.91$; $P < 0.001$; Table 2; Figure 6) and a model with only average propulsive GRF by itself ($R^2 = 0.92$; $P < 0.001$; Table 2; Figure S6, C). A model with all average positive and negative joint moments except average dorsiflexion moment explained 65% of the variance in metabolic rate ($R^2 = 0.65$; P -values < 0.001 ; Table 2; Figure S6, A). A model with all average joint powers except average negative knee power explained 88% of the variance in metabolic rate ($R^2 = 0.88$; P -values < 0.001 ; Table 2; Figure S6, B). A model with average EMGs of the gastrocnemius medialis, biceps femoris, and gluteus maximus explained 73% of the variance in metabolic rate ($R^2 = 0.73$; P -values < 0.001 ; Table 2; Figure S6, D) except average EMGs of the tibialis anterior, soleus, vastus medialis and rectus femoris muscles that did not significantly contribute. A model with squared average EMGs of the gastrocnemius medialis, biceps femoris, vastus medialis, and gluteus maximus explained 71% of the variance in metabolic rate ($R^2 = 0.71$; P -values < 0.001 ; Table 2; Figure S6, E) except squared average EMGs of tibialis anterior, soleus and rectus femoris muscles that did not significantly contribute. A combined model with all average joint moments and powers except average positive ankle moment, average negative ankle power and average positive and negative hip power explained 91% of the variance in metabolic rate ($R^2 = 0.91$; P -values < 0.001 ; Table 3; Figure S6, F). A combined model that included a number of peak joint angles in addition to average joint moments and average joint powers did not explain a higher percentage of the variance in metabolic rate ($R^2 = 0.86$; P -values < 0.001 ; Table 3; Figure S6, G). In summary, we found that adding multiple joint parameters did not improve the explained variance in metabolic rate compared to models that used COM power or propulsion GRF to explain the variance in metabolic rate.

[Insert table 2 about here]

[Insert figure 6 about here]

[Insert table 3 about here]

4. Discussion

The aim of the present study was to investigate the interaction effects of varying footwear outsole geometry and treadmill grade on metabolic rate, joint mechanics, muscle activity, ground reaction forces and COM mechanics. During level walking, we found that a shoe inclination of 0° minimized metabolic rate (Figure 43A). Remarkably, we found that a -3° foot segment angle (obtained with a 3° upward shoe) minimized metabolic rate during walking on a -6° downhill grade. A $+3^\circ$ foot segment angle (obtained with a -3° downward shoe) minimized metabolic rate during walking on a $+6^\circ$ uphill grade.

Extrapolating the trend of metabolic rate versus treadmill grade showed that the treadmill grade that minimizes metabolic rate was within 1° to the tested downhill treadmill grade of -6° (Figure 4B). Studies from Margaria, [7] and Minetti et al., [9] also found that the metabolic rate of walking is minimized around a -6° downhill grade. The minima for positive COM work rate and propulsive GRF were very close to the treadmill grade that minimized metabolic rate suggesting that these parameters possibly contribute to metabolic rate being minimal around a -6° downhill treadmill grade (Figs. 5B and S4, B).

The decreases in negative COM work rate and braking GRF with smaller downhill and greater uphill walking grades are consistent with previous studies [10,42–46]. The increases in positive COM work rate are in line with a study from Franz et al., [10] and the increases in propulsion GRF are in agreement with a number of studies [10,42–46]. We found that smaller downhill and greater uphill walking also led to increasing dorsiflexion moments, positive ankle work rate, tibialis anterior, and plantarflexor muscle activations. These results appear to be consistent with studies that found increased positive ankle work rate [47–49], tibialis anterior [50,51] and plantarflexor muscle activations [11,12,50,51] during uphill walking. Additionally, we found that uphill walking decreased magnitudes of negative ankle work rate similar to other studies [47–49].

The fact that level shoes minimized metabolic rate during walking on a level treadmill could be related to reduced effort of the ankle joint. If we assume that shoes that result in upward foot segment angles would lead to longer plantarflexor contractile element lengths, then if all other parameters would stay constant, based on the energetics model from Umberger et al., [17] we would expect lower metabolic rate values in conditions with upward foot segment angles. In contrast to this hypothesis, we found increasing metabolic rates in conditions with greater upward foot segment angles. Knowing whether upward shoe conditions led to longer contractile elements lengths would require additional measurements (e.g., ultrasonography). It is also likely that other

parameters that affect metabolic rate of muscles were induced such as contractile element force,
velocity, and muscle activation. As a matter of fact, we found that soleus muscle activity showed
a descending-rising trend with a minimum that aligned with a foot segment angle of 0° which
minimized metabolic rate (Figure 6A)~~Soleus muscle activity was minimal with a foot segment of~~
~~0° (Figure 4A).~~ A musculoskeletal simulation from Dorn et al., [52] suggests that the soleus
consumes the greatest amount of metabolic energy from all muscles during level walking which
could explain the similar U-shaped relationship in metabolic rate found in our study between
soleus muscle activity and foot segment angles. It could also be that the U-shaped landscape in
metabolic rate is the net result of the summation of metabolic energy consumptions in muscles
that have increasing trends with smaller negative (downward) and greater positive (upward) foot
segment angles, and other muscles that have decreasing trends with smaller negative
(downward) and greater positive (upward) foot segment angles. In our results, we did indeed find
variables that increased with smaller negative (downward) and greater positive (upward) foot
segment angles such as average plantarflexion moment (Figure S1, A) as well as variables that
decrease smaller negative (downward) and greater positive (upward) foot segment angles, such
as average dorsiflexion moment (Figure S1, C) and a number of knee variables (Figure S3, A, G
and I).

Most changes in ankle moments, work rates powers and muscle activations (Figs. 64 and
S1) appear to be consistent with prior studies. Panizzolo et al., [53] examined changes in
kinematic and kinetic parameters in response to stepping with different parts of the foot on a small
bump. Similar to our study, they found increasing plantarflexion moment, increasing plantarflexion
work rate power and decreasing magnitudes of dorsiflexion moment when stepping on a bump
with the forefoot, which results in a foot angle similar to our upward shoe conditions. Simonsen et
al., [54] and Stefanyshyn et al., [55] also found higher plantarflexion moments in level shoes
compared to high-heeled (i.e., downward) shoes. This could be due to a longer moment arm of
the Achilles tendon in level shoes compared to high-heeled shoes [55]. We found increasing
tibialis anterior activation with smaller negative (downward) and greater positive (upward) foot
segment angles. Inspection of time series plots shows that this occurred during the swing phase
and could have been due to an increased need for toe clearance. Curiously, we found no
significant effect of foot segment angle on gastrocnemius medialis muscle activity. Similarly,
Stefanyshyn et al., [55] did not find an effect of heel height on gastrocnemius medialis muscle
activity whereas other studies did find differences in gastrocnemius medialis muscle activity in
high-heel and negative heel shoes compared to level shoes [56,57]. The absence of a significant

effect of foot segment angle in our study could be due to the fact that the modular shoes were
designed not to affect bending stiffness, which is different from most high-heel or low heel shoes.

~~Extrapolating the trend of metabolic rate versus treadmill grade showed that the treadmill~~
~~grade that minimizes metabolic rate was within -1° to the tested downhill treadmill grade of -6°~~
~~(Figure 43B). Studies from Margaria, [7] and Minetti et al., [9] also found that the metabolic rate~~
~~of walking is minimized around a -6° downhill grade. The minima for average positive COM work~~
~~ratepower and propulsive GRF were very close to the treadmill grade that minimized metabolic~~
~~rate suggesting that these parameters possibly contribute to metabolic rate being minimal around~~
~~a -6° downhill treadmill grade (Figs. 5B and S4, B).~~

~~The decreases in negative COM work rate power and braking GRF with smaller downhill~~
~~and greater uphill walking grades are consistent with previous studies [10,48–52]. The increases~~
~~in positive COM work ratepower are in line with a study from Franz et al., [10] and the increases~~
~~in propulsion GRF are in agreement with a number of studies [10,48–52]. We found that smaller~~
~~downhill and greater uphill walking also led to increasing dorsiflexion moments, positive ankle~~
~~work ratepower, tibialis anterior, and plantarflexor muscle activations. These results appear to be~~
~~consistent with studies that found increased positive ankle work rate power [53–55], tibialis~~
~~anterior [56,57] and plantarflexor muscle activations [11,12,56,57] during uphill walking.~~
~~Additionally, we found that uphill walking decreased magnitudes of negative ankle work rate~~
~~power similar to other studies [53–55].~~

Contrary to our expectations, we did not find a shoe that exactly offsets the treadmill grade
minimizes the metabolic rate for uphill and downhill human walking. A -3° foot segment angle
reduced metabolic rate (by $\approx 8\%$ compared to level shoes) during downhill walking and a 3° foot
segment angle reduced metabolic rate (by $\approx 2\%$ compared to level shoes) during uphill walking.
We found two similar interaction effects on biomechanical parameters; more downward foot
segment angles reduced positive COM work rate power and positive knee work rate power during
downhill walking whereas more upward foot segment angles reduced positive COM work rate
power (Figure 5A) and positive knee work rate power (Figure 7) during uphill walking. These
interaction effects between foot segment angle and treadmill grade could possibly explain that a
more downward foot segment angle minimized metabolic rate during downhill walking, and a more
upward foot segment angle minimized metabolic rate during uphill walking.

[Insert figure 7 about here]

In summary, we found several changes in biomechanical variables and metabolic rate. To
answer which biomechanical variables contribute the most to changes in metabolic rate, we
evaluated the relationships between groups of biomechanical variables and metabolic rate.
Interestingly, we found the highest explained variances in metabolic rate using a univariate model
with only average propulsive GRF and a univariate model with only average positive COM power.
We found that a model with only joint powers had a higher explained variance than a model with
only joint moments. This shows that under the conditions of our study, joint powers are more
related to changes in metabolic rate than joint moments. We also found that average negative
COM power and average negative knee power did not significantly contribute. This could be
because negative mechanical work requires less metabolic energy than positive mechanical work
[7,48]. A model with average muscle activity of the gastrocnemius medialis, biceps femoris, and
gluteus maximus explained 73% of the variance in metabolic rate. This is consistent with Jackson
and Collins, [31] and Silder et al., [32] studies that show that an unweighted or weighted sum of
muscle activities can be related to changes in metabolic rate. Simulation studies often assume
that the sum of squares of muscle activities is related to metabolic cost [49,50]. Similar to Silder
et al., [32] study, we found that changes in metabolic rate are related to changes in a combination
of squared muscle activities. However, using squared muscle activities did not show a higher
explained variance in metabolic rate compared to using first-order terms of muscle activities.
Silder et al., [32] showed that a combination of a joint angle (peak knee flexion) with a number of
peak joint moments could explain 89% of the variance in metabolic rate during walking on different
inclines. In our study, we found that a combination of joint angles, joint moments and joint powers
explained 86% of the variance in metabolic rate. We also tested a model with only joint moments
and joint powers and found that this explained a higher percentage of the variance in metabolic
rate than the model with joint angles, joint moments and joint powers. Analyzing relationships
between groups of variables this way yields coefficients that do not represent the relationship of
each individual parameter with metabolic rate, and it could be possible to have certain non-
intuitive coefficients. For example, when analyzing the relationship between joint moments and
metabolic rate we found a negative coefficient for plantarflexion moment. Similar non-intuitive
coefficients were found in the study from Silder et al., [32] that reported relationships between
groups of biomechanical variables and metabolic rate.

A possible limitation of our study is that the weight of our modular shoes (≈ 1.1 kg) is
higher than normal shoes which typically range from 0.3 to 0.5 kg. Although we matched the total
weight and the front-back weight distribution, there might have been differences in moment of
inertia due to the mass being either closer or further from the center of the shoe in different

footwear conditions. We designed outsoles that were segmented in separate blocks to avoid
altering sole bending stiffness. This construction is different from normal outsoles. However, none
of the participants commented that it was difficult to walk or to maintain balance. Although the
participants walked comfortably with different shoes, it is possible that lower and perhaps different
metabolic rate landscapes could have resulted if participants were given a longer habituation time
(e.g., multiple days). The maximum downward shoe inclination that we tested appears to be within
the range of moderate high-heeled shoes. The range of shoe inclinations that were tested per
treadmill grade (15° from most downward to most upward) was higher than the range of treadmill
grades (12°). Despite the considerable range of tested shoe inclinations, the effects of shoe
inclination were relatively shallow and outweighed by the effects of treadmill grade~~However, the~~
~~effects of treadmill grade on metabolic rate outweighed the effects of foot segment angle which~~
~~could have affected the resulting statistical trends.~~ It is possible that different results might have
been found if we had tested a larger number of participants or higher range of shoe inclinations.
While the number of participants and range of shoe inclinations have been adequate to reject the
null hypothesis for 5 of the 6 coefficients of our statistical model for metabolic rate (P-values \leq
0.023), we have no certainty whether the non-significant effect of the first order of foot segment
angle on metabolic rate (P = 0.318) is related to our sample size (n = 10). On the other hand, the
finding that metabolic rate remains relatively constant over a range of downward and upward shoe
inclinations appears to demonstrate how the foot and ankle ~~could~~ are capable to adapt to uneven
terrain without a large metabolic penalty [58].

Because of the large number of conditions being tested and the changes in metabolic rate
and muscle activity that were found, we expect that the dataset provided by this study can be
useful for testing and validating musculoskeletal simulations to estimate the metabolic rate of
locomotion. Even though the effects of foot segment angles on metabolic cost were relatively
small, they could be clinically meaningful. Changes of 7% in walking speed have been considered
clinically meaningful in elderly and stroke patients [59]. By slightly extrapolating the trend in shoe
conditions in the level treadmill condition, we can estimate that a shoe with a 9° inclination could
cause a similar percentage change in metabolic rate. This could be used as a potential guideline
to inform patient populations with limited effort capacity to use shoes with a lesser inclination.
The results of our study could also inform designs of shoes for walking on rolling terrain. Since
humans typically require more time and have a higher metabolic rate during uphill portions of
terrain with uphill and downhill grades, it could be advantageous to use shoes with a slight
downward shoe inclination that are closer to optimal at these uphill portions. Perhaps this also
explains why most existing shoes have a small heel to toe drop (a typical offset of 1 cm over a

shoe length of 28 cm corresponds to a downward shoe inclination of -2°). It could also be possible
to design shoes that allow changing the inclination depending on the grade of the terrain (e.g.,
similar to heel lift on snow shoes or alpine touring skis). Our data could further be related to
differences in metabolic rate between walking on stairs and ramps at an equivalent average
grade. At a 6° grade, we found that the shoe condition that offsets the average grade (similar to
a stairway) was 13% more metabolically economic than walking down the same grade with level
(0° shoe inclination) shoes. If we extrapolate this difference to the average grade for which stairs
are recommended (37° based on California Code of Regulations,
<https://www.dir.ca.gov/title8/3231.html>), we can estimate that a shoe that offsets the average
grade could be 82% more metabolically economic. This extrapolation should be interpreted with
caution since the grade that we tested was much lower than 37° , and walking on stairs can
constrain foot placement which is different from our modular shoe experiment. Finally, a future
experiment could benefit from developing a more lightweight outsole (e.g., through 3D printing)
as this would not require as much added mass to keep the total mass and mass distribution
constant. These modular shoes could ~~_also_~~ be further used as a research instrument to optimize
parameters other than metabolic rate, such as minimizing joint loading to prevent injuries.

**5. Conclusions**

Shoes that are in between level shoes and shoes that entirely offset the treadmill grade minimized
the metabolic rate of downhill and uphill human walking. We found that optimal shoe inclinations
are primarily related to trends in soleus muscle activity, and interaction effects in positive COM
~~work rate power~~ and positive knee ~~work rate power~~ could explain why different foot segment
angles were optimal at different treadmill grades. Finally, we found that the optimal treadmill grade
was about the same as the treadmill grade that minimizes positive COM ~~work rate power~~ and
propulsive GRFs. ~~Of note, both of these parameters by themselves explained most of the~~

[revised manuscript text omitted]

59. Perera S, Mody SH, Woodman RC, Studenski SA. 2006 Meaningful Change and
Responsiveness in Common Physical Performance Measures in Older Adults. *J. Am.*
*Geriatr. Soc.* **54**, 743–749. (doi:10.1111/j.1532-5415.2006.00701.x)

Figure legends

Figure 1. Modular outsole geometry shoe assembly. By attaching wooden blocks of different heights on the sole, we were able to change outsole geometry without altering bending stiffness or damping (blocks are only shown in the rear part of the shoe). We placed thin lead strips under the forefoot or heel of the shoe to keep the anterior-posterior weight distribution constant across shoes with different geometries. Finally, we attached a pouch with lead weights on top of the center of the shoe to match the total weight between shoe configurations.

Figure 2. Experimental conditions. A: Definition of shoe and treadmill angles. The outsole angle was defined as the angle of the shoe versus the bottom of blocks. Treadmill angle was defined as the angle of the treadmill versus the horizontal plane. The foot segment angle was defined as the angle of the foot versus the horizontal plane. B: We evaluated five different shoe inclinations at three treadmill grades (shown on the vertical axis): downhill (-6°), level and uphill (6°). During downhill walking, we tested shoe inclinations (shown as blue angle indicators) ranging from: -3° to $+12^\circ$ (downward shoe inclinations are labeled negative, upward shoe inclinations are labeled positive). During level walking, we tested shoe inclinations ranging from -7° to $+7^\circ$. During uphill walking, we tested shoe inclinations ranging from -12° to $+3^\circ$. Colored bands connect shoe inclinations that were tested at multiple treadmill grades. The combination of the shoe inclinations and treadmill grades resulted in foot segment angles (shown on the horizontal axis) ranging from -9° to 9° .

Figure 3. Role of significance testing of each coefficient in the statistical model. Subplots show the landscape shape of a variable when certain coefficients are not significantly different from zero.

Figure 43. Metabolic rate. A: Change in metabolic rate versus foot segment angle during downhill, level, and uphill walking. Black lines represent the formula from the linear mixed-effects model analysis mixed-model ANOVA evaluated over the tested range of foot segment angles at each treadmill grade. Grey crosses indicate the interpolated minima \pm s.e.m. B: Change in metabolic rate versus treadmill grade during walking for different shoe inclinations. Colored lines represent the formula from the linear mixed-effects model analysis mixed-model ANOVA evaluated over the

[revised manuscript text omitted]

**Figure 6.** Relationship between center of mass power and metabolic rate. Measured metabolic
rate vs. estimate metabolic rate based on COM power. R^2 value represents the coefficient of
determination of measured metabolic rate versus estimated metabolic rate over all trials from all
participants based on model from mixed-model ANOVA. Blue and cyan downward-pointing
triangles represent mean values of conditions with downward shoe inclinations. Green diamonds
represent mean values of conditions with level shoes. Orange and red upward-pointing triangles
represent mean values of conditions with upward shoe inclinations. Error bars are s.e.m. The
black line is the identity line.

**Figure 7.** Positive knee powerwork rate. A: PAverage positive (pos.) knee work rate power
changes versus foot segment angle during downhill, level, and uphill walking. Black lines
represent the formula from the linear mixed-effects model analysis mixed-model ANOVA
evaluated over the tested range of foot segment angle at each treadmill grade. B: P-Average
positive (pos.) knee work rate power changes versus treadmill grade during walking in different
shoe inclinations. Colored lines represent the formula from the linear mixed-effects model analysis
mixed-model ANOVA evaluated over the range of treadmill grades for the three shoe inclinations
that were tested on all treadmill grades. Blue and cyan downward-pointing triangles represent
mean values of conditions with downward shoe inclinations. Green diamonds represent mean
values of conditions with level shoes. Orange and red upward-pointing triangles represent mean
values of conditions with upward shoe inclinations. Error bars are s.e.m.

Table 1. Effects of independent condition variables on metabolic rate and biomechanical variables. Results of linear mixed-effects model analysis (random effect: participant; fixed effects: first and second order of foot segment angle and treadmill grade; outcome parameters: metabolic rate, ankle mechanics, COM and GRF mechanics). Values indicate resulting equation coefficients.

Variables	Intercept coefficient	Foot segment angle	Foot segment angle ²	Treadmill grade	Treadmill grade ²	Foot segment angle · Treadmill grade
Metabolic rate (W·kg ⁻¹)	3.35**	NA	0.003*	0.41**	0.03**	-0.003*
Average plantarflexion moment (N·m·kg ⁻¹)	0.40**	0.006**	NA	-0.004**	NA	NA
Average dorsiflexion moment (N·m·kg ⁻¹)	-0.01**	0.0003**	NA	NA	NA	NA
Positive ankle work rate (W·kg ⁻¹)	0.18**	0.001**	NA	0.01**	0.0003*	NA
Negative ankle work rate (W·kg ⁻¹)	-0.15**	-0.0009*	NA	0.01*	-0.0008**	NA
Average soleus EMG (%)	26.93**	NA	0.03**	0.67**	0.08**	NA
Average gastrocnemius medialis (%)	26.21**	NA	NA	0.99**	NA	NA
Average tibialis anterior (%)	31.28**	0.41**	NA	0.39**	0.09**	NA
Positive COM work rate (W·kg ⁻¹)	0.18**	NA	NA	0.03**	0.002**	-0.0001*
Negative COM work rate (W·kg ⁻¹)	-0.19**	NA	NA	0.04**	-0.002**	NA
Average positive parallel GRF (BW)	0.02**	NA	NA	0.003**	0.0002**	NA
Average negative parallel GRF (BW)	-0.02**	NA	NA	0.004**	-0.0001**	0.00001*

*p < 0.05, **p < 0.01

NA = not applicable to final model due to non-significant contribution

BW, bodyweight

Answers to revision

“Modular footwear that partially offsets downhill or uphill grades minimizes the metabolic cost of human walking”

We would like to thank the referees and the editor for their time and valuable comments. We sincerely appreciate that our manuscript is considered transferring to the Royal Society Open Science. Below are point-by-point answers to all comments. Original comments are presented in black font, and answers are formulated in blue color. Manuscript text is indicated in *italic*, and changes are underlined. In addition to this response document, we also provide a revised manuscript document with tracked changes and a version without tracked changes.

Referee: 1

Comments to the Author

The study described aimed to show the effects of adjusting shoe outsole geometry on the mechanics and energetic cost of walking on surfaces of different grades. Specifically, it was predicted that metabolic cost of walking would be minimized by shoes that offset the surface gradient such that the foot was level to horizontal at foot flat. Furthermore, the study attempted to provide insight into biomechanical variables that explain the observed trends in metabolic cost. To achieve these aims, the authors performed spirometry and gait analysis measurements during level, inclined, and decline walking. For each surface grade, 5 different shoe outsoles designed to manipulate the horizontal angle of the foot from below horizontal to above were tested. To address the first aim of the study, the authors employed a mixed-model anova with metabolic power as the dependent variable, and foot segment angle and treadmill grade as fixed effects. For initial biomechanical analysis, the same mixed-model was run with biomechanical and EMG variables as the dependent variable. Further mixed models were then run on various groups of dependent variables to describe what variables predict metabolic cost across all conditions.

General Comments:

Overall I consider there to be interesting aspects to the study presented and an elegant experimental manipulation, but found the analysis and writing to lack focus. The aims, intervention, and initial analysis seem novel and interesting, but are somewhat lost in the vast array of data presented and discussed. I think a manuscript more focussed on the main question (i.e. do footwear that offset gradient reduce metabolic cost) would read better and make a more valuable contribution to the literature. I have been more specific below.

We thank the referee for the thorough analysis of the strengths and areas of improvement of our work.

R1.Q1.a) My main comment would be that I do not see novelty or relevance to the research question of the latter part of data analysis. Specifically I refer to the running of many mixed models with groups of variables as effects, to see which best predict metabolic cost across the whole data set. I have several comments on this:

First, this does not really contribute to the aims/hypotheses of the experiment. Yes, the authors have added a final aim to perform this analysis, but it feels rather like an add-on in a bid to present a large amount of data.

Second, even if one takes the final aim (“to explain the changes in metabolic rate, provide validation data for predictive models, and inform footwear designs”) at face value, it does not really describe an aim to answer a question, but more to produce a public data set. I am all for publishing data sets to accompany articles, but this should not be part of the manuscript aims.

Most importantly, I do not feel it is valuable to pool all the data across conditions and attempt to
predict metabolic cost with biomechanical or EMG variables. Primarily this is because metabolic
cost is so overwhelmingly dominated by treadmill gradient, such that the nuances of the
experiment are lost. Essentially, this part of the analysis becomes about the mechanics and
energetics of slope walking. It is shown that metabolic cost is predicted by the increased overall
mechanical work associated with walking on an incline. This has been known for a long time and
has been shown multiple times by the studies that are cited in the manuscript. The authors do
show some predictive value of EMG in determining the metabolic cost of walking on gradients,
but this has also been shown previously (as the authors acknowledge in the manuscript). A large
part of the discussion (pages 13/14) ends up dedicated to stating that this analysis confirms
previous findings. I think this space would be better dedicated to a mechanistic discussion of the
observed relationships between outsole angle, treadmill gradient, and mechanics and energetics.
b) Finally, analysing downhill and uphill data all together seems hazardous. Fundamentally these
are different tasks with different mechanical factors dictating the metabolic cost. I think it is unlikely
that the same mechanical variable(s) will dominate the cost of downhill and uphill locomotion.
Thus, by lumping them together, you run the risk of missing important findings.
By focusing the manuscript more, much of the data in tables and figures could be left out of the
main manuscript, making the key findings easier to discern.

a) We thank the referee for expressing his/her concerns about the latter part of our analysis. We
agree that the effects are dominated by the effects of uphill walking and our initial final aim does
not answer a question.

We also agree that a lot of the discussion revolved around stating which results confirmed
previous findings. We originally tried to structure our discussion based on guidelines from Brand
and Huijckes, (2001) that recommend mentioning which results are 'in the ballpark' of results from
the literature, but we understand the point of view from the referee that this is not very interesting.

**We have therefore removed the following sections from our manuscript:**

- **The final aim** (analysis of which biomechanical variables explain changes in metabolic rate).
- **Results section 3.4, Tables 2-3, Figure S6, and the associated methods and discussion**
**parts.**

We have moved Figure 6 from the main manuscript to the supplementary material. We have
added a sentence in the supplementary figure caption to acknowledge that the trend of positive
COM work rate explaining most of the variance in metabolic rate is likely due to the overwhelming
effects of the uphill walking conditions.

b) To address the sub-question whether lumping the treadmill conditions together affected the
results, we conducted one supplementary analysis in which we analyzed the effects of foot
segment angle variables on metabolic rate within only the 5 level walking conditions.

$$\text{Metabolic rate} = c_0 + c_1 \cdot \text{foot segment angle} + c_2 \cdot \text{foot segment angle}^2$$

Using only the level walking conditions, we found an intercept (c_0) with a P-value of $2 \cdot 10^{-23}$ and a
positive coefficient for the square of foot segment angle with a P-value of 0.0506. **This confirms**
**that the second-order effect of foot segment angle that was found in the overall analysis**
**is also present when considering only the level treadmill condition.**

Additional reference not in manuscript:

Brand RA, Huiskes R. Structural outline of an archival paper for the Journal of Biomechanics. Journal of biomechanics. 2001 Nov 1;34(11):1371-4.

R1.Q2. Considering the initial analysis of how foot angle and treadmill grade relate to metabolic cost,

a) I wonder if the metabolic measurements really show a clear minimum at a specific foot angle. Measurements of metabolic power by gas analysis are not particularly sensitive, and the fits in figure 3A show quite shallow curves for each treadmill grade.

b) While it seems reasonable to state that metabolic cost is increased at the extremes, I think the authors could discuss more the magnitude of effects of outsole angle, and how far from a zero foot angle one has to deviate to generate a meaningful change in metabolic cost.

c) As the data appears presently, a linear fit might be as good as the quadratic fits presented. This is particularly worthy of consideration with reference to the repeatability of the metabolic power measures.

We thank the referee for these very insightful comments.

a) We acknowledge that the effects are shallow. But we think that this is not unique to our experiment. For example, it is known that the metabolic landscapes of speed or step frequency are relatively shallow when one stays close to the optimal values [Ralston, 1958; Zarrugh et al., 1974] and this allows humans to walk in a variety of ways without considerable increased metabolic cost (if the landscapes were steep then any small perturbation would have a large cost). The effects of the stiffness setting of an elastic ankle exoskeleton also appear to be shallow and variable across participants similar to our experiment [Collins et al., 2015; Figure R1].

Additional figure (not in manuscript) from Collins et al., [2015]. Figure R1. Effects of exoskeleton stiffness on metabolic rate. Colored lines show the individual trends (every color is one participant). The thick black line shows the average trend. This plot is generated using the online supplementary data from this paper.

Similar to these examples from other studies, the metabolic landscape in our data is often not
visible in a single participant but only becomes apparent after performing curve fits through data
from all participants and conditions together. **In summary, this type of shallow landscape is**
**similar to other studies.**

**We have added the following (underlined) text to acknowledge this in the discussion:**

*The range of shoe inclinations that were tested per treadmill grade (15° from most downward to*
*most upward) was higher than the range of treadmill grades (12°). Despite the considerable range*
*of tested shoe inclinations, the effects of shoe inclination were relatively shallow and outweighed*
*by the effects of treadmill grade.*

b) We strongly like the suggestion of the referee to analyze how far the outsole angle has to
deviate to generate a meaningful change in metabolic rate.

To choose a threshold for a meaningful change, we based ourselves on a highly cited paper from
Perera et al., [2006] that reports changes in preferred walking speed around 7% as clinically
meaningful in older adults and people with stroke.

By slightly extrapolating the trend from linear mixed-effects model analysis (Figure 4a in the
manuscript), **we can conclude that a deviation in foot segment angle of 8.6° is required to**
**result in a ‘clinically meaningful’ 7% increase in metabolic rate (i.e., about 1.6° more than**
**the steepest shoe inclination in the level treadmill condition).**

**We have added the following text in the discussion:**

*Even though the effects of foot segment angles on metabolic cost were relatively small, they could*
*be clinically meaningful. Changes of 7% in walking speed have been considered clinically*
*meaningful in elderly and stroke patients [59]. By slightly extrapolating the trend in shoe conditions*
*in the level treadmill condition, we can estimate that a shoe with a 9° inclination could cause a*
*similar percentage change in metabolic rate. This could be used as a potential guideline to inform*
*patient populations with limited effort capacity to use shoes with a lesser inclination.*

c) To answer the question whether not including a second-order term would fit the data better, **we**
**conducted a supplementary analysis in which we started from the following model (without**
**a second-order term for foot segment angle):**

$\text{Metabolic rate} = c_0 + c_1 \cdot \text{foot segment angle} + c_2 \cdot \text{treadmill grade} + c_3 \cdot \text{treadmill grade}^2 + c_4 \text{foot}$
$\text{segment angle} \cdot \text{treadmill grade}$

When evaluating this model, we found significant P-values for the intercept term, treadmill grade,
and the square of treadmill angle but we found no significant effect (or interactions) of the first
order of foot segment angle ($P = 0.331$). Therefore, we excluded the ‘foot segment angle’ and
evaluated the model in a next iteration (i.e., backward stepwise elimination; similar to Liew et al.,
2016 and Antonellis et al., 2018).

Since the effect of the first order of foot segment angle is non-significant, the three curve fits in
the left panel of Figure R2 are horizontal lines.

Additional figure (not in manuscript) Figure R2. Effects of foot segment angle and treadmill angle on metabolic rate evaluated with the statistical model without a second-order term for foot segment angle.

Although it could be possible to fit the data with a model that does not include a second-order term for the effect of foot segment angle, we prefer not to go this route because:

- We theoretically know that the underlying landscape must have a U-shape simply because we know that at some point walking with shoes that result in a very high downward foot angle and walking with shoes that result in a very high upward angle becomes hard.
- The resulting horizontal linear fits do not seem to capture the observation when pilot testing that certain shoes are harder to walk with than others.

We were instructed by a biostatistician that linear mixed-effects model analysis is an appropriate method for analyzing the effects of continuous independent variables on a continuous dependent variable. In contrast to methods like linear regression, linear mixed-effects model analysis provides P-values that can inform whether certain terms are consistent across participants. **The fact that the resulting P-value for the square of foot segment angle is significant ($P = 0.013$) shows that a downward parabolic trend is relatively consistent across participants.**

The significant P-value for the square of foot segment angle is NOT an indication of whether or not the landscape is shallow. We agree with the referee that the effect is shallow. We also believe that a model with a second-order term for the effect of foot segment angle is advantageous because it allows analyzing how shallow (or not) the landscape is (as we did in response to sub-question b) above). If we were to use a linear curve fit, we would not be able to discuss the shallowness of the landscape.

We have added additional text in the discussion to acknowledge and discuss the implications of the shallowness of the effect of foot segment angle in response to question 2 from referee 1 (R1.Q2.a and b).

We have also added the following sentence in the 3.1 metabolic rate section in results to justify why we used the second-order of foot segment angle instead of the first-order term:
The first order of foot segment angle did not show a significant effect in the first iteration of the linear mixed-effects model analysis ($P = 0.318$) and was therefore removed.

We have added the following reference:

Perera S, Mody SH, Woodman RC, Studenski SA. Meaningful change and responsiveness in common physical
performance measures in older adults. Journal of the American Geriatrics Society. 2006 May;54(5):743-9.

References (already in the manuscript):

Liew BX, Morris S, Netto K. The effects of load carriage on joint work at different running velocities. Journal of
biomechanics. 2016 Oct 3;49(14):3275-80.

Antonellis P, Galle S, De Clercq D, Malcolm P. Altering gait variability with an ankle exoskeleton. PloS one. 2018 Oct
24;13(10):e0205088.

Additional references not in manuscript:

Collins SH, Wiggin MB, Sawicki GS. Reducing the energy cost of human walking using an unpowered exoskeleton.
Nature. 2015 Jun;522(7555):212.

http://biomechatronics.cit.cmu.edu/publications/Collins_2015_Nature---SourceData.zip

Ralston HJ. Energy-speed relation and optimal speed during level walking. Internationale Zeitschrift für Angewandte
Physiologie Einschliesslich Arbeitsphysiologie. 1958 Oct 1;17(4):277-83.

Zarrugh MY, Todd FN, Ralston HJ. Optimization of energy expenditure during level walking. European journal of applied
physiology and occupational physiology. 1974 Dec 1;33(4):293-306.

R1.Q3. The introduction contains a section pertaining to muscle contractile mechanics and
energetics (begins bottom of page 3) and how this might mechanistically explain changes in
energetic cost that come with manipulating ankle mechanics. However, such topics are never
brought up in the discussion. I would expect such theory to appear in both if relevant, or neither if
not. The discussion brings up the potential applications of the knowledge gained, and I would
expect some or all of these to be a part of the rationale for conducting the study. This would
provide context for a broader audience.

We thank the referee for this suggestion to improve our discussion.

From the parameters that go into a muscle-energetics model, **the ones that are most likely
affected by the shoe inclination conditions are probably the length parameters of the
plantarflexor muscles such as contractile element length.** Of course, it is possible that other
parameters (e.g., force, velocity, activation) are also affected, but it is hard to predict in which way
since shoe inclination is mostly a position manipulation.

In the equations below [Umberger et al., 2003], it can be seen that when the contractile element
length is larger than the optimal length, then the first and second term of \dot{E} (activation-
maintenance and shortening-lengthening heat rate) are multiplied by a force-length multiplier
(F_{ISO}) which is smaller than 1 for lengths greater than the optimal length. As such, **if everything
other than the contractile element lengths would stay the same, then we would expect
lower metabolic costs of the plantarflexor muscles in (upward) shoe conditions that lead
to increased contractile element length.**

if $L_{CE} \leq L_{CE(OPT)}$,

$$\dot{E} = \dot{h}_{AM} A_{AM} S$$

$$+ \begin{cases} [-\alpha_{S(ST)} \tilde{V}_{CE} (1 - \%FT/100) - \alpha_{S(FT)} \tilde{V}_{CE} (\%FT/100)] A_s S & \text{if } \tilde{V}_{CE} \leq 0 \\ \alpha_L \tilde{V}_{CE} A S & \text{if } \tilde{V}_{CE} > 0 \end{cases}$$

$$- (F_{CE} V_{CE})/m$$

if $L_{CE} > L_{CE(OPT)}$,

$$\dot{E} = (0.4 \times \dot{h}_{AM} + 0.6 \times \dot{h}_{AM} F_{ISO}) A_{AM} S$$

$$+ \begin{cases} [-\alpha_{S(ST)} \tilde{V}_{CE} (1 - \%FT/100) - \alpha_{S(FT)} \tilde{V}_{CE} (\%FT/100)] F_{ISO} A_s S & \text{if } \tilde{V}_{CE} \leq 0 \\ \alpha_L \tilde{V}_{CE} F_{ISO} A S & \text{if } \tilde{V}_{CE} > 0 \end{cases}$$

$$- (F_{CE} V_{CE})/m$$

Muscle-energetics model from Umberger et al., [2003].

However, understanding how other variables such as force, velocity, activation of the plantarflexors and other muscles change would require additional analyses using ultrasonography or musculoskeletal simulation, which could be future applications.

We have added the following (underlined) sentences in the discussion to provide possible explanations and future applications related to muscle energetics:

The fact that level shoes minimized metabolic rate during walking on a level treadmill could be related to reduced effort of the ankle joint. If we assume that shoes that result in upward foot segment angles would lead to longer plantarflexor contractile element lengths, then if all other parameters would stay constant, based on the energetics model from Umberger et al., [17] we would expect lower metabolic rate values in conditions with upward foot segment angles. In contrast to this hypothesis, we found increasing metabolic rates in conditions with greater upward foot segment angles. Knowing whether upward shoe conditions led to longer contractile elements lengths would require additional measurements (e.g., ultrasonography). It is also likely that other parameters that affect metabolic rate of muscles were induced such as contractile element force, velocity, and muscle activation. As a matter of fact, we found that soleus muscle activity showed a descending-rising trend with a minimum that aligned with a foot segment angle of 0° which minimized metabolic rate (Figure 6A).

Reference (already in the manuscript):

Umberger BR, Gerritsen KG, Martin PE. A model of human muscle energy expenditure. Computer methods in biomechanics and biomedical engineering. 2003 May 1;6(2):99-111.

Minor comments

R1.Q4. a) The chosen walking speed was quite slow. Was there a reason for this? b) How easily could the participants walk in the modified shoes and were they given some familiarisation to them?

a) The referee is right that the speed of 1 m·s⁻¹ is relatively slow compared to the speeds that are
used in studies on the biomechanics and energetics of walking.

**To be able to analyze the effects of footwear and treadmill inclination isolated from**
**confounding effects from differences in speed, we chose to use one single speed.** Since
some of the conditions with higher positive and higher negative footwear inclinations at uphill and
downhill grades were somewhat challenging, we chose a relatively slow speed that would be
feasible under all conditions.

**It appears that other studies with challenging locomotion conditions also used a speed of**
**1 m·s⁻¹ (possibly this is also because they wanted the speed to be constant under all**
**conditions).** For example, Voloshina et al., [2013] used the same speed (1 m·s⁻¹) to investigate
the biomechanics and energetics of walking on uneven terrain. Another study from Kent et al.,
[2019] examined the effects of an uneven terrain surface on whole-body locomotor dynamics at
a fixed speed of 1 m·s⁻¹ immediately following exposure and after a familiarization period.

b) **Before the measurements, participants received ten minutes habituation to walking on**
**a treadmill with the modified shoes.** This is mentioned in the following text in the 2.4 protocol
section in methods: *Participants then performed a 10-minute warm-up [23] to become habituated*
*with the shoe construction and treadmill before data collection.*

**This habituation duration is relatively short, and it is possible that slightly different results**
**might have occurred e.g. if participants were given multiple days to get used to the**
**different shoes. We have added the following sentence in the discussion to acknowledge**
**this limitation:**

*Although the participants walked comfortably with different shoes, it is possible that lower and*
*perhaps different metabolic rate landscapes could have resulted if participants were given a*
*longer habituation time (e.g., multiple days).*

However, from pilot testing the shoes ourselves, anecdotally we have the impression that
adapting to the shoes is relatively easy (e.g., compared to experiments with exoskeletons). **None**
**of our participants had any difficulties walking with the modified shoes at uphill and**
**downhill grades.** This is mentioned in the following sentence in the discussion:

*... none of the participants commented that it was difficult to walk or to maintain balance.*

The statistically significant P-values for the different condition variables and the fact that most
results of downward and upward shoe conditions are consistent with results from studies with
high-heeled and low-heeled shoes seems to **support that the habituation duration was at least**
**sufficient to obtain statistically consistent trends across participants and consistent**
**results with the literature.**

Reference (already in the manuscript):

Voloshina AS, Kuo AD, Daley MA, Ferris DP. Biomechanics and energetics of walking on uneven terrain. *Journal of*
*Experimental Biology.* 2013 Nov 1;216(21):3963-70.

Additional reference not in manuscript:

Kent JA, Sommerfeld JH, Mukherjee M, Takahashi KZ, Stergiou N. Locomotor patterns change over time during walking
on an uneven surface. *Journal of experimental biology.* 2019 Jul 15;222(14):jeb202093.

R1.Q5. How was placement of EMG electrodes standardised? Also, was the skin prepped for
electrode placements at all?

Thank you for pointing out that we forgot to mention this. **We positioned the electrodes**
**following the SENIAM guidelines and prepared the skin by shaving and cleaning of the**
**skin prior to attaching the electrodes.** We have added the following text in the 2.5
measurements section in methods:

*Electrodes were positioned according to SENIAM guidelines [29]. Skin preparation prior to*
*attaching the electrodes included shaving hair from the recording site and wiping with an alcohol*
*swab.*

We have added the following reference:

Hermens HJ, Freriks B, Disselhorst-Klug C, Rau G. Development of recommendations for SEMG sensors and sensor
placement procedures. Journal of electromyography and Kinesiology. 2000 Oct 1;10(5):361-74.

R1.Q6. The low end of the band-pass EMG filter is quite high for surface EMG (50 Hz) – why was
this chosen?

We thank the referee for this comment. **We chose the cut-off based on EMG processing**
**guidelines documentation** (https://www.c-motion.com/v3dwiki/index.php/Tutorial_EMG). **The**
**same low-pass cut-off frequency has been used in other studies in the literature** (e.g., Galle
et al., 2017; Thibordee and Prasartwuth, 2014; Cacciatore et al., 2011). We have added the
following references in our sentence in the 2.6 data processing section in methods:

*We filtered EMG signals with a 50-400 Hz band-pass filter [34-36].*

We have added the following references:

Galle S, et al. Reducing the metabolic cost of walking with an ankle exoskeleton: interaction between actuation timing
and power. Journal of neuroengineering and rehabilitation. 2017 Dec;14(1):35.

Cacciatore TW, et al. Increased dynamic regulation of postural tone through Alexander Technique training. Human
movement science. 2011 Feb 1;30(1):74-89.

Thibordee S, et al. Effectiveness of roundhouse kick in elite Taekwondo athletes. Journal of Electromyography and
Kinesiology. 2014 Jun 1;24(3):353-8.

R1.Q7. I believe the authors should describe how the foot segment was defined using anatomical
markers, as this is central to the paper.

We thank the referee for this suggestion. **We have added the following text in the 2.5**
**measurements section in methods:**

*The foot segment was defined by using markers placed at the calcaneus (lateral on the heel of*
*the shoe, at furthest point from heel marker on rigid portion of counter), heel (on heel counter at*
*the same height as the toe, centrally when viewing from a posterior position along the long axis*
*of the shoe), first metatarsal (medially on shoe at a point approximating position of first metatarsal*
*head), fifth metatarsal (laterally on shoe at a point approximating position of fifth metatarsal*
*head), and toe (at a point approximating position of second metatarsal head on dorsum of shoe).*

R1.Q8. Section 3.2 – I don't think EMG would come under a 'mechanics' heading.

We thank the referee for this comment. **We have revised this heading in section 3.2 (changes**
**are underlined):** *Ankle mechanics and muscle activation*

R1.Q9. Page 10, line 10 – the final sentence of this paragraph is a rather empty statement.

**The referee is correct about this. We have removed this sentence from the results section.**

R1.Q10. Page 10, line 29 – do you mean treadmill angle instead of foot angle here? (last line of
first paragraph in section 3.3)

**Thank you for noticing this. We have changed this sentence from ‘upward foot segment**
**angles’ to ‘uphill treadmill grades’**

R1.Q11. Page 10, line 35 – should this be ‘positive COM power’

**We were not entirely certain, but we assume that this comment is about line 19 (instead of line**
**35) where we accidentally omitted the word ‘power’.**

**We have corrected this omission. Thank you again for helping us find this.**

R1.Q12. Page 10, line 38 – can you reference a table or figure that contains the relevant data
here?

**Thank you for this comment. We have now referenced the relevant figure in this sentence.**

Referee: 2

Comments to the Author

Thank you for the opportunity to review this manuscript by Antonellis and colleagues. The study appears well planned and executed and the paper is generally well written. The study seeks to understand if wedged shoes that offset inclined and declined walking, will improve metabolic cost, in a similar manner to walking up stairs, rather than walking up a slope. I have a number of general comments and specific suggestions, outlined below;

Thank you for your appreciation and critical review. Your comments helped us understanding which aspects needed further clarification.

General Comments

R2.Q1. a) The statistical approach is not adequately described and seems somewhat like an exploratory fishing trip, rather than a carefully controlled analysis guided by specific hypotheses. b) There is a large number of comparisons and it is unclear if this has been accounted for in the statistical approach (eg. Bonerroni corrections). c) Further, with only 10 participants, is the study powered to conduct a multiple regression analysis with so many variables? d) The statistical approach needs to be clarified and potentially re-considered

Thank you for this feedback. We understand that we should have done a better job at the description and motivation of the statistical approach.

a) We agree with the referee that the second part of our results that focused on identifying which biomechanical variables explained the variance in metabolic rate was more exploratory than hypothesis-based. Therefore, **we have now removed this entire section from the results (as well as the associated parts from the aims, methods, discussion, figures, tables) (also in response to question 1 from referee 1 (R1.Q1)).**

For the first part of the results, we have tried to address the comment by **providing the (underlined) additional explanation and additional figure in the 2.7 statistical analyses section in methods regarding how each term of the statistical model tests a number of specific hypotheses:**

Dependent variable = $c_0 + c_1 \cdot \text{foot segment angle} + c_2 \cdot \text{foot segment angle}^2 + c_3 \cdot \text{treadmill grade} + c_4 \cdot \text{treadmill grade}^2 + c_5 \text{foot segment angle} \cdot \text{treadmill grade}$

Starting from this model, we could test a number of specific hypotheses regarding the landscapes of metabolic rate and other biomechanical variables in one single analysis. The P-value associated with c_0 will inform us whether a variable has an intercept that is consistently different (or not) from zero across participants. For example, if the P-value for c_0 for metabolic rate is different from zero, it indicates that the best fitting statistical model has a metabolic rate that is different from zero for walking with level shoes on a level treadmill grade. The P-value associated with c_1 will inform us whether a variable follows a non-horizontal trend versus foot segment angle, or in the case that the trend is consistently parabolic, it will inform us whether the location of the minimum is different from a foot segment angle of zero. The P-value associated with c_2 will inform us whether a variable follows a parabolic trend versus foot segment angle. The P-value associated with c_3 will inform us whether a variable follows a non-horizontal trend versus treadmill grade and whether the location of the minimum is different from a level treadmill grade. The P-value associated with c_4 will inform us whether a variable follows a parabolic trend versus treadmill grade. The P-value associated with c_5 will inform us whether the trend of a variable versus foot

segment angle is different at treadmill grades. The landscape shape of the dependent variables would be different depending on which coefficients in the statistical model are not significantly different from zero (Figure 3).

Figure 3. Role of significance testing of each coefficient in the statistical model. Subplots show the landscape shape of a variable when certain coefficients are not significantly different from zero.

b) Our sincere apologies if we misunderstood the question about multiple testing corrections but to the best of our understanding, corrections for multiple testing are used for example when conducting pairwise comparisons between conditions after a repeated measures ANOVA.

We have consulted a biostatistician and were advised that for our type of data, **it is more advantageous to use linear mixed-effects model analysis than repeated measures ANOVA because:**

- both of our independent variables are continuous variables (i.e., foot segment angle and treadmill grade can be plotted on a continuous scale).
- our dependent variable is continuous.
- participants underwent multiple conditions and we are interested in the within-subject effects.

Using linear mixed-effects model analysis we were able to obtain best fitting coefficients and P-values for each factor in one single test, therefore, **there is no multiple testing (and no correction for multiple testing).**

We have added the following text in the 2.7 statistical analyses section in methods to clarify that all the independent factors are tested in one single analysis:

Starting from this model, we could test a number of specific hypotheses regarding the landscapes of metabolic rate and other biomechanical variables in one single analysis.

c) We understand the concern of the referee that the number of participants might seem low.
Perhaps this concern is based on previous experience with studies with statistical designs that
are less sensitive (e.g., statistical designs where the independent variables are treated as
categorical variables such as repeated measures ANOVA which are often less sensitive).

Not only the number of participants but also the number of conditions and the range over which
the independent variables are varied affects the capability of a linear mixed-effects model analysis
to reject or accept null hypotheses for each independent variable. While it could be argued that
our number of participants is low compared to certain other studies, we think that the number of
tested conditions and the range over which we altered the foot segment angle and treadmill grade
are not low.

If we would have used multiple regression (which the referee appears to assume) instead of linear
mixed-effects model analysis, we would have only obtained an overall P-value. **Linear mixed-**
**effects model analysis gives P-values for each coefficient that indicate the probability of**
**finding the observed, or more extreme, coefficient value when the null hypothesis is true.**
Regarding the metabolic rate, the number of participants, and the number and range of conditions
**appear to have been adequate to reject that the coefficients of the following (5 out of 6)**
**factors are equal to zero (null hypothesis):**

- - Intercept ($P < 0.001$)
=> This indicates that the metabolic rate from walking with level shoes on a level treadmill
grade is different from zero.
- Foot segment angle² ($P = 0.013$)
=> This indicates that the metabolic rate follows a U-shaped trend versus foot segment
angle (as we hypothesized).
- Treadmill grade ($P < 0.001$)
=> This indicates that the minimum of the U-shaped trend is not at a treadmill grade of
zero degrees (as known from the literature, e.g., Minetti et al., 1993).
- Treadmill grade² ($P < 0.001$)
=> This indicates that the metabolic rate follows a U-shaped trend versus treadmill grade
(as known from the literature, e.g., Minetti et al., 1993).
- Foot segment angle · treadmill grade ($P = 0.023$)
=> Since the metabolic rate also follows a U-shaped trend versus foot segment angle, this
indicates that the minimum of metabolic rate versus foot segment angle is different
between treadmill grade conditions.

For the non-significant factor (i.e., foot segment angle), we agree with the referee that it would be
interesting if there would be a way to conduct a statistical power analysis. Linear mixed-effects
model analysis is a sensitive statistical test and based on advice from our biostatistician it is the
most appropriate test for our data which contains continuous independent and dependent factors
and within-subject conditions. However, it is unfortunate that:

- - **There appears to be no accepted general standard for statistical power analysis with**
**mixed-effects models [Castelloe and O'Brien, 2001; Guo et al., 2013].**
- Existing methods require generating simulated data (instead of the actual recorded data)
[Castelloe and O'Brien, 2001; Guo et al., 2013; Green and MacLeod, 2016].
- Certain authors even advise against using statistical power analysis when the goal is to explain
current data and not planning future studies [Hoenig and Halsey, 2001; Yuan and Maxwell, 2006].

To acknowledge our uncertainty about the statistical power of the non-significant factor, we have added the following text in the limitations section of the discussion:

It is possible that different results might have been found if we had tested a larger number of participants or higher range of shoe inclinations. While the number of participants and range of shoe inclinations have been adequate to reject the null hypothesis for 5 of the 6 coefficients of our statistical model for metabolic rate (P -values ≤ 0.023), we have no certainty whether the non-significant effect of the first order of foot segment angle on metabolic rate ($P = 0.318$) is related to our sample size ($n = 10$).

As a further clarification, we do not interpret the significant P -values as indications that certain effects are large. We completely agree with the interpretation of referee 1 that the effect of foot segment angle is shallow. **We have added the following (underlined) text to acknowledge this in the discussion (see also response to R1.Q2.a):**

The range of shoe inclinations that were tested per treadmill grade (15° from most downward to most upward) was higher than the range of treadmill grades (12°). Despite the considerable range of tested shoe inclinations, the effects of shoe inclination were relatively shallow and outweighed by the effects of treadmill grade.

d) In addition to the extra paragraph and figure that we have added in the manuscript in response to subquestion a) above, **we have also cited the following handbooks (in the 2.7 statistical analyses section in methods) that introduce and explain linear mixed-effects model analysis:**

To determine the effects and interaction effects of foot segment angle and treadmill grade on metabolic rate and biomechanical variables, we used linear mixed-effects model analyses with participant number as random effect [38,39].

We have added the following references:

Vispoel WP. Book Reviews: Statistical Reasoning for the Behavioral Sciences (2nd ed.) Richard J. Shavelson Needham Heights, MA: Allyn and Bacon, 1988. viii + 744 pp. J. Educ. Stat. 15, 179–183; 1990.

Gałecki A, Burzykowski T. Linear mixed-effects models using R: A step-by-step approach. Springer Science & Business Media; 2013 Feb 5.

Reference (already in the manuscript):

Minetti AE, Ardigo LP, Saibene F. Mechanical determinants of gradient walking energetics in man. The Journal of physiology. 1993 Dec 1;472(1):725-35.

Additional references not in manuscript:

Castelloe JM, O'Brien RG. Power and sample size determination for linear models. In Proceedings of the Twenty-Sixth Annual SAS Users Group International Conference 2001 Apr (pp. 240-26).

Guo Y, Logan HL, Glueck DH, Muller KE. Selecting a sample size for studies with repeated measures. BMC medical research methodology. 2013 Dec;13(1):100.

Green P, MacLeod CJ. SIMR: an R package for power analysis of generalized linear mixed models by simulation. Methods in Ecology and Evolution. 2016 Apr;7(4):493-8.

Hoenig JM, Heisey DM. The abuse of power: the pervasive fallacy of power calculations for data analysis. The American Statistician. 2001 Feb 1;55(1):19-24.

Yuan KH, Maxwell S. On the post hoc power in testing mean differences. Journal of Educational and Behavioral Statistics. 2005 Jun;30(2):141-67.

R2.Q2. a) Centre of mass mechanics - Is there a reason why average COM power is used, rather than COM work?

b) I wonder if the wedged shoes may influence the magnitude of negative, positive and net-work during the walking tasks. This has the potential to influence metabolic cost and does not seem to be accounted for here.

a) Thank you for this comment and our apologies for the confusion.

We chose to report all our results as work divided by stride time (Joules/s or Watt) instead of work (Joules) because:

- Our main outcome metric is metabolic rate (in Watts). The metabolic cost could be expressed in Joules if we provide the reader with information about the duration over which this amount of Joules is measured. However, a more direct (and more common) way to present the effort intensity is by reporting metabolic rate in Watts. To interpret where the changes in metabolic rate could result from it makes more sense to report joint and COM energetics in the same units as metabolic rate (by dividing work by stride time).
- A certain shoe could reduce work per stride at a joint or COM work by a small amount but result in a relatively larger increase in stride frequency. In this hypothetical scenario, analyzing the joint or COM energetics based on work per stride could give a confusing impression because the total work during a certain condition could be increased because of a higher number of strides. It appears that joint and COM energetics are reported in Joules for non-cyclic behaviors (e.g., a single vertical jump) whereas it is common to report energetics in Watts for cyclic behaviors (e.g., walking with exoskeletons; Collins et al., 2015; Jackson and Collins, 2015).

We calculated all our joint and COM energetics by integrating the positive or negative portions of the power time series of one stride over time and dividing by stride time (Figure R3). We originally reported these summary metrics as 'average positive powers' or 'average negative powers'. Based on the comment of the referee we understand that this could potentially confuse because the term 'power' is usually associated with an attribute of a time series (e.g., peak power). To avoid this confusion, we have now chosen to replace average power by work rate throughout the manuscript (also used in Jackson and Collins, 2015).

Additional figure (not in manuscript) Figure R3. Example of calculation of summary metric for COM power.

We have revised the following text in the 2.6 data processing section in methods to clarify this:

We reported joint and COM mechanics by integrating joint and COM powers over stride time and dividing by stride time. We chose to report joint and COM mechanics as work rates (W) instead of work (J) to present these mechanics in comparable units as metabolic rate.

b) The referee is correct. **The different shoe configurations (or treadmill grades) may affect COM and joint powers (and consequently affect COM and joint work rates). For example, figure 5, figure S1 and figure S2 show how:**

- More upward foot segment angles resulted to increasing positive COM work rate during downhill walking and decreasing COM work rate during uphill walking (figure 5, A and C).
- More upward foot segment angles resulted to increasing positive ankle work rate (figure S1, E and figure S2, E).

Figure 5, A and C. Center of mass mechanics.

Figure S1, E (top). Positive (pos.) ankle work rate.

Figure S2, E (bottom). Ankle joint power.

In the first part of the discussion, we tried to give an analysis of how these mechanical changes influence the metabolic rate.

In the second part of the discussion section, we provide an overview of the effects of shoe inclination on joint work rates and COM work rates. We have greatly reduced this part in response to R1.Q1.

Additional references not in manuscript:

Collins SH, Wiggin MB, Sawicki GS. Reducing the energy cost of human walking using an unpowered exoskeleton. *Nature*. 2015 Jun;522(7555):212.

Jackson RW, Collins SH. An experimental comparison of the relative benefits of work and torque assistance in ankle exoskeletons. *Journal of Applied Physiology*. 2015 Jul 9;119(5):541-57.

Specific comments

R2.Q3. Introduction (page 3 line 51-2) - should this be “efficiency” is maximized at optimal fascicle length, rather than “metabolic cost”?

Thank you for verifying this. **Isometric strength is of course maximal when the contractile element is at optimal length, but the metabolic energy consumption is not necessarily the most efficient according to the muscle-energetics model from Umberger et al., [2003].**

In the equations below [Umberger et al., 2003], it can be seen that when the contractile element length is larger than the optimal length, then the first and second term of \dot{E} (activation-maintenance and shortening-lengthening heat rate) are multiplied by a force-length multiplier (F_{ISO}). This force-length multiplier is smaller than 1 for lengths greater than the optimal length.

if $L_{CE} \leq L_{CE(OPT)}$,

$$\dot{E} = \dot{h}_{AM} A_{AM} S$$

$$+ \begin{cases} [-\alpha_{S(ST)} \tilde{V}_{CE} (1 - \%FT/100) - \alpha_{S(FT)} \tilde{V}_{CE} (\%FT/100)] A_s S & \text{if } \tilde{V}_{CE} \leq 0 \\ \alpha_L \tilde{V}_{CE} A S & \text{if } \tilde{V}_{CE} > 0 \end{cases}$$

$$- (F_{CE} V_{CE})/m$$

if $L_{CE} > L_{CE(OPT)}$,

$$\dot{E} = (0.4 \times \dot{h}_{AM} + 0.6 \times \dot{h}_{AM} F_{ISO}) A_{AM} S$$

$$+ \begin{cases} [-\alpha_{S(ST)} \tilde{V}_{CE} (1 - \%FT/100) - \alpha_{S(FT)} \tilde{V}_{CE} (\%FT/100)] F_{ISO} A_s S & \text{if } \tilde{V}_{CE} \leq 0 \\ \alpha_L \tilde{V}_{CE} F_{ISO} A S & \text{if } \tilde{V}_{CE} > 0 \end{cases}$$

$$- (F_{CE} V_{CE})/m$$

Muscle-energetics model from Umberger et al., [2003].

We originally wrote the following statement in three parts based on how this was written in Umberger et al., [2003]:

A model of muscle energy expenditure [17] based on the Hill-type muscle model [18] suggests that (1) metabolic cost is maximal at the optimal fascicle length on the force-length relationship, (2) metabolic cost decreases with greater fascicle lengths, and (3) metabolic cost does not change with shorter fascicle lengths than the optimal length on the force-length relationship.

To improve clarity, we have shortened this statement (changes are underlined) and have added references to the original studies that describe the length dependency of muscle-metabolic cost:

A model of muscle energy expenditure [17] based on the Hill-type muscle model [18] suggests that (1) metabolic cost is near maximal when contractile element lengths are at or below the optimal length on the force-length curve, and (2) metabolic cost decreases with longer contractile element lengths [19,20].

We have added the following references:

Wolledge RC, Curtin NA, Homsher E. Energetic aspects of muscle contraction. Monographs of the physiological society. 1985;41:1-357.

Hilber K, Sun YB, Irving M. Effects of sarcomere length and temperature on the rate of ATP utilisation by rabbit psoas muscle fibres. The Journal of physiology. 2001 Mar;531(3):771-80.

Reference (already in the manuscript):

Umberger BR, Gerritsen KG, Martin PE. A model of human muscle energy expenditure. Computer methods in biomechanics and biomedical engineering. 2003 May 1;6(2):99-111.

R2.Q4. Introduction (p4 line 2-3) Please add a reference after “during human walking”

Thank you for pointing this out. The second part of this sentence which mentions that contractile element lengths and stiffnesses are important for minimizing metabolic cost during walking is indeed missing appropriate references.

We have added the following references after ‘during human walking’:

Uchida TK, Hicks JL, Dembia CL, Delp SL. Stretching your energetic budget: how tendon compliance affects the metabolic cost of running. PloS one. 2016 Mar 1;11(3):e0150378.

Markowitz J, Herr H. Human leg model predicts muscle forces, states, and energetics during walking. PLoS computational biology. 2016 May 13;12(5):e1004912.

R2.Q5. Methods - sample size. Is this study adequately powered for the large number of comparisons and the multiple regression analysis?

We understand the concern of the referee that the number of participants might seem relatively low in comparison to other studies. **We believe that the following results support that our study was adequately powered (i.e., R2.Q1.c).**

For metabolic rate, the number of participants, and the number and range of conditions appear to have been adequate to reject that the coefficients of the following (5 out of 6) factors are equal to zero (null hypothesis):

- Intercept ($P < 0.001$)
- Foot segment angle² ($P = 0.013$)
- Treadmill grade ($P < 0.001$)
- Treadmill grade² ($P < 0.001$)
- Foot segment angle · treadmill grade ($P = 0.023$)

This is described in more detail in the response to R2.Q1.c.

R2.Q6. a) Methods (2.2 Modular footwear) - the footwear has the potential to change the effective leg length. For example shoes with increased heel wedging (for uphill walking) will increase the leg length, while shoes with the negative heel (higher at ball of foot) will not have the same effect. This has the potential to influence joint and COM mechanics.

b) Has this been accounted for?

a) Thank you for verifying this. The different shoe configurations do indeed have the potential to affect the effective leg length during different parts of the stance phase and the COM kinematics.

The effects of COM displacement can be seen in Figure S5, A. From this figure, it can be seen that **the rise of the COM happens a little bit earlier in the upward shoe conditions and the**

drop of the COM happens a little bit earlier in the upward shoe conditions (and vice versa for both of the downward shoe conditions).

Figure S5, A. Effects of shoe conditions on COM displacement.

b) To the best of our knowledge, **the changes in effective leg length have been taken into account in the appropriate way in the calculations of joint mechanics and COM mechanics as described in the 2.6 data processing section in methods:**

We calculated whole-body COM acceleration based on the body mass and the total ground reaction force of both legs. We calculated COM velocity by integrating the COM acceleration over time [32].

Estimating COM velocity using ground reaction force measurements takes all sources of displacement of mass into account (e.g., displacement due to soft tissue movement, or displacements due to 'effective length' changes of segments) [Gard et al., 2004]. We only measured the COM displacement, we did not measure COM position or effective leg length but this is not needed since COM power is defined as the product of COM velocity and the individual leg ground reaction force [Donelan et al., 2002].

As described in the 2.6 data processing section in methods:

We calculated the foot segment angle, joint angles, joint moments, and powers of the right leg using 3D kinematical analyses and inverse dynamical analyses (Visual3D, C-Motion, Germantown, MD, USA).

We have also added the following sentence immediately after the above sentence in the 2.6 data processing section in methods:

We estimated body segment mass distribution based on Dempster, [31], whereby foot segment mass was adjusted to account for shoe mass.

Based on methods from other studies that are similar to our study which involved walking on wedged surfaces (Figure R4) or walking with high-heeled or low-heeled shoes, it appears that no additional adjustments are needed for studying moments and powers of the joints.

Additional figure (not in manuscript) from Willwacher et al., [2013]. Figure R4. Schematic description showing how the position of the ground reaction force relative to a force plate under a wedged surface can be used to study the inverse dynamics of running on a wedged surface (i.e., on top of the force plate without requiring any adjustment for the height of the wedged surface).

In summary, similar to the COM displacement and power, we see that the shoe conditions have effects on joint moments and powers (e.g., Figure 5 and responses to R2.Q2). **We conclude that these analyses are not missing any effects of leg length we should have accounted for.**

We have added the following reference:

Dempster WT. Space requirements of the seated operator, geometrical, kinematic, and mechanical aspects of the body with special reference to the limbs. Michigan State Univ East Lansing; 1955 Jul.

Reference (already in the manuscript):

Donelan JM, Kram R, Kuo AD. Simultaneous positive and negative external mechanical work in human walking. *Journal of biomechanics*. 2002 Jan 1;35(1):117-24.

Additional references not in manuscript:

Gard SA, Miff SC, Kuo AD. Comparison of kinematic and kinetic methods for computing the vertical motion of the body center of mass during walking. *Human movement science*. 2004 Apr 1;22(6):597-610.

Willwacher S, Fischer KM, Benker R, Dill S, Brüggemann GP. Kinetics of cross-slope running. *Journal of biomechanics*. 2013 Nov 15;46(16):2769-77.

R2.Q7. Methods, Data processing (p7 line 14) Please change “kinematical” and “dynamical” to
kinematic and dynamic

We thank the referee for this comment. **We have revised this sentence (changes are**
**underlined) in the 2.6 data processing section in methods as follows:** *We calculated the foot*
*segment angle, joint angles, joint moments, and powers of the right leg using 3D kinematic*
*analyses and inverse dynamic analyses (Visual3D, C-Motion, Germantown, MD, USA).*

R2.Q8. Methods, statistical analysis (p7 line 46) - see earlier comments about clarity of description
and a large number of multiple comparisons

Thank you for this reminder. **We have added the following elements in the manuscript to**
**improve the clarity of the statistical analysis:**

- - Additional text in the methods to explain how each term of the statistical model tests a specific
- hypotheses.
- - An additional figure that shows the hypothetical effect if each coefficient was not significantly
- different from zero.
- - Additional references that introduce and explain the statistical approach that we used (linear
- mixed-effects model analysis).

Regarding the number of comparisons, we have added extra text in the manuscript to specify that
all the fixed effects are evaluated in a single analysis and there is no multiple testing.

**All the above are described in more detail in the responses to R2.Q1.a,b,c,d.**

R2.Q9. Results - comment about order of data presentation. Perhaps start with metabolic data,
the COM data, then present joint based data. The regression analysis can be left at the end

Thank you for your comment. **We have changed the order of data presentation in the results**
**section based on your suggestion.** This is clearly an improvement as it is also more consistent
with the order of the introduction. We have also removed the 3.4 explained metabolic rate variance
based on different biomechanical variables section in the results regarding the regression analysis
based on R1.Q1.

Appendix B

Answers to revision

“Modular footwear that partially offsets downhill or uphill grades minimizes the metabolic cost of human walking” (Manuscript ID RSOS-191527)

We would like to thank the reviewers and the editor for their valuable comments. It is clear that the reviewers and the editor have put a lot of time and effort into this. We apologize that the study rationale and the statistics were difficult to follow and we appreciate the suggestions from the reviewers. After consulting with the co-authors, colleagues and another biostatistician, we are confident that we were able to provide answers to all the comments and questions.

Summary of main changes:

- We **simplified the reporting of the statistics**. We looked at examples from other studies in the literature to try to understand how to explain the statistics more clearly. Based on these examples, we decided to move Table 1 to supplementary material but emphasized that the statistical results can be most easily summarized from the figures (i.e., the lines that are drawn in the figures are based on the results that were found to be statistically significant).

- We conducted **six additional analyses** based on suggestions from reviewers:

Repeated measures ANOVA, analyses of individual results, residual analysis of different fitting equations, metabolic cost **drift analysis** and characterization of the **hardness and bending stiffness** of the modular footwear and we provided responses according to the results of these analyses.

- We completely **re-wrote the first part of the introduction and aims** based on a number of comments regarding justification and relevance.

In addition to these main changes, we have also provided point-by-point answers to all other major and minor comments. Original comments are presented in black font and answers are formulated in blue color. Manuscript text is indicated in *italic*, and changes are underlined. Actions that we took to implement the comments from the reviewers in the manuscript are marked with an arrow (=>). In addition to this response document, we also provide a revised manuscript document with tracked changes and a version without tracked changes.

Reviewer: 1

Comments to the Author(s)

The comments are included (in red and 2 remarks) in the text part of the PDF file RSOS-191527_Proof_hi_review.pdf

We graciously thank the reviewer for rereviewing the manuscript and for helping us improve the manuscript. We have added all the comments from reviewer 1 (in red color) from the PDF file below along with our responses in blue.

R1.Q1. *Although we know that uneven terrain or grades influence the mechanics and energetics of human walking, we have an incomplete understanding of the effects of whole-body and distal mechanics do you mean relation between biomechanical adaptations and metabolic rate ?*

=> Thank you, we have entirely **revised this sentence** in the abstract to address the question about relevance:

Walking on different grades becomes challenging on energetic and muscular levels compared to level walking.

R1.Q2. *Is it possible to minimize metabolic rate with shoe outsoles that offset downhill or uphill grades? ("is it possible" = descriptive statement = seems not hypothesis driven; and what is the relevance?)*

=> Thank you for this suggestion. We have **added** sentences to support the **relevance** and provide a **hypothesis**:

While it is not possible to eliminate the cost of raising or lowering the center of mass, it could be possible to minimize the cost of distal joints with shoes that offset downhill or uphill grades.

R1.Q3. *We investigated the interaction effects of outsole geometry and treadmill grade on metabolic rate and biomechanics of walking. Add "Therefore",.*

=> Thank you for this suggestion. We ended up **entirely revising this sentence** in the abstract:

We investigated the effects of shoe outsole geometry in ten participants walking at 1 m·s⁻¹ on downhill, level and uphill grades.

R1.Q4. *The resulting trends indicate that level shoes minimize metabolic rate during level walking.*

The resulting trends (**why "trends"; aren't there statistical significant differences or correlations?**) indicate that level shoes minimize metabolic rate during level walking.

Thank you for this excellent question. Our intended meaning was that the coefficients that were found to be statistically significant in the linear mixed-effects model analysis show a minimum for metabolic rate during walking with level shoes.

=> We have revised this sentence in the abstract and have also **listed the P-value to make it clear that this was statistically significant**:

Level shoes minimized metabolic rate during level walking ($P_{\text{second-order effect}} < 0.001$).

R1.Q5. *Contrary to our hypothesis, shoes that entirely offset the (overall) treadmill grade (similar to stairs) did not minimize the metabolic rate of walking on grades.*

Contrary to our hypothesis (**important !**), shoes that **entirely** offset the (overall) treadmill grade (similar to stairs) did not minimize the metabolic rate of walking on grades.

=> Thank you for this suggestion. We have **made the changes suggested by the reviewer** (changes are underlined) in the abstract:

However, shoes that **entirely** offset the (**overall**) treadmill grade did not minimize metabolic rate of walking on grades

R1.Q6. During level walking, altering shoe inclination could influence tendon elongation and change the contractile element length (**in lower leg muscles ?**) away from the region that minimizes metabolic rate. Conversely, altering (**formulate more directly that you want to offset the grade**) the shoe inclination during uphill or downhill walking could possibly bring the plantarflexor muscle fascicle lengths back to the optimal region that minimize metabolic rate.

=> Thank you. We have **implemented the suggestions from the reviewer** (changes are underlined) in the introduction:

*During level walking, altering shoe inclination could influence tendon elongation and change the contractile element length (**in lower leg muscles**) away from the region that minimizes metabolic rate. Conversely, altering the shoe inclination **to offset uphill or downhill walking grades** could bring the plantarflexor muscle fascicle lengths back to the optimal region that minimizes metabolic rate.*

R1.Q7. Studying how varying outsole geometry influences metabolic rate on different grades (**formulate more directly that you want to offset the grade**) could help to identify relationships between biomechanical changes and the resulting changes in metabolic rate.

=> Thank you for this helpful suggestion. We have revised the sentence and **implemented the suggestion of the reviewer** (changes are underlined) in the introduction:

*Using modular footwear as a new tool for altering treadmill grade while keeping the foot segment angle constant (**by offsetting the treadmill grade**) could provide new insights into the relationships between biomechanical changes and the resulting changes in metabolic rate.*

R1.Q8. Hence, the aim of this study was to investigate the effects of varying footwear outsole geometry on human mechanics and energetics under different walking grades.

Hence, the aim of this study was to investigate the effects of varying footwear outsole geometry on human mechanics and energetics under different walking grades (**the former 2 sentences say about the same**).

=> Thank you for pointing out that the two sentences were redundant. We have revised this sentence (changes are underlined) in the introduction **to differentiate it from the beginning of the introduction**:

Hence, the aim of this study was to investigate the interaction effects of varying footwear outsole geometry and treadmill grade on metabolic rate, joint mechanics, muscle activity, ground reaction forces and COM mechanics in a controlled experiment (all footwear and treadmill parameters were kept constant except the outsole geometry and treadmill grade).

R1.Q9. We segmented each time series based on heel strike detection (**how reliable was this crucial event detected ? from de 2000Hz VGRF ? or from kinematics ? or ?**) and calculated the representative profile per stride by taking the median of all strides.

=> We apologize for forgetting to specify this. We have **added the (underlined) information requested by the reviewer** in the 2.6 data processing section in methods:

*We segmented each time series based on heel strike detection (**using the GRF**) and calculated the representative profile per stride by taking the median of all strides.*

R1.Q10. A +3° foot segment angle (obtained with a -3° downward shoe) minimized metabolic rate during walking on a +6° uphill grade. **thus a partial offset of the grade by the shoe inclination minimizes metabolic rate: now one expects why such a partial offset "works" best**

=> We have **implemented the suggestion from the reviewer** to add the (underlined) text in the discussion:

A +3° foot segment angle (obtained with a -3° downward shoe) minimized metabolic rate during walking on a +6° uphill grade. Thus, a shoe inclination that partially offsets uphill or downhill grades can minimize metabolic rate during walking.

R1.Q11. Here, you start with confirmation of existing results = not innovative; serves only validity of the method; consider to write shorter mentioning purpose validity)

=> Thank you for this comment. We have revised **this paragraph in the discussion and mentioned that the purpose is validity as requested by the reviewer:**

The minima for the positive COM work rate and propulsive GRF were very close to the treadmill grade (-6° downhill) that minimized metabolic rate (Figs. 5B and S4, B). Both the mechanical work rate and force variables could indeed be responsible for the changes in metabolic rate because energy consumption at the muscle fiber level is dependent on the fiber work force [17]. Concentric muscle work is four times more expensive than eccentric muscle work [5] which could explain why the positive COM work rate and propulsive GRF are closely related to the metabolically optimal treadmill grade. The metabolically optimal treadmill grade was similar to the optimal grade from other studies [5,6] which supports the validity of our experiment.

R1.Q12. The decreases in negative COM work rate and braking GRF with smaller thandownhill and greater than uphill walking grades are consistent with previous studies [10,42–46].

=> We thank the reviewer for this suggestion. We have **implemented the suggestion:**

The **reductions** in negative COM work rate and braking GRF due to increases in treadmill grade **from downhill (negative) to level or from level to uphill (positive)** for all shoe inclinations that were tested on all three treadmill grades are consistent with those from other studies [13,53–57].

R1.Q13. **what about the effect of a possible change in knee angle; would affect gastroc) more important: is the effect on the metabolic rate mostly due to distal = ankle effect or due to total body COM effect ?)**

We thank the reviewer for this idea. **We looked at** the plantarflexion moment and **knee moment** to see if we could find explanations for the behavior of the gastrocnemius medialis (since it is a biarticular muscle that produces plantarflexion and knee flexion).

=> We have added the following sentence in the discussion:

Additionally, more downward foot segment angles led to a decrease in the plantarflexion moment (Figure S1, A) but also to a **decrease in knee flexion moment** (Figure S3, C). As such, **this does not explain the lack of reduction in the activation of the gastrocnemius medialis because this muscle is a biarticular muscle that performs plantarflexion and knee flexion.**

R1.Q14. These interaction effects between foot segment angle and treadmill grade could possibly explain that a more downward foot segment angle minimized metabolic rate during downhill walking, and a more upward foot segment angle minimized metabolic rate during uphill walking. **last sentence is a repetition of results, not a kinesiological explanation, a functional biomechanical explanation would be appropriate**

Thank you for this comment. We agree with the reviewer that a biomechanical explanation is helpful.

=> We have **added the following explanation** in the discussion:

We found interaction effects on **the biomechanical parameters that followed a similar direction to that of the interaction effect on metabolic rate. More downward foot segment angles reduced the positive COM work rate during downhill walking whereas more upward foot segment angles reduced the positive COM work rate during uphill walking (Figure 5A). Analyses of the time series that were used for the calculation of COM power indicated that this interaction effect is a result of changes in the perpendicular GRF and perpendicular COM velocity during the rebound phase. Slightly downward foot segment angles could help avoid excessive rebounding of the COM during downhill walking and vice versa for uphill walking. The second interaction effect involves a positive knee work rate with more downward foot segment angles, leading to a reduced positive knee work rate during downhill walking whereas more upward foot segment angles reduced the positive knee work rate during uphill walking (Figure 7). The **minimum positive knee work rate during downhill walking appears to be related to a reduction in the extension velocity (possibly related to the COM power effect) and the minimum positive knee work rate during uphill walking seems to be due to a reduction in the extension moment. Taken together, these interaction effects between****

foot segment angle and treadmill grade on the positive COM work rate and positive knee work rate could explain why a more downward foot segment angle minimized metabolic rate during downhill walking, and why a more upward foot segment angle minimized metabolic rate during uphill walking.

R1.Q15. By slightly extrapolating the trend in shoe conditions in the level treadmill condition, we can estimate that a shoe with a 9° inclination could cause a **similar to** percentage change in metabolic rate.

=> Thank you for pointing out this omission. We **ended up removing this sentence** from the discussion in response to question 11e) from reviewer 4 (R4.Q11.e)).

R1.Q16. Since humans typically require more time and have a higher metabolic rate during uphill portions of terrain with uphill and downhill grades, it could be advantageous to use shoes with a slight downward shoe inclination that are closer to optimal at these uphill portions. Perhaps this explains why most existing shoes have a small heel to toe drop (a typical offset of 1 cm over a shoe length of 28 cm corresponds to a downward shoe inclination of -2°). (nice suggestion, but aren't there different explanations ? fit and avoiding slip and friction ? orientation of push off vector and timing of forward COP displacement during fore foot roll off, certainly with stiffer sole constructions, etc ...

Thank you for appreciating our original explanation and **we agree with the reviewer that there might be other explanations.**

=> We have revised this text (changes are underlined) in the discussion:

Since humans typically require more time and have a higher metabolic rate during the uphill portions of terrain with uphill and downhill grades, it could be advantageous to use shoes with a slight downward shoe inclination that is closer to optimal during these uphill portions. This could partly explain why most existing shoes have a small heel to toe drop (a typical offset of 1 cm over a shoe length of 28 cm corresponds to a downward shoe inclination of -2°). However, most existing shoes are probably not designed for only uphill walking and there are other considerations in the design of shoes than minimizing metabolic rate (e.g., injury prevention, comfort, and traction).

R1.Q17. It could also be possible to design shoes that allow changing the inclination depending on the grade of the terrain (e.g., similar to heel lift on snow shoes or alpine touring skis (nice = mostly to avoid repetitive overstretching of calf muscles).

=> Thank you for your positive comment about this sentence. We **have implemented the suggestion of the reviewer** (changes are underlined) in the discussion:

It could also be possible to design shoes that allow changing the inclination depending on the grade of the terrain to avoid repetitive overstretching of the calf muscles, similar to the heel lift on snow shoes or alpine touring skis.

Reviewer: 2

Comments to the Author(s)

Thank you to the authors for such a comprehensive response to my questions and suggestions.

We would like to thank the reviewer for critically reviewing our manuscript and for appreciating our responses.

Reviewer: 3

Comments to the Author(s)

MAJOR COMMENTS

R3.Q1. I thank the authors for the opportunity to review this work. The experimental question is particularly interesting, and the research team should be highly commended on a thoughtful experimental approach. The hypothesis motivating this manuscript is that metabolic rate of walking over various slopes can be minimized if the slope is offset by footwear with complementing outsole angles. However, after hours of pouring over the statistical analyses and connecting these to the results, I still do not feel I have the strongest grasp on the work. I was overwhelmed by this and have no doubt others in the field will also get caught up here. Furthermore, if accepted for publication, this work will be Open Access. The authors reference its clinical application so it may also be important to make findings palatable to a broader audience.

We sincerely thank the reviewer for the appreciation of the interesting question and for the advice to simplify the explanation of the statistics.

We have been **advised by three independent biostatisticians from different universities** (Harvard University, the University of Nebraska at Omaha and the University of Nebraska Medical Center) **that linear-mixed effects model analysis is the most appropriate** analysis for a within-subject design wherein the independent and dependent variables are continuous variables; however, we have also explored other analyses as suggested by the reviewers).

We understand the concern of the reviewer that we should do explain the results more clearly. To determine how to communicate the results more clearly, **we looked for examples of other studies that used a similar statistical design:**

- Liew BX, Morris S, Netto K. The effects of load carriage on joint work at different running velocities. *Journal of biomechanics*. 2016 Oct 3;49(14):3275-80.

<https://www.sciencedirect.com/science/article/abs/pii/S0021929016308880>

- Castillo ER, Lieberman GM, McCarty LS, Lieberman DE. Effects of pole compliance and step frequency on the biomechanics and economy of pole carrying during human walking. *Journal of Applied Physiology*. 2014 Jul 3;117(5):507-17. <https://www.physiology.org/doi/pdf/10.1152/jappphysiol.00119.2014>

- Collins SH, Wiggin MB, Sawicki GS. Reducing the energy cost of human walking using an unpowered exoskeleton. *Nature*. 2015 Jun;522(7555):212.

<https://www.ncbi.nlm.nih.gov/pmc/articles/PMC4481882/pdf/nihms662317.pdf>

- Hedrick EA, Stanhope SJ, Takahashi KZ. The foot and ankle structures reveal emergent properties analogous to passive springs during human walking. *PloS one*. 2019 Jun 7;14(6):e0218047.

<https://journals.plos.org/plosone/article/file?id=10.1371/journal.pone.0218047&type=printable>

When looking at these examples our main impression was that studies with linear-mixed effects model analyses appear easier to understand when the results are presented as trends in a plot rather than in a table. For example, in the table below (Table R1) from Liew et al., [2016], it is not immediately apparent what the meaning of the positive and negative coefficients is or what the meaning of the sign of the interaction coefficient is, whereas in the plot below (Figure R1), the results from Collins et al., [2015], are presented as a dashed line. This dashed line visually shows that they found a significant effect for the square of the exoskeleton stiffness because it is a U-shaped curve. The plot also shows how the coefficients that they found result in an optimum close to the orange condition.

Additional table (not in manuscript). **Table R1: Results from the linear-mixed effects model analysis from Liew et al., [2016].** We think that this presentation gives more quantitative data, but it could be hard to follow for the readers.

Table 1

Mixed effect model of dimensionless joint work parameters (unit), and load (%BW) and velocity (m/s).

Variables	Intercept	β_1 (%Load)	β_2 (Velocity)	β_3 (Load:Velocity)
Ankle positive work (W_{ankle}^+)	7.63×10^{-2}	6.95×10^{-4}	1.73×10^{-2}	NA
Ankle negative work (W_{ankle}^-)	-3.19×10^{-2}	-4.69×10^{-4}	-7.48×10^{-3}	NA
Knee positive work (W_{knee}^+)	4.60×10^{-2}	1.12×10^{-3}	6.87×10^{-3}	NA
Knee negative work (W_{knee}^-)	-6.30×10^{-2}	-2.47×10^{-4}	-1.03×10^{-2}	NA
Hip positive work (W_{hip}^+)	1.72×10^{-1}	-1.22×10^{-3}	1.40×10^{-3}(ns)	2.29×10^{-4}
Hip negative work (W_{hip}^-)	-1.20×10^{-2}	-7.79×10^{-4}	-1.12×10^{-2}	NA
% ankle positive work	57.11	1.87×10^{-1} (ns)	8.10×10^{-1}	-6.7×10^{-2}
% ankle negative work	28.33	6.94×10^{-3} (ns)	5.05×10^{-2} (ns)	NA
% knee positive work	33.20	2.43×10^{-1}	-7.07×10^{-1}	NA
% knee negative work	53.17	-2.06×10^{-1}	-1.57	NA
% hip positive work	10.36	-5.39×10^{-1}	-2.87×10^{-1}(ns)	9.45×10^{-2}
% hip negative work	18.49	1.93×10^{-1}	1.52	NA
Total positive work (W_{total}^+)	1.48×10^{-1}	-2.69×10^{-4}(ns)	2.37×10^{-2}	4.42×10^{-4}
Total negative work (W_{total}^-)	-1.05×10^{-1}	-1.51×10^{-3}	-2.98×10^{-2}	NA
Total joint work (W_{stance}^{total})	2.67×10^{-1}	2.28×10^{-4}(ns)	4.97×10^{-2}	7.07×10^{-4}
Net joint work (W_{stance}^{net})	2.47×10^{-2}(ns)	-6.91×10^{-5}(ns)	-8.53×10^{-4}(ns)	NA
Stance duration	3.63×10^{-1}	1.98×10^{-3}	-3.69×10^{-2}	-2.62×10^{-4}
Cycle duration	8.80×10^{-1}	-1.51×10^{-3}	-4.77×10^{-2}	NA
Step length	4.76×10^{-1}	4.25×10^{-3}	2.26×10^{-1}	-1.98×10^{-3}
Stride length	8.82×10^{-1}	6.53×10^{-3}	4.67×10^{-1}	-3.21×10^{-3}

β =beta coefficient; ns=non-significant; NA=not applicable to final model due to non-significant interaction. Example: Total positive work at 3.0 m/s carrying 20% BW equates to: $0.148 - 0.000269(20) + 0.0237(3) + 0.000442(20)(3) = 0.240$ units.

Additional figure (not in manuscript). **Figure R1: Results from the linear-mixed effects model analysis of metabolic rate from Collins et al., [2015].** The results of the linear mixed-effects model analysis are represented with a dashed line. We found this presentation easier to follow (compared to Table R1).

=> **Based** on this analysis and **on how other papers present results from linear-mixed effects model analyses, we have made the following changes to try to simplify the results:**

- **We moved Table 1** which lists the values of the coefficients of the linear-mixed effects model analysis **to the supplementary material** (now Table S1) **since this information is redundant with the figures** and since it is easier to understand the meaning of the coefficients directly from the figures.
- We tried to **emphasize that the significant effects of the linear-mixed effects model analysis can simply be understood from the lines on the figures:** For example, the lines in Figure 4A are not simply plotting curve fits through the data of every treadmill condition. These lines are obtained by evaluating the equation resulting from the linear-mixed effects model analysis at the tested treadmill grades and range of foot segment angles. The lines are slightly U-shaped because the effect of the square of the foot segment angle was significantly different from zero ($P = 0.013$). The location of the minima for each treadmill grade is different because there was a significant interaction effect.
- We have **added text labels** in Figure 4A to clearly **indicate the minima and pictograms** to clearly indicate **which** combination of outsole, treadmill and foot segment **angles were optimal**.

=> We have **added the following (underlined) text** in the 2.7 statistical analyses section in methods to emphasize this:

To facilitate the interpretation of the meaning of the significant coefficients (positive or negative interaction coefficients), we evaluated the formula resulting from the linear-mixed effects model analysis over the tested range of foot segment angles at each treadmill grade and plotted the results as lines in the scatter plots. As such, the results of the statistical analysis can be understood from the figures; for example, if the coefficient for a second-order term was not significant, then the results were plotted as straight lines, or when there was a significant interaction effect, this effect could be understood by analyzing the differences in the slope or the location of the minima.

=> We have **added** the following (underlined) **text in the caption** of Figure 4:

Black lines represent the formula from the linear mixed-effects model analysis evaluated over the tested range of foot segment angles at each treadmill grade. The lines are slightly U-shaped because the coefficient for the second order of foot segment angle was significantly different from zero and positive. The optimum foot segment angle at each treadmill grade was different because there was a significant interaction effect between the foot segment angle and treadmill grade. Gray crosses indicate the individual interpolated minima \pm s.e.m.

=> We have **added** the following (underlined) **text** in the 3.1 metabolic rate section in **results**:

We found significant effects of c_2 (square of foot segment angle, $P = 0.013$; Table S1; Figure 4A) on metabolic rate indicating that the effect of the foot segment angle on metabolic rate at each treadmill angle followed a parabolic trend. There was a significant effect of c_3 (treadmill grade, $P < 0.001$; Table S1; Figure 4B) and of c_4 (the square of treadmill grade, $P < 0.001$; Table S1; Figure 4B). This shows that the effect of the treadmill grade on metabolic rate followed a U-shaped trend that was not centered at a 0° treadmill grade. ... The combinations of outsole angle and the resulting foot segment angle that minimized metabolic rate at each treadmill grade are shown with pictograms in Figure 4A.

R3.Q2. The complex statistical analyses are unfortunate because the question itself is very interesting. I wonder if there are cleaner ways to present this information. a) One such example is to focus on individual subject responses. b) Another might be to run a two way repeated measures ANOVA to determine how outsole angle and treadmill grade affect key dependent variables.

Thank you again for this advice. We hope that the changes we made which show that the statistical results are presented in the figures make the presentation easier to follow.

a) We appreciate the suggestion of the reviewer to present the individual subject responses.

We have already reported the mean \pm s.e.m. of the location of the minima for each subject with gray and colored crosses.

=> **We have now tried to highlight** this more clearly in the revised version of the manuscript by indicating the visualization of the **mean \pm s.e.m. of the individual minima with text labels** in Figure 4.

=> Following up on the recommendation of the reviewer, we have also **added a supplementary plot (Figure S7) that shows the individual trends** similar to the figure from the analysis of the supplementary data from Collins et al., [2015].

Figure S7. Individual participant responses of foot segment angle on metabolic rate during downhill, level, and uphill walking. Colored lines show the individual trends (every color is one participant). The thick black line shows the average trend. The dark gray crosses show the mean \pm s.e.m. of the locations of the minima.

We agree with the reviewer that this analysis can provide an idea of the variability in individual responses but we also believe that part of the variability in this analysis could be due to trial-to-trial variability (i.e., the landscape for a particular participant might be slightly different every time the participant is tested). Optimizing footwear for individual participants (impaired or unimpaired) would be an interesting application of modular footwear, but because of the trial-to-trial variability, optimization would preferably use techniques in which the level of statistical confidence in the location of the individual optimum is measured by repeating certain conditions (e.g., human-in-the-loop optimization).

=> We have **added the following (underlined) text to discuss individual responses** in the discussion: *Despite the considerable range of tested shoe inclinations, the effects of shoe inclination were relatively shallow and outweighed by the effects of the treadmill grade. There was also high variability in the individual trends (Figure S7) similar to studies with exoskeletons and prostheses [62-64]. This interindividual variability is likely a combination of real differences in optimal footwear for each individual and trial-to-trial variability from noisy metabolic rate measurements. A potential application of modular footwear would be to optimize footwear parameters in individual (impaired or unimpaired) participants using human-in-the-loop protocols [64,65] in which the footwear changes are prescribed by a gradient descent algorithm.*

We have added the following references:

Collins SH, Wiggin MB, Sawicki GS. Reducing the energy cost of human walking using an unpowered exoskeleton. *Nature*. 2015 Jun;522(7555):212.

Quesada RE, Caputo JM, Collins SH. Increasing ankle push-off work with a powered prosthesis does not necessarily reduce metabolic rate for transtibial amputees. *Journal of Biomechanics*. 2016 Oct 3;49(14):3452-9.

Zhang J, Fiers P, Witte KA, Jackson RW, Poggensee KL, Atkeson CG, Collins SH. Human-in-the-loop optimization of exoskeleton assistance during walking. *Science*. 2017 Jun 23;356(6344):1280-4.

Felt W, Selinger JC, Donelan JM, Remy CD. "Body-In-The-Loop": Optimizing Device Parameters Using Measures of Instantaneous Energetic Cost. *PloS one*. 2015 Aug 19;10(8):e0135342.

b) We appreciate the suggestion of the reviewer to simplify the statistical analyses by using a two-way repeated measures ANOVA. We consulted three independent biostatisticians who advised that linear-mixed effects model analysis is the most appropriate analysis when dependent variables are continuous (instead of categorical) variables.

Nevertheless, **we followed the suggestion of the reviewer and re-analyzed our data using repeated measures ANOVA**. Unfortunately, since we had already decided that we would not use repeated measures ANOVA before the data collection, our outsole angles and treadmill grades were not chosen in a way that we could have exactly the same foot segment angles for all 3 treadmill grades. Therefore, we could only run repeated measures ANOVA with the treadmill grade and outsole angle (instead of the foot segment angle) and we could only use the limited range of outsole angles that were tested under all three treadmill grades (the range is highlighted in yellow). We apologize that we did not choose our condition ranges differently but again we would like to point out that this is because we planned to use linear-mixed effects model analysis based on the consultations with the biostatisticians before we conducted the experiment.

The results from the repeated measures analysis showed a significant effect of treadmill grade ($P < 0.001$) but no significant effect of outsole angle ($P = 0.265$) and no significant interaction effect ($P = 0.899$). It may be surprising that the significant results of an experiment change depending on the statistical test but we hope to point out that this is not a finding unique to our study. For example, we have again reanalyzed the supplementary source data from Collins et al., [2015]. Using linear-mixed effects model analysis, the authors demonstrated a significant effect of stiffness ($P = 0.016$) and of the square of stiffness ($P = 0.008$) of their exoskeleton on metabolic rate. When we reanalyzed their data using repeated measures ANOVA, there was also no significant effect ($P = 0.216$) of the exoskeleton conditions on metabolic rate, which shows that it is not uncommon for the results of a study to depend on the choice of statistical test.

Additional table (not in manuscript). **Table R2: Metabolic rate data that were used to conduct repeated measures ANOVA.** Only the yellow highlighted conditions were used since these conditions contained data from outsole angles that were repeated for each treadmill grade.

	Metabolic rate (W/kg)																				
	-6	-6	-6	-6	-6	-6	0	0	0	0	0	0	0	6	6	6	6	6	6		
Treadmill grade (deg.)	-12	-7	-3	0	3	7	12	-12	-7	-3	0	3	7	0	-12	-7	-3	0	3		
Outsole angle (deg.)	-18	-13	-9	-6	-3	1	6	-12	-7	-3	0	3	7	0	-6	-1	3	6	9		
Foot segment angle (deg.)																					
s1	N/A	N/A	1.937	2.149	2.140	2.379	2.101	N/A	4.097	3.863	3.684	3.856	3.997	N/A	8.024	7.820	7.352	7.333	7.920	N/A	N/A
s2	N/A	N/A	1.720	1.899	1.882	1.670	1.901	N/A	3.034	2.838	3.102	3.219	2.955	N/A	6.600	6.291	6.616	6.276	6.284	N/A	N/A
s3	N/A	N/A	2.010	1.823	2.729	1.945	2.631	N/A	3.614	3.397	3.073	3.621	3.633	N/A	8.131	7.049	6.805	7.331	6.965	N/A	N/A
s4	N/A	N/A	2.212	2.110	1.898	1.948	N/A	N/A	4.164	3.171	3.042	3.168	3.379	N/A	7.227	7.471	6.945	6.553	7.238	N/A	N/A
s5	N/A	N/A	1.698	2.149	1.851	1.300	2.149	N/A	3.133	3.089	3.136	3.185	2.757	N/A	6.741	6.253	6.562	6.548	6.540	N/A	N/A
s6	N/A	N/A	2.017	1.946	1.886	1.921	2.145	N/A	3.237	3.878	2.922	3.232	3.537	N/A	7.142	6.822	6.679	7.523	6.728	N/A	N/A
s7	N/A	N/A	1.673	1.487	1.386	1.535	1.939	N/A	2.952	2.764	3.568	2.737	2.580	N/A	6.761	6.763	6.244	6.362	6.681	N/A	N/A
s8	N/A	N/A	1.485	1.764	1.490	1.402	1.596	N/A	2.847	2.776	2.737	2.806	2.899	N/A	6.526	6.454	6.642	6.488	6.483	N/A	N/A
s9	N/A	N/A	2.599	2.866	2.508	2.214	2.475	N/A	5.358	3.544	4.726	3.015	5.450	N/A	7.355	7.628	7.182	7.641	7.340	N/A	N/A
s10	N/A	N/A	2.294	3.159	1.876	2.204	2.752	N/A	4.583	3.528	4.229	3.903	3.680	N/A	7.113	8.559	7.953	8.235	8.105	N/A	N/A

Additional table (not in manuscript). **Table R3: Within-subject effects of treadmill grade and shoe outsole angle and their interaction using repeated measures ANOVA.**

Tests of Within-Subjects Effects						
Measure: MEASURE_1						
Source		Type III Sum of Squares	df	Mean Square	F	Sig.
treadmill	Sphericity Assumed	397.246	2	198.623	1759.637	.000
	Greenhouse-Geisser	397.246	1.460	272.056	1759.637	.000
	Huynh-Feldt	397.246	1.671	237.681	1759.637	.000
	Lower-bound	397.246	1.000	397.246	1759.637	.000
Error(treadmill)	Sphericity Assumed	2.032	18	.113		
	Greenhouse-Geisser	2.032	13.141	.155		
	Huynh-Feldt	2.032	15.042	.135		
	Lower-bound	2.032	9.000	.226		
outsole	Sphericity Assumed	.343	2	.171	1.430	.265
	Greenhouse-Geisser	.343	1.326	.259	1.430	.266
	Huynh-Feldt	.343	1.468	.234	1.430	.267
	Lower-bound	.343	1.000	.343	1.430	.262
Error(outsole)	Sphericity Assumed	2.159	18	.120		
	Greenhouse-Geisser	2.159	11.937	.181		
	Huynh-Feldt	2.159	13.210	.163		
	Lower-bound	2.159	9.000	.240		
treadmill * outsole	Sphericity Assumed	.101	4	.025	.263	.899
	Greenhouse-Geisser	.101	2.499	.040	.263	.817
	Huynh-Feldt	.101	3.537	.029	.263	.880
	Lower-bound	.101	1.000	.101	.263	.620
Error(treadmill*outsole)	Sphericity Assumed	3.456	36	.096		
	Greenhouse-Geisser	3.456	22.492	.154		
	Huynh-Feldt	3.456	31.830	.109		
	Lower-bound	3.456	9.000	.384		

A possible reason why repeated measures ANOVA did not show a significant effect whereas linear-mixed effects model analysis did show significant effects for certain parameters in our study and in the Collins et al., [2015] study could be that repeated measures ANOVA considers the condition variables as categorical variables. For example, repeated measures analysis does not “know” that the data from the 0 degree outsole angle condition comes from trials with a shoe angle that falls in between the -3° and +3° shoe angle condition. Another possible reason could be that the outsole angle conditions were too close to each other.

Since we were advised by three independent biostatisticians that repeated measures ANOVA is not the most appropriate analysis, we considered the fact that repeated measures ANOVA did not show a

significant effect of outsole angle as a type II error because the test is not sensitive enough. We would therefore politely request to not implement the suggestion of the referee to use repeated measures ANOVA.

We do not interpret the fact that we found significant effects for the square of foot segment angle and the interaction with treadmill grade as an indication that these effects are large. In contrast, we agree with the comments from previous reviewers that the effects are shallow but significant, and we have mentioned this in the discussion.

=> We have **added a limitation statement** (underlined) in the discussion to acknowledge that the statistical results might be different when using different statistical analyses:

It is possible that different results might have been found if we had used less sensitive statistical analyses (e.g., repeated measures ANOVA) or if we had tested a larger number of participants or a larger range of shoe inclinations.

References (already in manuscript):

Collins SH, Wiggin MB, Sawicki GS. Reducing the energy cost of human walking using an unpowered exoskeleton. *Nature*. 2015 Jun;522(7555):212.

Source data: http://biomechatronics.cit.cmu.edu/publications/Collins_2015_Nature---SourceData.zip

R3.Q3. It was also unclear whether foot segment angle was a dependent variable because, according to P8, ~L50, it was computed using individual stride data. Yet, it appears to be referenced in results as an independent variable. Should outsole angle not be consistently referenced instead?

The reviewer makes a very good point. We could have measured the outsole angle during benchtop tests without human subjects (e.g., using an angle measuring tool). However, since our statistical analysis uses the foot segment angle as a continuous variable **we wanted to measure the foot segment angle very precisely when the shoes were loaded by the human subjects standing in them.** We accomplished this by attaching the shoe markers at exactly the same height as that of the outsole (using lines on the shoe) and by paying close attention to the bubble level indicator on the L-frame of our motion capture system.

As far as we know this approach of obtaining a more realistic measure of the independent variable during the actual human experiments, instead of during benchtop experiments, is not unique to our study. For example, a number of parameter sweep studies have investigated the effects of certain actuation parameters (i.e., actuation magnitude or timing) of robotic exoskeletons and prostheses on metabolic cost [Galle et al., 2017; Quinlivan et al., 2017; Jackson and Collins, 2015; Caputo and Collins, 2014]. The control software for these robotic exoskeletons and prostheses often allows us to specify the values of the actuation parameters that are to be tested. However, this control is often not very precise, and therefore, the actuation parameters that are the independent parameters are measured during the human experiments together with metabolic rate (dependent parameter). After this, the relationship of metabolic rate is analyzed relative to the actually measured independent parameters.

We were unable to find any mention of using the foot segment angle as a dependent variable in the results section as mentioned by the reviewer. In fact, the foot segment angles are specified in the methods. Our apologies if we overlooked something.

=> **We have added the following text** in the 2.6 data processing section in methods **to make it clear that the foot segment angle was used as an independent parameter** in the statistical analyses:

This foot segment angle that was measured during realistic loading conditions was used as an independent parameter in further analyses together with our second independent parameter, treadmill grade.

Reference (already in the manuscript):

Galle S, Malcolm P, Collins SH, De Clercq D. Reducing the metabolic cost of walking with an ankle exoskeleton: interaction between actuation timing and power. *Journal of neuroengineering and rehabilitation*. 2017 Dec;14(1):35.

Additional references not in manuscript:

Quinlivan BT, Lee S, Malcolm P, Rossi DM, Grimmer M, Siviý C, Karavas N, Wagner D, Asbeck A, Galiana I, Walsh CJ. Assistance magnitude versus metabolic cost reductions for a tethered multiarticular soft exosuit. *Sci Robot.* 2017 Jan 18;2(2):4416.

Jackson RW, Collins SH. An experimental comparison of the relative benefits of work and torque assistance in ankle exoskeletons. *Journal of Applied Physiology.* 2015 Jul 9;119(5):541-57.

Caputo JM, Collins SH. Prosthetic ankle push-off work reduces metabolic rate but not collision work in non-amputee walking. *Scientific reports.* 2014 Dec 3;4:7213.

R3.Q4. a) More broadly, it feels like an individual subject analysis/discussion is missing in this manuscript. If, as stated in the response to Reviewer 1, the metabolic landscape isn't visible for individual subjects then this is an important and noteworthy finding and I would encourage devoting a figure to this (similar to the Collins example provided). Presenting individual responses in metabolic rate is particularly crucial because the authors state that this work could be applicable to clinicians prescribing footwear to populations with limited effort capacity, and therefore having an understanding of individual variation is useful. b) Given the similar U-shaped relationship uncovered for soleus activity, it would also be beneficial to know how individual subjects responded here.

a) We thank the reviewer for the advice.

=> We have **added the individual analysis to the supplementary material. We have also added discussion** in response to question 2a) from the reviewer (R3.Q2.a)).

b) We found no significant effect for the first order of foot segment angle which indicates that the foot segment angle that results in the minimum soleus EMG is not significantly different from zero. Consequently, we also eliminated this term from the statistical model to avoid overfitting; therefore, we originally did not report the variability in minima for soleus EMG in Figure 6.

Because of the suggestion of the reviewer, we rerun our statistical analysis without eliminating the non-significant term so that we could report the variability in the foot segment angle that minimizes soleus EMG. => We have **added the following (underlined) text in the 3.3 ankle mechanics and muscle activation section in results to describe the individual variation in soleus activity:**

*We found a U-shaped trend in average soleus EMG versus c_2 (square of foot segment angle, $P = 0.002$; Table S1; Figure 6A). The c_1 (foot segment angle) and the c_5 (foot segment angle and treadmill grade) did not have significant effects on soleus EMG which indicates that a foot segment angle of 0° minimized soleus EMG for downhill, level and uphill walking. Evaluating the statistical model without eliminating the non-significant terms showed that the **individual minima varied with a s.e.m. of 1.1, 0.9 and 0.7 for downhill, level and uphill walking, respectively.***

R3.Q5. One way to quantify this might be use a residual analysis to assess the quality of a second order polynomial fit to the individual subject data points at each grade.

We thank the reviewer for this original suggestion. As suggested by the reviewer in question 2 (R3.Q2) and question 4 (R3.Q4), we have calculated individual coefficients of the different possible statistical model options (Table R4). Every statistical model was fitted with individualized coefficients for all the trials of each participant (i.e., there are not separate curve fits using only the data for one treadmill grade, doing so would, of course, lead to better R^2). For every participant, we then evaluated how well the equation that fit all the trials explained the changes within each treadmill grade, as suggested by the reviewer.

Additional table (not in manuscript). **Table R4: Residual analysis for metabolic rate for different statistical model options.** Values in the table are mean \pm s.e.m. of the R^2 of the individual fits for each participant.

Model (X = foot segment angle, Y = treadmill grade)	Mean \pm s.e.m. of R^2 of individual fits for all conditions	Mean \pm s.e.m. of R^2 of individual fits for downhill conditions	Mean \pm s.e.m. of R^2 of individual fits for level conditions	Mean \pm s.e.m. of R^2 of individual fits for uphill conditions
Model with all possible terms: $C_0 + C_1 * X + C_2 * X^2 + C_3 * Y + C_4 * Y^2 + C_5 * (X * Y)$	0.9772 \pm 0.0081	0.2653 \pm 0.1164	0.3572 \pm 0.1031	0.3252 \pm 0.0913
Model without second order term for foot segment: angle $C_0 + C_1 * X + C_2 * X^2 + C_3 * Y + C_4 * Y^2 + C_5 * (X * Y)$	0.9764 \pm 0.0086	0.1505 \pm 0.0424	0.1336 \pm 0.0451	0.1531 \pm 0.0478
Model with second order term but without first order term for foot segment angle (this model was selected by stepwise elimination of non-significant contributors) $C_0 + C_1 * X + C_2 * X^2 + C_3 * Y + C_4 * Y^2 + C_5 * (X * Y)$	0.9771 \pm 0.0081	0.2655 \pm 0.1291	0.4098 \pm 0.1006	0.3157 \pm 0.0896

The results from Table R4 show that the elimination of the second order term lowers the R^2 of the individual fits for each treadmill grade by approximately 0.16 (numbers marked in red). The results also show that elimination of the first order term does not drastically lower the R^2 (sometimes there is even a slight improvement). Overall, this appears to confirm that our original statistical approach (i.e., stepwise elimination of non-significant contributors) was effective.

We appreciate the suggestion of the reviewer regarding the residual analysis, however

- As far as we know, there is no clear guideline for how much of a change in R^2 is needed to justify the addition or removal of a term, whereas there is a clear threshold for statistical significance ($P = 0.05$), which is widely used (even though it is also arbitrary).
- We prefer to select the terms based on statistical significance testing because it is more common to use statistical significance testing to accept or reject hypotheses.

Therefore, we would like to kindly request to keep the original statistical method which started with a statistical model that was based on the assumptions that we wanted to test and then exclude terms that did not significantly contribute but keeping terms that were significant.

=> **We have added the following (underlined) text** in the 3.1 metabolic rate section in results:
Supplementary residual analyses confirmed that the inclusion of the square of foot segment angle improved the fits of the individual participant data, whereas the first order of foot segment angle did not improve the fits.

MINOR COMMENTS

R3.Q6. P4 L6: Alongside this reference it may be worth noting that humans find the metabolically optimal locomotor pattern during novel situations when exposed to the broader energetic landscape.

=> We thank the reviewer for this excellent suggestion. **We have implemented the suggestion from the reviewer** (changes are underlined) in the introduction:

Humans are able to find the metabolically optimal locomotor pattern during novel situations [1] when exposed to the broader energetic landscape, rather than only being able to be metabolically economic during previously known gaits.

R3.Q7. P7 Paragraph 1: This is a long protocol, was metabolic data checked for drift?

We agree with the reviewer that this was a relatively long protocol. We plotted the metabolic data from all conditions in chronological order and it seems that there is indeed a certain amount of drift occurring (Figure R2). This is probably not unique to our study since **other studies have tested a similar number of conditions in a single session** [Galle et al., 2017; Quinlivan et al., 2017; Jackson and Collins, 2015; Caputo and Collins, 2014; Grabowski et al., 2005; Sanchez et al., 2019; Abram et al., 2019].

To avoid that drift would cause a consistent bias to our results, we did the following:

- We randomized all conditions using a random order generating software.
- The order of the conditions for every second participant was the inverse of the order of the previous participant. As such, all conditions were symmetrically distributed over the first and second halves of the protocol, which should have eliminated the effects of drift on average.

Additional figure (not in manuscript). **Figure R2: Metabolic drift.** Individual participant responses in metabolic rate during all conditions (every color is one participant). The conditions were plotted according to when they were tested in the protocol. Since the order of the conditions was randomized across participants, the average trend (thick black line) gives an idea of the average drift in metabolic cost.

=> **We have added the following (underlined) text** in the 2.4 protocol section in methods to **explain how we avoided potential bias from metabolic drift:**

The conditions (shoe inclination angles and treadmill grades) were semi-randomized for each participant using a random order generating software, and to minimize the number of shoe inclination angle changes, we tested each shoe inclination on all applicable treadmill grades before changing to the next shoe inclination. The order of the conditions for every second participant was the inverse of the order of the previous participant. As such, all conditions were symmetrically distributed over the first and second halves of the protocol to avoid the effects of metabolic drift on the results.

Reference (already in the manuscript):

Galle S, Malcolm P, Collins SH, De Clercq D. Reducing the metabolic cost of walking with an ankle exoskeleton: interaction between actuation timing and power. Journal of neuroengineering and rehabilitation. 2017 Dec;14(1):35.

Additional references not in manuscript:

Quinlivan BT, Lee S, Malcolm P, Rossi DM, Grimmer M, Siviyy C, Karavas N, Wagner D, Asbeck A, Galiana I, Walsh CJ. Assistance magnitude versus metabolic cost reductions for a tethered multiarticular soft exosuit. Sci Robot. 2017 Jan 18;2(2):4416.

Jackson RW, Collins SH. An experimental comparison of the relative benefits of work and torque assistance in ankle exoskeletons. Journal of Applied Physiology. 2015 Jul 9;119(5):541-57.

Caputo JM, Collins SH. Prosthetic ankle push-off work reduces metabolic rate but not collision work in non-amputee walking. Scientific reports. 2014 Dec 3;4:7213.

Grabowski A, Farley CT, Kram R. Independent metabolic costs of supporting body weight and accelerating body mass during walking. Journal of Applied Physiology. 2005 Feb;98(2):579-83.

Sanchez N, Simha SN, Donelan JM, Finley JM. Taking advantage of external mechanical work to reduce metabolic cost: the mechanics and energetics of split-belt treadmill walking. The Journal of physiology. 2019 Jun 13.

Abram SJ, Selinger JC, Donelan JM. Energy optimization is a major objective in the real-time control of step width in human walking. Journal of biomechanics. 2019 Jun 25;91:85-91.

R3.Q8. Also, it may be easier on the reader to list, in-text, the exact treadmill and outsole angles tested instead of a range.

=> Thank you for your suggestion. **We have implemented the suggestion from the reviewer** to list the exact angles (underlined) in the 2.3 experimental conditions section in methods:

Specifically, five footwear inclinations (-3°, 0°, 3°, 7° and 12°) were tested during (-6°) downhill walking, five footwear inclinations (-7°, -3°, 0°, 3° and 7°) were tested during level walking, and five footwear inclinations (-12°, -7°, -3°, 0° and 3°) were tested during (+6°) uphill walking.

R3.Q9. P7 L32: Can the authors clarify (in-text) that the shoe inclination angles were randomized for each participant and the treadmill grades were randomized for each participant?

=> As suggested by the reviewer, we have **added the following (underlined) text** in the 2.4 protocol section in methods to clarify that the shoe inclination angles were randomized for each participant and that the treadmill grades were randomized for each participant:

The conditions (shoe inclination angles and treadmill grades) were semi-randomized for each participant using a random order generating software, and to minimize the number of shoe inclination angle changes, we tested each shoe inclination on all applicable treadmill grades before changing to the next shoe inclination.

R3.Q10. P8 L27: 6Hz seems very low for ground reaction force data. What was the rationale for this?

We thank the reviewer for verifying this. The reviewer is correct that there are studies that used higher cut-off frequencies for walking data, especially when analyzing data from walking with faster speeds or walking under conditions that could elicit rapid movements (e.g., with sudden perturbations from an exoskeleton). However, we also considered the following:

- For normal walking, a 6 Hz cut-off frequency appears to be a **suggested cut-off frequency in handbooks** [Winter, 1991].

- We could have used a higher frequency specifically for the ground reaction force data but since we combined ground reaction force data with kinematic data **for inverse dynamics analysis, we chose to filter the ground reaction force data at the same frequency as our kinematic data** because of the noise sensitivity of the inverse dynamics calculations [e.g., as in Voloshina et al., 2015].

- A **6 Hz cut-off frequency was also used in similar studies** with uneven terrain similar to our study [e.g., Voloshina et al., 2013; Panizzolo et al., 2017].

- Since **our study was conducted at a relatively low walking speed** of 1 m·s⁻¹, and since we focused on relative changes in inverse dynamics and did not focus on reporting high-frequency parameters such as peak ground reaction forces or vertical loading rate, it is likely that a 6 Hz cut-off frequency was adequate.

- We **compared** the chosen 6Hz cut-off frequency **to a cut-off frequency of 10Hz** using the data from one participant, which confirmed that the **differences were negligible** and that the 6Hz cut-off frequency was adequate.

=> **We have added the following references** (that used the same cut-off frequency for calculating inverse dynamics from data from similar uneven terrain walking) in our sentence in the 2.6 data processing section in methods:

We filtered marker positions and ground reaction forces with a fourth-order low-pass Butterworth filter with a 6 Hz cut-off frequency [35,41].

References (already in the manuscript):

Voloshina AS, Kuo AD, Daley MA, Ferris DP. Biomechanics and energetics of walking on uneven terrain. *Journal of Experimental Biology*. 2013 Nov 1;216(21):3963-70.

Panizzolo FA, Lee S, Miyatake T, Rossi DM, Siviyy C, Speeckaert J, Galiana I, Walsh CJ. Lower limb biomechanical analysis during an unanticipated step on a bump reveals specific adaptations of walking on uneven terrains. *Journal of experimental biology*. 2017 Nov 15;220(22):4169-76.

Additional references not in manuscript:

Winter DA. Biomechanics and motor control of human gait: normal, elderly and pathological. 1991.

Voloshina AS, Ferris DP. Biomechanics and energetics of running on uneven terrain. *Journal of Experimental Biology*. 2015 Mar 1;218(5):711-9.

R3.Q11. P8 L43: Artefacts

Thank you for finding this typo. We have corrected this error.

R3.Q12. P8 L44: Heel strike detection – via ground reaction force or marker data?

=> We detected heel strike using the GRF. We have added the following (underlined) **text in the 2.6 data processing section in methods to clarify this:**

We segmented each time series based on heel strike detection (using the GRF) and calculated the representative profile per stride by taking the median of all strides.

R3.Q13. P9 L5: Another option for this would be to consider work as J/kg/m (a mechanical cost of transport incorporating the distance traveled per step in case step frequency changes in different conditions, which I suspect it may), and metabolic data as a cost of transport too.

Thank you for this suggestion. The reviewer is correct that the conditions could affect step length. In fact, we found an increase in step length with increasing upward foot segment angles and an increase in step length with increasing uphill treadmill grades.

To account for possible changes in step length, we have reported all mechanical work results as **mechanical work rates in Watt/kg** (instead of Joule/kg).

We could convert all these mechanical work rate and metabolic rate results from W/kg to J/kg/m by dividing all the results by the walking speed. **Since the walking speed was kept constant at 1 m·s⁻¹**, this would be equivalent to dividing all the results by 1. As such, the **results would be exactly the same** as in the current version of the manuscript and therefore, we did not implement this suggestion.

Other studies in the literature have **reported** mechanical work as **work rates for similar reasons** [e.g., Caputo and Collins, 2014; Jackson and Collins, 2015; Donelan et al., 2002; Zelik and Kuo, 2010; Minetti et al., 2002].

=> We have added the following (underlined) sentence in the 2.6 data processing section in methods:
We chose to report joint and COM mechanics as work rates (in W) instead of as work (in J) so that they had comparable units to those of metabolic rate and to avoid the potential for changes in step frequency to confound the mechanical work results.

References (already in the manuscript):

Minetti AE, Moia C, Roi GS, Susta D, Ferretti G. Energy cost of walking and running at extreme uphill and downhill slopes. *Journal of applied physiology*. 2002 Sep 1;93(3):1039-46.

Additional references not in manuscript:

Donelan JM, Kram R, Kuo AD. Mechanical work for step-to-step transitions is a major determinant of the metabolic cost of human walking. *Journal of Experimental Biology*. 2002 Dec 1;205(23):3717-27.

Zelik KE, Kuo AD. Human walking isn't all hard work: evidence of soft tissue contributions to energy dissipation and return. *Journal of Experimental Biology*. 2010 Dec 15;213(24):4257-64.

Jackson RW, Collins SH. An experimental comparison of the relative benefits of work and torque assistance in ankle exoskeletons. *Journal of Applied Physiology*. 2015 Jul 9;119(5):541-57.

Caputo JM, Collins SH. Prosthetic ankle push-off work reduces metabolic rate but not collision work in non-amputee walking. *Scientific reports*. 2014 Dec 3;4:7213.

R3.Q14. P9 L11: How did the authors confirm their sample size was large enough?

Thank you for this comment. **We were able to confirm that the number of participants, and the number and range of conditions appear to have been adequate to reject that the coefficients of the following (5 out of 6) factors are equal to zero (null hypothesis):**

- Intercept (**P < 0.001**)

This indicates that metabolic rate when walking with level shoes on a level treadmill grade is different from zero.

- Foot segment angle² (**P = 0.013**)

This indicates that metabolic rate follows a U-shaped trend versus foot segment angle (as we hypothesized).

- Treadmill grade (**P < 0.001**)

This indicates that the minimum of the U-shaped trend is not at a treadmill grade of zero degrees (as known from the literature, e.g., Minetti et al., 1993).

- Treadmill grade² (**P < 0.001**)

This indicates that metabolic rate follows a U-shaped trend versus treadmill grade (as known from the literature, e.g., Minetti et al., 1993).

- Foot segment angle · treadmill grade (**P = 0.023**)

This indicates that the minimum of metabolic rate versus foot segment angle is different between treadmill grade conditions.

In order to be able to confirm that the sample size was large enough to confirm the non-significant factor (i.e., foot segment angle), it would be interesting if there would be a way to conduct a statistical power analysis. Linear mixed-effects model analysis is a sensitive statistical test, therefore it is likely that the non-significance is not a type I error. However, there appears to be no accepted general standard for statistical power analysis with mixed-effects model analysis [Castelloe and O'Brien, 2001; Guo et al., 2013]. However, since the P-value of the non-significant term is rather far from significant in our study, it is also very unlikely that this effect would become significant with a larger number of participants.

We provide the following text about the non-significant factor in the discussion:

It is possible that different results might have been found if we had used less sensitive statistical analyses (e.g., repeated measures ANOVA) or if we had tested a larger number of participants or a larger range of shoe inclinations. While the number of participants and the range of shoe inclinations have been adequate to reject the null hypothesis for 5 of the 6 coefficients in our statistical model for metabolic rate (P-values ≤ 0.023), we have no certainty as to whether the non-significant effect of the first order of foot segment angle on metabolic rate (P = 0.318) is related to our sample size (n = 10).

Reference (already in the manuscript):

Minetti AE, Ardigo LP, Saibene F. Mechanical determinants of gradient walking energetics in man. The Journal of physiology. 1993 Dec 1;472(1):725-35.

Additional references not in manuscript:

Castelloe JM, O'Brien RG. Power and sample size determination for linear models. In Proceedings of the Twenty-Sixth Annual SAS Users Group International Conference 2001 Apr (pp. 240-26).

Guo Y, Logan HL, Glueck DH, Muller KE. Selecting a sample size for studies with repeated measures. BMC medical research methodology. 2013 Dec;13(1):100.

R3.Q15. P10 L11: "The P-value associated with c4 will inform us whether a variable follows a parabolic trend versus treadmill grade." – Some outsole angles did not have three samples. Is this only applicable to the range of shared data points across the three treadmill grades?

We apologize that we had not explained this sufficiently. **All 15 conditions were used as input to the linear mixed-effects model analysis but we only plotted (i.e., interpolated) the parabolic trend for the shoe inclination conditions that were tested on all three treadmill grades.**

=> We have **added the following clarification in the captions** of Figures 4-7 (B):

Colored lines represent the formula from the linear mixed-effects model analysis evaluated over the range of treadmill grades. Although five shoe inclinations were tested at each treadmill grade and were included in the evaluation of the statistical model, we only plotted the lines representing the evaluation of the linear mixed-effects model analysis for the three shoe inclinations that were tested on all three treadmill grades.

R3.Q16. P11: Statements in Results that extrapolate beyond the experimental data could be excluded in an attempt to simplify this section. Examples:

"As for the effect of treadmill angle, the resulting trends indicate that a -7deg (downhill) treadmill grade minimized metabolic rate." "By extrapolating our data, we found that the positive COM work rate and average propulsive GRF are minimized around a downhill treadmill grade of -8°."

We thank the reviewer for the help with simplifying the manuscript. **We have removed these sentences** from the results.

R3.Q17. P11 L11: For consistency, perhaps referring to C0-C5 would be easier than now referring to the “square of foot segment angle...” for example.

Thank you for this suggestion.

=> We have consulted with different colleagues and we would kindly request that we could still keep the description of the terms but **we have also added the reference to C0-C5** next to the description of each term in the results. One example to illustrate our changes to the text can be found below (changes are underlined):

We found significant effects of c_2 (square of foot segment angle, $P = 0.013$; Table S1; Figure 4A) on metabolic rate indicating that the effect of foot segment angle on metabolic rate at each treadmill grade followed a parabolic trend.

R3.Q18. P16 L21, P17 L17: What is rolling terrain?

Thank you for asking this question. Our intended meaning of rolling terrain is that there are natural slopes gently rising above and falling below the terrain grade.

=> We **have added this clarification** (underlined text) in the discussion:

The results of our study could also inform designs of shoes for walking on rolling terrain (i.e., natural slopes gently rising above and falling below the terrain grade).

R3.Q19. P16: Is it possible additional shoe sole material reduces sensory perception and perhaps impairs stability in uneven terrain environments?

Thank you for this question. We agree with the reviewer that the additional material on the shoe could affect sensory perception and stability during walking on uneven terrains.

=> We have **added the following limitation statement** in the discussion:

*The additional shoe material could also have reduced sensory **perception** and impaired **stability** during downhill or uphill walking.*

R3.Q20. P16 L38-47: I would consider removing this text.

We appreciate the suggestion of the reviewer to simplify the discussion text.

=> **We have followed the suggestion and removed this text.**

R3.Q21. Figure 2: When printed in BW this figure is very hard to interpret - consider adding thin lines to the edges of the shaded regions. Also consider removing 4, 2, -2, -4 from the y axis.

If foot segment angle is to be incorporated in this figure, it is hard to read from the x-axis. For example, the far left shoe with -7 slope and 0 treadmill angle seems to align best with -8 foot segment angle, not -7. Perhaps consider writing above/below each shoe what the resulting foot segment angle is. Or, if foot segment angle is a dependent variable, then perhaps consider presenting this as a separate table in the results with mean±standard error.

Thank you for these suggestions for improving the visual appeal of Figure 2.

=> The **revised figure** can be found below:

R3.Q22. Figures 4-7: Angle lines on shoes are hard to see, maybe these could be thicker and white?

Thank you again for this suggestion.

=> We understand the concern that this was not clear enough and **we have implemented these suggestions** to improve the lines on the shoes in all figures.

Reviewer: 4

Comments to the Author(s)

Review of "Modular footwear that partially offsets downhill or uphill grades minimizes the metabolic cost of human walking"

Major Concerns

R4.Q1. Much stronger justification is needed for background, hypotheses, and application of the results of the proposed study. a) Why is it important to keep damping and bending stiffness of the shoe constant? What were the values for damping and bending stiffness and how were these verified across conditions? b) Why was muscle activity measured from the specified 7 muscles of the leg? c) Why was individual leg power calculated? d) Why was only Soleus muscle activity reported?

We thank the reviewer for reviewing the paper and for the advice regarding the justification of the different aspects of the study. We strongly appreciate this advice and understand how this will improve the paper.

a) In footwear research, we have seen a number of studies that compare footwear prototypes that differ in more than one property [e.g., Hoogkamer et al., 2018; Hoogkamer et al., 2019]. Such an experiment has the advantage that it is more externally valid (e.g., if it uses footwear that is commercially available to athletes and/or patients). However, when footwear differ in more than one property, it can make it challenging to investigate the underlying mechanisms.

Since our goal was to investigate the relationships between metabolic rate and biomechanics, we chose to conduct our study according to the definition of a "controlled experiment". A controlled experiment is defined as a scientific test that is directly manipulated by a scientist to test a single variable at a time (the independent variable). The controlled variables are held constant to minimize or stabilize their effects on the subject. In this sense, our study was inspired by studies on prosthetic feet where the authors attempted to isolate one parameter [e.g., foot radius by Adamczyk and Kuo, 2013 and toe shape by Honert et al., 2018].

=> We have revised (changes are underlined) the following text in the aim of our study:

*Using modular footwear as a new tool for altering treadmill grade while keeping the foot segment angle constant (by offsetting the treadmill grade) could provide new insights into the relationships between biomechanical changes and the resulting changes in metabolic rate. **Studies on negative heel shoes [15,24] and shoes with different heel heights [25-27] have used footwear that are commonly available but differ in more than one property (outsole geometry, hardness, etc.). Using commonly available footwear has the advantage of being more ecologically valid but makes it difficult to attribute changes in metabolic rate and biomechanics to a specific shoe parameter. Hence, the aim of this study was to investigate the interaction effects of varying footwear outsole geometry and treadmill grade on metabolic rate, joint mechanics, muscle activity, ground reaction forces and COM mechanics in a controlled experiment (all footwear and treadmill parameters were kept constant except the outsole geometry and treadmill grade).***

Previous studies have shown that the following footwear parameters affect energetics and biomechanics (and could therefore confound the effects of geometry):

- Vertical **hardness** [Nigg et al., 1987; Morio et al., 2009]
- **Bending stiffness** [Takahashi et al., 2016; Roy and Stefanyshyn, 2006]
- Weight [Frederick, 1984, Browning et al., 2007]
- Outsole angle [Mika et al., 2012; Thies et al., 2015]

Additional figure (not in manuscript). **Figure R3: Illustration of how single part outsoles could affect hardness and bending stiffness together with outsole geometry.**

In studies with high heeled or negative heel shoes in which the outsole is designed as a single section, it is likely that changes in the outsole geometry affect parameters such as hardness (because the material can compress over a longer distance in the thicker part) and bending stiffness (because the material will be more resistant to bending in the thicker part) (Figure 1B). For this reason, we designed the outsoles in separate blocks and used (almost) non-compressible material for the blocks so that there would be almost no difference in hardness for the thicker and thinner areas of the outsole.

We have quantified the bending stiffness and hardness based on the reviewer's comment:

- We quantified the bending stiffness by attaching one half of the shoe and moving the other half through its range of motion with a tension dynamometer. The shoe was attached such that the bending occurred horizontally to avoid gravity would affecting the dynamometer readings. We found an average (relatively low) bending stiffness of 0.0011 Nm/deg. The bending stiffness was not affected by the blocks since they were mounted such that they could move independently.

- We tested the hardness of the wooden blocks by compressing a stack of wooden blocks and measuring the compression with a caliper. This test confirmed that the wooden blocks behaved like a rigid segment (within a reasonable range of forces for walking).

Additional figure (not in manuscript). **Figure R4: Block hardness test.** Since the blocks were made of (hard) pieces of wood, we could not observe any measurable compression.

=> We **have added the following (underlined) text** in the 2.2 modular footwear section in methods **to explain the need for controlling bending stiffness and hardness** and have provided the measurements: Outsole geometry was altered by attaching blocks with different heights to the sole. In studies with high-heel shoes or negative-heel shoes in which the outsole is designed as a single block [15,24,25,27], it is possible that the thicker parts of the outsoles have a greater bending stiffness and lower vertical

hardness, which can both affect walking biomechanics [28,29]. We chose a design with separate blocks as opposed to single wedge-shaped outsoles to avoid differences in outsole stiffness due to the outsole inclination. The blocks were made from rigid material (1.27 and 0.32 cm medium-density fiberboard; **compression tests with a clamp and digital caliper did not show deformation**) to avoid having different block heights result in differences in vertical hardness. **Tests of the bending stiffness when dorsiflexing the toe region showed an average stiffness of 0.0011 Nm:deg⁻¹.**

=> We have added an extra image to Figure 1 (i.e., Figure 1B) to show that the blocks did not affect the bending of the outsole, and we have added a note to clarify that the blocks were made out of wood that does not compress over the expected range of forces.

Figure 1. B: The attachment of the blocks does not change the bending stiffness.

We have added the following references:

Morio C, Lake MJ, Gueguen N, Rao G, Baly L. The influence of footwear on foot motion during walking and running. *Journal of biomechanics*. 2009 Sep 18;42(13):2081-8.

Takahashi KZ, Gross MT, Van Werkhoven H, Piazza SJ, Sawicki GS. Adding stiffness to the foot modulates soleus force-velocity behaviour during human walking. *Scientific reports*. 2016 Jul 15;6:29870.

References (already in the manuscript):

Stefanyshyn DJ, Nigg BM, Fisher V, O'Flynn B, Liu W. The influence of high heeled shoes on kinematics, kinetics, and muscle EMG of normal female gait. *Journal of Applied Biomechanics*. 2000 Aug 1;16(3):309-19.

Mika A, Oleksy Ł, Mika P, Marchewka A, Clark BC. The influence of heel height on lower extremity kinematics and leg muscle activity during gait in young and middle-aged women. *Gait & posture*. 2012 Apr 1;35(4):677-80.

Simonsen EB, Svendsen MB, Nørreslet A, Baldvinsson HK, Heilskov-Hansen T, Larsen PK, Alkjær T, Henriksen M. Walking on high heels changes muscle activity and the dynamics of human walking significantly. *Journal of applied biomechanics*. 2012 Feb 1;28(1):20-8.

Li J. Gait and metabolic adaptation of walking with negative heel shoes. *Research in Sports Medicine*. 2003 Dec 1;11(4):277-96.

Li JX, Hong Y. Kinematic and electromyographic analysis of the trunk and lower limbs during walking in negative-heeled shoes. *Journal of the American Podiatric Medical Association*. 2007 Nov;97(6):447-56.

Additional references not in manuscript:

Hoogkamer W, Kipp S, Frank JH, Farina EM, Luo G, Kram R. A comparison of the energetic cost of running in marathon racing shoes. *Sports Medicine*. 2018 Apr 1;48(4):1009-19.

Hoogkamer W, Kipp S, Kram R. The biomechanics of competitive male runners in three marathon racing shoes: a randomized crossover study. *Sports Medicine*. 2019 Jan 25;49(1):133-43.

Adamczyk PG, Kuo AD. Mechanical and energetic consequences of rolling foot shape in human walking. *Journal of Experimental Biology*. 2013 Jul 15;216(14):2722-31.

Honert EC, Bastas G, Zelik KE. Effect of toe joint stiffness and toe shape on walking biomechanics. *Bioinspiration & biomimetics*. 2018 Oct 10;13(6):066007.

Nigg BM, Bahlsen HA, Luethi SM, Stokes S. The influence of running velocity and midsole hardness on external impact forces in heel-toe running. *Journal of biomechanics*. 1987 Jan 1;20(10):951-9.

Roy JP, Stefanyshyn DJ. Shoe midsole longitudinal bending stiffness and running economy, joint energy, and EMG. *Medicine & Science in Sports & Exercise*. 2006 Mar 1;38(3):562-9.

Frederick EC. Physiological and ergonomics factors in running shoe design. *Applied ergonomics*. 1984 Dec 1;15(4):281-7.

Browning RC, Modica JR, Kram R, Goswami A. The effects of adding mass to the legs on the energetics and biomechanics of walking. *Medicine & Science in Sports & Exercise*. 2007 Mar 1;39(3):515-25.

Thies SB, Price C, Kenney LP, Baker R. Effects of shoe sole geometry on toe clearance and walking stability in older adults. *Gait & posture*. 2015 Jul 1;42(2):105-9.

b) We selected the muscles in order to have at least one muscle for every flexion and extension action of the ankle, knee and hip joints using muscles that could be measured using surface EMG:

Muscle chosen	Function
Soleus	Mono-articular plantarflexion
Gastrocnemius medialis	Bi-articular plantarflexion and knee flexion
Tibialis anterior	Dorsiflexion
Vastus medialis	Knee extension
Rectus femoris	Hip flexion and knee extension
Biceps femoris	Knee flexion
Gluteus maximus	Hip extension

=> We have **added the following justification (underlined text) for the selection of muscles** in the 2.5 measurements section in methods:

We recorded muscle activation using a wireless electromyography (EMG) system (Trigno TM, Delsys, USA; 2000 Hz). Since the **footwear conditions were expected to primarily influence the ankle muscles [24-27]** and the **treadmill grades were expected to influence all lower limb joints [7,8,10,38]** and since both manipulations happen **mostly in the sagittal plane (i.e., we did not use shoe wedges that were tilted in the frontal plane)**, **we chose to record the muscle activation of the most accessible (major) flexor and**

extensor muscles of the ankle, knee and hip joints: soleus, gastrocnemius medialis, tibialis anterior, vastus medialis, rectus femoris, biceps femoris, and gluteus maximus.

References (already in the manuscript):

Stefanyshyn DJ, Nigg BM, Fisher V, O'Flynn B, Liu W. The influence of high heeled shoes on kinematics, kinetics, and muscle EMG of normal female gait. *Journal of Applied Biomechanics*. 2000 Aug 1;16(3):309-19.

Mika A, Oleksy Ł, Mika P, Marchewka A, Clark BC. The influence of heel height on lower extremity kinematics and leg muscle activity during gait in young and middle-aged women. *Gait & posture*. 2012 Apr 1;35(4):677-80.

Simonsen EB, Svendsen MB, Nørreslet A, Baldvinsson HK, Heilskov-Hansen T, Larsen PK, Alkjær T, Henriksen M. Walking on high heels changes muscle activity and the dynamics of human walking significantly. *Journal of applied biomechanics*. 2012 Feb 1;28(1):20-8.

Li JX, Hong Y. Kinematic and electromyographic analysis of the trunk and lower limbs during walking in negative-heeled shoes. *Journal of the American Podiatric Medical Association*. 2007 Nov;97(6):447-56.

Lay AN, Hass CJ, Nichols TR, Gregor RJ. The effects of sloped surfaces on locomotion: an electromyographic analysis. *Journal of biomechanics*. 2007 Jan 1;40(6):1276-85.

Silder A, Besier T, Delp SL. Predicting the metabolic cost of incline walking from muscle activity and walking mechanics. *Journal of biomechanics*. 2012 Jun 26;45(10):1842-9.

Leroux A, Fung J, Barbeau H. Adaptation of the walking pattern to uphill walking in normal and spinal-cord injured subjects. *Experimental brain research*. 1999 May 1;126(3):359-68.

Franz JR, Kram R. The effects of grade and speed on leg muscle activations during walking. *Gait & posture*. 2012 Jan 1;35(1):143-7.

c) We calculated the individual leg power because treadmill manipulation is a whole-body manipulation (i.e., inclining the treadmill does not act only on a single joint) and foundational studies on the biomechanics of uphill and downhill walking describe the effects on ground reaction forces and work required.

=> We have **added the following justification** (underlined text) in the introduction:

*We also hypothesized that a shoe inclination that exactly offsets the treadmill grade would minimize the metabolic rate during downhill and uphill walking because it would mimic walking on stairs [Corlett, 1972]. Since both experimental manipulations (footwear and treadmill) occurred in the sagittal plane, we sought potential explanations in the sagittal plane biomechanical parameters. Therefore, we focused mostly on ankle kinetics and muscle activation to explain the effects of footwear changes [14, 15, 24-27] and on **whole-body parameters (GRF and COM work rate) to explain the effects of treadmill grade changes.***

References (already in the manuscript):

Ebbeling CJ, Hamill J, Crussemeyer JA. Lower extremity mechanics and energy cost of walking in high-heeled shoes. *Journal of Orthopaedic & Sports Physical Therapy*. 1994 Apr;19(4):190-6.

Stefanyshyn DJ, Nigg BM, Fisher V, O'Flynn B, Liu W. The influence of high heeled shoes on kinematics, kinetics, and muscle EMG of normal female gait. *Journal of Applied Biomechanics*. 2000 Aug 1;16(3):309-19.

Mika A, Oleksy Ł, Mika P, Marchewka A, Clark BC. The influence of heel height on lower extremity kinematics and leg muscle activity during gait in young and middle-aged women. *Gait & posture*. 2012 Apr 1;35(4):677-80.

Simonsen EB, Svendsen MB, Nørreslet A, Baldvinsson HK, Heilskov-Hansen T, Larsen PK, Alkjær T, Henriksen M. Walking on high heels changes muscle activity and the dynamics of human walking significantly. *Journal of applied biomechanics*. 2012 Feb 1;28(1):20-8.

Li JX, Hong Y. Kinematic and electromyographic analysis of the trunk and lower limbs during walking in negative-heeled shoes. *Journal of the American Podiatric Medical Association*. 2007 Nov;97(6):447-56.

d) As far as we understand, we reported the statistical results of all seven muscles in Figures 6, S2 and S3 which are all cited in the manuscript. In the results and discussion, we focused on the soleus, gastrocnemius medialis and tibialis anterior muscles since those muscles were hypothesized to be affected by the foot segment angle.

We apologize as based the reviewer's comment, it may have seemed like we left out a part of the results. => We have revised the following text (changes are underlined) in the discussion to clarify why we focused on certain muscles in the results and discussion and clarify that we also analyzed the other muscles: To investigate why level shoes minimized metabolic rate during walking on a level treadmill, we started by analyzing data from the ankle muscles and the joint kinetics since we expected that the footwear manipulation to mostly effect the ankle joint.

R4.Q2. The initial aim is vague: "the aim of this study was to investigate the effects of varying footwear outsole geometry on human mechanics and energetics under different walking grades." compared to the final aim: "The aim of the present study was to investigate the interaction effects of varying footwear outsole geometry and treadmill grade on metabolic rate, joint mechanics, muscle activity, ground reaction forces and COM mechanics.". There is no connection made between metabolic power and biomechanics.

We thank the reviewer for the advice.

=> We have **revised our initial aim** that was too vague by the following version (changes are underlined): Hence, the aim of this study was to investigate the *interaction* effects of varying footwear outsole geometry and treadmill grade **on metabolic rate, joint mechanics, muscle activity, ground reaction forces and COM mechanics in a controlled experiment** (all footwear and treadmill parameters were kept constant except the outsole geometry and treadmill grade).

R4.Q3. Why and how would positive COM power and propulsive force contribute to metabolic power?

The reviewer raises excellent questions.

Regarding the question of why, this was our motivation:

Although metabolic energy consumption originally comes from the internal muscle fiber behavior, we often study relationships between energetics and external mechanical work and force [e.g., Margaria, 1938; Gottschall and Kram, 2003] because of the challenges associated with measuring whole-body muscle fiber behavior and because external mechanical work and force are ultimately caused by muscle contractions when there are no other assisting forces. Since the treadmill grade manipulation does not act on a specific joint (compared to footwear manipulation), we believe that it makes sense to start looking for explaining variables at a whole-body level: the work rate on the COM performed by the individual legs [Donelan et al., 2002] and the force delivered by the legs [Donelan et al., 2002].

=> Regarding the question of how, we have **added the following text in the discussion:**

The minima for the positive COM work rate and propulsive GRF were very close to the treadmill grade (-6° downhill) that minimized the metabolic rate (Figs. 5B and S4, B). Both the mechanical work rate and force variables could indeed be responsible for the changes in metabolic rate because **energy consumption at the muscle fiber level is dependent on the fiber work force [17]. Concentric muscle work is four times more expensive than eccentric muscle work [5]** which could explain why the positive COM work rate and propulsive GRF are closely related to the metabolically optimal treadmill grade.

References (already in the manuscript):

Donelan JM, Kram R, Kuo AD. Simultaneous positive and negative external mechanical work in human walking. *Journal of biomechanics*. 2002 Jan 1;35(1):117-24.

Umberger BR, Gerritsen KG, Martin PE. A model of human muscle energy expenditure. *Computer methods in biomechanics and biomedical engineering*. 2003 May 1;6(2):99-111.

Margaria R. Sulla fisiologia e specialmente sul consumo energetico della marcia e della corsa a varie velocita ed inclinazioni del terreno. 1938.

Additional reference not in manuscript:

Gottschall JS, Kram R. Energy cost and muscular activity required for propulsion during walking. *Journal of Applied Physiology*. 2003 May 1;94(5):1766-72.

R4.Q4. Why would longer plantarflexor contractile element lengths reduce metabolic power?

In the equations below [Umberger et al., 2003] which are based on muscle studies from Woledge et al., [1985] and Hilber et al., [2001], it can be seen that **when the contractile element length is larger than the optimal length, the first and second terms of \dot{E} (activation-maintenance and shortening-lengthening heat rate) are multiplied by a force-length multiplier (FISO)**. This force-length multiplier is smaller than 1 for lengths greater than the optimal length. As such, the model from Umberger suggests that metabolic rate decreases with longer contractile element lengths.

if $L_{CE} \leq L_{CE(OPT)}$,

$$\dot{E} = \dot{h}_{AM} A_{AM} S$$

$$+ \begin{cases} [-\alpha_{S(ST)} \dot{V}_{CE} (1 - \%FT/100) - \alpha_{S(FT)} \dot{V}_{CE} (\%FT/100)] A_s S & \text{if } \dot{V}_{CE} \leq 0 \\ \alpha_L \dot{V}_{CE} A S & \text{if } \dot{V}_{CE} > 0 \end{cases}$$

$$- (F_{CE} V_{CE})/m$$

if $L_{CE} > L_{CE(OPT)}$,

$$\dot{E} = (0.4 \times \dot{h}_{AM} + 0.6 \times \dot{h}_{AM} F_{ISO}) A_{AM} S$$

$$+ \begin{cases} [-\alpha_{S(ST)} \dot{V}_{CE} (1 - \%FT/100) - \alpha_{S(FT)} \dot{V}_{CE} (\%FT/100)] F_{ISO} A_s S & \text{if } \dot{V}_{CE} \leq 0 \\ \alpha_L \dot{V}_{CE} F_{ISO} A S & \text{if } \dot{V}_{CE} > 0 \end{cases}$$

$$- (F_{CE} V_{CE})/m$$

Muscle-energetics model from Umberger et al., [2003].

=> We have **added the underlined text** to explain why longer contractile element lengths lead to lower metabolic rate based on the muscle-metabolic energetics model from Umberger et al., [2003] and the underlying muscle studies:

The energetics model from Umberger et al., [17] which is based on muscle studies [19,20] predicts that when contractile elements are longer than the optimal length, the estimated metabolic rate is multiplied by a force-length dependent coefficient that is lower than one resulting in a lower metabolic rate. If we assume that shoes that result in upward foot segment angles would lead to longer plantarflexor contractile element lengths, then if all other parameters remain constant, based on the muscle-energetics model from Umberger et al., [17] we would expect lower metabolic rate values in conditions with upward foot segment angles.

References (already in the manuscript):

Umberger BR, Gerritsen KG, Martin PE. A model of human muscle energy expenditure. Computer methods in biomechanics and biomedical engineering. 2003 May 1;6(2):99-111.

Woledge RC, Curtin NA, Homsher E. Energetic aspects of muscle contraction. Monographs of the physiological society. 1985;41:1-357.

Hilber K, Sun YB, Irving M. Effects of sarcomere length and temperature on the rate of ATP utilisation by rabbit psoas muscle fibres. The Journal of physiology. 2001 Mar;531(3):771-80.

R4.Q5. a) I am confused by the wording that describes the shoe. I have looked at Figure 2 and looked at the first paragraph of the results section and I can't figure out which shoe and treadmill grade combination minimizes metabolic power. Specifically, this statement and others like it are confusing: "A -3° (downward) foot segment angle (or a +3° upward outsole)". b) How was the minimum metabolic rate established for the combination of treadmill grade and shoe inclination?

a) Again, we apologize for not writing this clearly.

We attempted to clearly distinguish all three variables by using consistent terminology

- Outsole angle = angle of the shoe versus the bottom of the blocks.
- Treadmill angle = angle of the treadmill versus the horizontal plane.
- Foot segment angle = angle of the foot versus the horizontal plane

All statistical analyses were conducted using "foot segment angle" as the independent variable because our central hypothesis was focused on the foot segment angle.

=> However, the outsole angle can easily be derived from the foot segment angle based on the following relationship which we have added to Figure 2A.

(A)

=> Since the foot segment angle, treadmill grade and outsole angle are connected to each other and because of the comments of the reviewer, **we have tried to simplify as follows by removing redundant information:**

~~A -3° downward foot segment angle (or A +3° (upward) outsole) minimized metabolic rate during walking on a -6° (downhill) treadmill. A +3° (upward) foot segment angle (or A -3° (downward) outsole) minimized metabolic rate during walking on a +6° (uphill) treadmill.~~

=> **To make it clear which combination of shoe and treadmill was optimal we have added mini-pictograms** and text labels with the specifications of the optimal condition for each treadmill grade in Figure 4:

b) The location of the minimum metabolic rate for each treadmill grade was established by calculating the minimum of the equation with the coefficients that were found to be statistically significant as shown in Table S1.

=> This was briefly explained in the captions of Figure 4 and onwards but we have now added the following (underlined) explanation in the 2.7 statistical analyses section in methods. **We have also added an explanation on the calculation of the individual minima** based on suggestions from reviewer 3 (R3.Q1): *To avoid overfitting and to adapt the model for dependent variables that have linear trends, we removed terms that did not significantly contribute using backward stepwise elimination similar to other studies [51,52]. If the resulting trend showed a minimum versus the foot segment angle and/or treadmill grade, the location of the minima was obtained by calculating the minimum of the equation from the linear mixed-effects model (with coefficients shown in Table S1) at the different treadmill grades and shoe angles. To obtain a sense of the inter subject variability of the location of the minima, we fitted the terms of the linear-mixed effects model that were statistically significant on each individual participant and calculated the location of the individual minima.*

R4.Q6. a) Also Figures 4-6 (B) seem to indicate that only 3 shoe inclinations were tested at each slope, but the methods section implies that five were tested. Which 15 conditions were tested? b) Were all 15 trials completed in one day? If so, how did you account for fatigue and day-to-day variability?

a) We apologize and understand the concern of the reviewer that this was not clear enough. As shown in Figure 2B, we tested 5 (not 3) outsole angles at each treadmill grade. All 15 conditions were included in the evaluation of the statistical model but in Figures 4-6 (B) we only drew the lines representing the result from the linear mixed-effects model for the outsole angles that were evaluated on all three treadmill grades.

=> We have **added the following clarification** (underlined text) in the 2.3 experimental conditions section in methods:

The footwear inclinations for each treadmill condition were approximately centered around the inclination that offset the treadmill grade. Specifically, five footwear inclinations (-3°, 0°, 3°, 7° and 12°) were tested during (-6°) downhill walking, five footwear inclinations (-7°, -3°, 0°, 3° and 7°) were tested during level walking, and five footwear inclinations (-12°, -7°, -3°, 0° and 3°) were tested during (+6°) uphill walking. Only the -3°, 0° and 3° footwear inclinations were tested at all treadmill grades.

=> We have added the following clarification in the captions of Figures 4-6 (B):

Although five shoe inclinations were tested at each treadmill grade and were included in the evaluation of the statistical model, we only plotted the lines representing the evaluation of the linear mixed-effects model analysis for the three shoe inclinations that were tested on all three treadmill grades.

b) All 15 trials were completed in a single session. We understand the concern of the reviewer that the protocol was relatively long but it appears to be within the range of protocols that tested a similar number of conditions in a single session [Galle et al., 2017; Quinlivan et al., 2017; Jackson and Collins, 2015; Caputo and Collins, 2014; Heitkamp et al., 2019; Grabowski et al., 2005; Sanchez et al., 2019; Abram et al., 2019].

=> We have added the following (underlined) text in the 2.4 protocol section in methods to explain how we avoided the confounding effects of fatigue and day-to-day variability:

All testing conditions lasted five minutes and were completed in a single session to avoid day-to-day variability and marker repositioning errors would affect metabolic rate and biomechanical

measurements. The conditions (shoe inclination angles and treadmill grades) were semi-randomized for each participant using a random order generating software, and to minimize the number of shoe inclination angle changes, we tested each shoe inclination on all applicable treadmill grades before changing to the next shoe inclination. The order of the conditions for every second participant was the inverse of the order of the previous participant. As such, all conditions were symmetrically distributed over the first and second halves of the protocol to avoid the effects of metabolic drift on the results. We also allowed the participants at least two minutes of rest between conditions.

=> We have also added the following (underlined) **limitation statement** in the discussion:

While we avoided fatigue confounding the results by randomizing the conditions such that every condition was situated in the beginning and the end of the protocol, it is also possible that different results could have been found if the protocol was divided into shorter sessions with a lower number of conditions.

Reference (already in the manuscript):

Galle S, Malcolm P, Collins SH, De Clercq D. Reducing the metabolic cost of walking with an ankle exoskeleton: interaction between actuation timing and power. Journal of neuroengineering and rehabilitation. 2017 Dec;14(1):35.

Additional references not in manuscript:

Quinlivan BT, Lee S, Malcolm P, Rossi DM, Grimmer M, Siviyy C, Karavas N, Wagner D, Asbeck A, Galiana I, Walsh CJ. Assistance magnitude versus metabolic cost reductions for a tethered multiarticular soft exosuit. Sci Robot. 2017 Jan 18;2(2):4416.

Jackson RW, Collins SH. An experimental comparison of the relative benefits of work and torque assistance in ankle exoskeletons. Journal of Applied Physiology. 2015 Jul 9;119(5):541-57.

Caputo JM, Collins SH. Prosthetic ankle push-off work reduces metabolic rate but not collision work in non-amputee walking. Scientific reports. 2014 Dec 3;4:7213.

Heitkamp LN, Stimpson KH, Dean JC. Application of a novel force-field to manipulate the relationship between pelvis motion and step width in human walking. IEEE Trans Neural Syst Rehabil Eng. 2019 Oct;27(10):2051-2058.

Grabowski A, Farley CT, Kram R. Independent metabolic costs of supporting body weight and accelerating body mass during walking. Journal of Applied Physiology. 2005 Feb;98(2):579-83.

Sanchez N, Simha SN, Donelan JM, Finley JM. Taking advantage of external mechanical work to reduce metabolic cost: the mechanics and energetics of split-belt treadmill walking. The Journal of physiology. 2019 Jun 13.

Abram SJ, Selinger JC, Donelan JM. Energy optimization is a major objective in the real-time control of step width in human walking. Journal of biomechanics. 2019 Jun 25;91:85-91.

R4.Q7. Abstract a) Include p-values and number of subjects and walking speed b) Please specify what "ankle joint parameters" and whole-body parameters are

a) Thank you for this suggestion.

=> We have **included the p-values, number of subjects and walking speed in the abstract.**

b) Again, thank you for this suggestion.

=> **We have specified the requested parameters (underlined text) in the abstract:**

Shoe inclination influenced (distal) **ankle joint parameters** including soleus muscle activity, ankle moment and work rate, whereas treadmill grade influenced (**whole-body**) ground reaction force and center-of-mass work rate as well as (distal) ankle joint parameters including tibialis anterior and plantarflexor muscle activity, ankle moment and work rate.

R4.Q8. Introduction a) More justification is needed for why the authors are studying walking on different grades. It is not apparent why stairs are compared to grades or why it's important to understand the biomechanical parameters. b) P4. Lines 21-35. A link between metabolic cost and the mechanical and muscle activity changes during uphill and downhill walking is needed. As written, it is not clear what the point of the paragraph is. c) P4. Line 43. Do shoes with the toes pointed upward have the same correlation between height and metabolic cost? d) P4. Line 48. This sentence lacks a comparison. e) P4-5. Starting at Line 29. Please switch the first two sentences in this paragraph and/or revise to better link to the previous paragraph. f) P5. Lines 19-20. There is no justification for why a shoe that offsets grade would minimize metabolic cost.

a) Our apologies, we agree with the reviewer that we had not sufficiently justified this. A lot of studies have focused on walking on level ground but walking on level ground only demands a low percentage of the maximum sustainable metabolic rate in healthy participants whereas walking becomes challenging on uphill or downhill grades. Because of this, we think it is interesting to study possible biomechanical parameters to facilitate walking on grades. As far as we know, no studies have investigated footwear for assisting walking on grades and the study from Corlett et al., [1972] which shows that stairs can reduce the metabolic cost of walking appear an interesting point of reference for the introduction (as further explained in response to question 8 f) from the reviewer (R4.Q8.f)).

=> We have **added the following text in the introduction to justify the relevance** (also consistent with suggestions from reviewer 1 (R1.Q1 and R1.Q2)):

Walking on level ground demands little effort, but walking on slopes quickly becomes challenging on a metabolic [2-6] and muscular level [7-10]. While it **might not be possible to entirely eliminate the cost of raising or lowering the center of mass (COM) against or with gravity during downhill or uphill walking, investigating the effects of whole-body and distal limb mechanics on metabolic energy cost could inform new ways of minimizing metabolic rate. For example, stairs are one possible way to reduce metabolic cost compared to climbing an equivalent slope on a ramp surface [11]. Walking on stairs differs from walking on a ramp since stairs allow horizontal foot placement but also require placing the feet in specific positions.**

b) Thank you for noticing this.

=> We have **revised this paragraph to improve the link between metabolic cost, mechanical and muscle activity:**

Changes in metabolic rate during uphill and downhill walking come from changes in whole-body mechanics [2-6,12]. Uphill walking leads to an increase in metabolic rate since the legs have to perform more positive work [13] to move the COM upward against gravity. Downhill walking causes a decrease in metabolic rate, but only up to approximately a -6° grade [2,4,6]. During downhill walking, the muscles produce more eccentric work than during level or uphill walking [13] to prevent the COM from accelerating downward. Negative mechanical work is less metabolically costly than positive mechanical work [5] indicating that downhill walking is less metabolically costly than uphill walking. This **mechanical work is delivered by the muscles: when walking uphill, hip, knee, and ankle extensor muscle activities are increased during the stance phase [7,8] and serve to perform positive work. In contrast, downhill walking causes an increase in the activity of knee extensors [7,9] which is required for limiting forward velocity.**

c) Thank you for your question. This line contains a sentence that refers to the study from Li, [2003] which found that walking with shoes that are pointed upward (also called negative heel shoes) increased the metabolic rate compared to walking with normal shoes. Since **only two shoes were used in this study (negative heel shoes and normal shoes), the author only claims that negative heel shoes increase metabolic rate and does NOT claim** that there is a correlation between heel height and metabolic rate.

=> We have verified that our text only indicates a difference but does not suggest a correlation.

d) Thank you for noticing this.

=> We have revised this sentence (changes are underlined) in the introduction:

For example, adding a curved surface with a radius of 30% of the leg length to the bottom of a rigid boot minimized metabolic rate **compared to a range of tested radii [16].**

e) Thank you for this suggestion.

=> **We have revised the first two sentences** to provide a better link to the previous paragraph in the introduction:

Changes in metabolic rate during uphill and downhill walking come from changes in whole-body mechanics [2-6,12]. Uphill walking leads to an increase in metabolic rate since the legs have to perform more positive work [13] to move the COM upward against gravity.

f) Thank you for pointing out that this sentence did not contain a justification.

=> We have **revised our hypothesis (changes are underlined) in the introduction to provide a justification:**

We also hypothesized that a shoe inclination that exactly offsets the treadmill grade would minimize the metabolic rate during downhill and uphill walking because it would mimic walking on stairs [11].

We also introduced results from the study that compared stairs to ramps [Corlett, 1972] earlier in the introduction section of the manuscript.

Additional figure (not in manuscript). **Figure R5: Schematic explaining the hypothesis of how walking with shoes that offset the treadmill grade could mimic the benefits of stairs.**

Reference (already in the manuscript):

Corlett EN, Hutcheson C, DeLugan MA, Rogozenski J. Ramps or stairs: The choice using physiological and biomechanic criteria. Applied ergonomics. 1972 Dec 1;3(4):195-201.

R4.Q9. Methods a) Line 7. Why all males? b) Line 5. Why were 15 combinations tested? c) P7. Line 29. Why was the speed 1 m/s? And how would this affect the results of the study? d) Did subjects fast prior to the experiment? e) P8. Line 19. The Brockway equation allows you to calculate metabolic power from the rates of oxygen consumption and carbon dioxide production. Please revise.

a) It is likely that experience with walking on shoes with certain inclinations (e.g., high heels) could confound the results. Thus, we expected that it would be advantageous to test a homogenous population. Second, for logistic reasons, we had to limit the number of shoe prototypes; therefore, it was advantageous to have shoe prototypes around the same size.

=> We have **added the following (underlined) justification** in the 2.1 participants section in methods:

We tested a homogeneous sample of male participants in order to limit the chances that differences in previous experience with wearing high heels among participants would affect the results and to limit the number of shoe sizes that we had to design for the testing population.

b) Thank you for asking this question.

=> We have added the following (underlined) text in the 2.3 experimental conditions section in methods:

We tested fifteen combinations of three different treadmill grades (-6° downhill, level and +6° uphill) and five footwear inclinations per treadmill condition (Figure 2). The number of combinations was selected based on an analysis of the results from pilot tests to detect interaction effects. A larger number of footwear conditions than treadmill grades was used because we expected that the effects of

treadmill grade would be larger than the effects of footwear inclination, which would be smaller and would therefore require a greater number of data points.

c) The reviewer asks an excellent question. To analyze the effects of footwear and treadmill inclination isolated from confounding effects from differences in speed, we chose to use a single speed (i.e., we chose to conduct our study based on the definition of a “controlled experiment”). **Since some of the conditions with higher positive and higher negative footwear inclinations at uphill and downhill grades were somewhat challenging, we chose the speed of 1 m·s⁻¹ that would be feasible under all conditions.** It appears that **other studies with challenging locomotion conditions also used a speed of 1 m·s⁻¹**; it is possible that this is also because they wanted the achievable speed to be constant under all conditions). For example, Voloshina et al., [2013] used the same speed (1 m·s⁻¹) to investigate the biomechanics and energetics of walking on uneven terrain. Another study from Kent et al., [2019] examined the effects of an uneven terrain surface on whole-body locomotor dynamics at a fixed speed of 1 m·s⁻¹ immediately following exposure and after a familiarization period. **A higher speed could have shifted the relative importance of different muscle groups** (e.g., if the metabolic cost of the ankle muscles would be higher, this could lead to a greater effect of foot segment angle) but this could also potentially make the protocol unfeasible to be completed at the same speed for all conditions.

=> **We have added the following references to justify the chosen speed** (that used the same speed on uneven terrain) in our sentence in the 2.4 protocol section in methods:

The treadmill speed was set at 1 m·s⁻¹ [35,36].

We have added the following reference:

Kent JA, Sommerfeld JH, Mukherjee M, Takahashi KZ, Stergiou N. Locomotor patterns change over time during walking on an uneven surface. *Journal of experimental biology*. 2019 Jul 15;222(14):jeb202093.

Reference (already in the manuscript):

Voloshina AS, Kuo AD, Daley MA, Ferris DP. Biomechanics and energetics of walking on uneven terrain. *Journal of Experimental Biology*. 2013 Nov 1;216(21):3963-70.

d) Our apologies, all participants fasted before the experiment, but we had forgotten to include this information. We followed the guidelines provided by Compher et al., [2006] that recommend participants fast for at least 6 hours and abstain from caffeine overnight.

=> We have **added the following** (underlined) text in the 2.4 protocol section in methods:

Participants were instructed to fast for at least 6 hours prior to the experiment and to abstain from caffeine overnight [33].

We have added the following reference:

Compher C, Frankenfield D, Keim N, Roth-Yousey L, Evidence Analysis Working Group. Best practice methods to apply to measurement of resting metabolic rate in adults: a systematic review. *Journal of the American Dietetic Association*. 2006 Jun 1;106(6):881-903.

e) The reviewer is correct. The Brockway equation allows the calculation of metabolic rate based on the rates of oxygen consumption and carbon dioxide production.

=> We have **implemented the suggested change from the reviewer** (changes are underlined) in the 2.6 data processing section in methods:

*We used the Brockway equation to **calculate metabolic rate based on the rates of oxygen consumption and carbon dioxide production** [40].*

R4.Q10. Results The results are extremely hard to follow. a) Why is a polynomial curve fit used for the three grades in Fig. 4B, Fig. 5B, and Fig. 6B? b) And the y-axis should be labelled "Change in Metabolic Power" c) Were any statistics done to ensure that the minimum metabolic power was significantly different for each foot segment angle/treadmill angle compared to the other angles? d) P12. Line 37. It is unclear what "Smaller downhill and great uphill treadmill grades" refers to or what it is being compared to.

a) We apologize that this was not clear enough. The curve fits represent the formula from the linear mixed-effects model analysis evaluated over the tested range of foot segment angles for each treadmill grade so these are not necessarily polynomial curve fits for each treadmill grade. **In the case of metabolic rate, we found a significant effect for the second order of foot segment angle which is why the curve fits look polynomial.** In the case of other variables, they might be linear (e.g., Figure S3, A) or even a constant horizontal line (e.g., Figure S3, E).

=> We have **added the following (underlined) text** in the 2.7 statistical analyses section in methods to **explain this:**

To facilitate the interpretation of the meaning of the significant coefficients (positive or negative interaction coefficients), we evaluated the formula resulting from the linear-mixed effects model analysis over the tested range of foot segment angles at each treadmill grade and plotted the results as lines in the scatter plots. As such, the results of the statistical analysis can be understood from the figures; for example, if the coefficient for a second-order term was not significant, then the results were plotted as straight lines, or when there was a significant interaction effect, this effect could be understood by analyzing the differences in the slope or the location of the minima.

=> We have also added the following (underlined) text in the captions of Figures 4-6 (B):

Colored lines represent the formula from the linear mixed-effects model analysis evaluated over the range of treadmill grades. Although five shoe inclinations were tested at each treadmill grade and were included in the evaluation of the statistical model, we only plotted the lines representing the evaluation of the linear mixed-effects model analysis for the three shoe inclinations that were tested on all three treadmill grades. The lines are parabolic because the effects of first and second order of treadmill grade were significantly different from zero.

b) Our apologies for this confusion. **We tried plotting the "change in metabolic rate (versus the level shoe condition)", but this made the figure harder to interpret because the three treadmill grades were superimposed. Therefore, we chose to plot net metabolic rate** [e.g., similar to the main result in figure 3 in Collins et al., 2015].

=> We have revised the title of the y-axis in Figure 4 to be: 'Net metabolic rate (W/kg)'

Reference (already in the manuscript):

Collins SH, Wiggin MB, Sawicki GS. Reducing the energy cost of human walking using an unpowered exoskeleton. Nature. 2015 Jun;522(7555):212.

c) We did not compare the foot segment angle/treadmill angle to the other angles. Since we tested 15 conditions, this would lead to a very large number of pairwise comparisons that could result in a high chance of type I error due to multiple testing. To avoid these large numbers of comparisons, **we tested the effects of all the conditions in a single statistical analysis** which provides a result in the form of an equation that is tested to significantly fit the individual data [e.g., similar to Liew et al., 2016].

Reference (already in the manuscript):

Liew BX, Morris S, Netto K. The effects of load carriage on joint work at different running velocities. Journal of biomechanics. 2016 Oct 3;49(14):3275-80.

d) We apologize. Our intended meaning was not to compare one condition to another condition but instead to say in words that these variables follow a trend with a positive slope versus treadmill grade starting from the downhill grades and continuing into the uphill grades.

=> We have replaced this sentence and all other sentences that had the same wording with: Increases in treadmill grade from downhill (negative) to level or from level to uphill (positive) led to increasing positive ankle work rate ($P = 0.001$; Table S1; Figure S1, F) and increasing average gastrocnemius medialis EMG ($P < 0.001$; Table S1; Figure S1, J).

R4.Q11. Discussion a) P13. Second and third paragraphs. Are these paragraphs referring to only the 0° shoe? b) What is a “a descending-rising trend”? c) P14. Lines 40-42. How would a higher plantarflexion moment result from a longer moment arm of the Achilles tendon in level shoes compared to high-heeled shoes? d) P15. Lines 7-9. Were these 2% and 8% metabolic power differences significant? e) P16. Lines 13-14. It is unclear how a percentage change in walking speed could relate to a percentage change in metabolic power. How can a speculation about and shoe angle that is outside of what was tested in the current study, which looked at healthy individuals, be used for potential guidance in a patient population? f) P16. The comparison with stairs seems like a stretch. It is not clear why the authors are comparing their data from slopes to stairs. Stairs are flat, level surfaces. Why would a shoe with an inclination angle be beneficial?

a) Thank you for pointing out that this was unclear. The differences in the effects of treadmill grade were very small between shoe inclination conditions. The findings discussed in the second and third paragraphs **apply to the three shoe inclinations (-3° , 0° and 3°) that were tested on all three treadmill grades (downhill, level and uphill).**

=> We have revised the text (changes are underlined) in the discussion:
The reductions in negative COM work rate and braking GRF due to increases in treadmill grade from downhill (negative) to level or from level to uphill (positive) for all shoe inclinations that were tested on all three treadmill grades are consistent with those from other studies [13,53–57].

b) We apologize and should have been more consistent with our terminology.

=> We have **replaced this with “U-shaped”**, which is used throughout the manuscript and appears to be clear to all reviewers.

c) Thank you for this question. This was not our hypothesis, but this was the hypothesis from Stefanyshyn et al., [2000]. In the discussion in their manuscript, the authors state that “as the ankle becomes more plantarflexed, as is the case for high heels, the length of the moment arm of the Achilles tendon decreases. Therefore, despite an increase in force in the Achilles tendon due to increased soleus muscle activity, there is a decrease in the moment that the force produces due to a decrease in moment arm length.”

We assume that the geometrical mechanism described in this figure (Figure R6) is responsible for this.

Additional figure (not in manuscript). **Figure R6: Assumed mechanism for the hypothesis from Stefanyshyn et al., [2000].**

=> We have **revised our text (changes are underlined) to point out that this hypothesis comes from Stefanyshyn et al., [2000]:**

It is hypothesized from a previous study [25] that this could be due to a longer moment arm of the Achilles tendon in level shoes compared to high-heeled shoes allowing the plantarflexors to apply higher moments.

Reference (already in the manuscript):

Stefanyshyn DJ, Nigg BM, Fisher V, O'Flynn B, Liu W. The influence of high heeled shoes on kinematics, kinetics, and muscle EMG of normal female gait. Journal of Applied Biomechanics. 2000 Aug 1;16(3):309-19.

d) We did not compare shoe inclination conditions to each other. Since we tested 15 conditions, this would lead to a very large number of pairwise comparisons that could result in a high chance of type I error due to multiple testing. To avoid these large numbers of comparisons, we tested the effects of all the conditions in a single statistical analysis which provides a result in the form of an equation that is tested to significantly fit to the individual data [e.g., similar to Liew et al., 2016].

=> To avoid giving the impression that we performed pairwise comparisons between shoe inclination conditions, **we have revised the sentence** (changes are underlined) and removed the report of the average reductions:

A -3° foot segment angle *minimized* metabolic rate during downhill walking and a 3° foot segment angle *minimized* metabolic rate during uphill walking.

Reference (already in the manuscript):

Liew BX, Morris S, Netto K. The effects of load carriage on joint work at different running velocities. Journal of biomechanics. 2016 Oct 3;49(14):3275-80.

e) The reviewer raises excellent points. Using a widely used equation from the literature that allows us to estimate the effects of changes in walking speed on metabolic rate changes [Pandolf et al., 1977], we can estimate that a (7%) change in speed from 1 to 0.93 m·s⁻¹ would lead to a change in metabolic rate (of 7.3%). Thus, we can hypothesize that a shoe with a 9° inclination would result in changes in walking speed of 7% for the same metabolic rate.

This text **was added based on a suggestion from reviewer 1** during the previous review round.

=> **We understand the point of view of the reviewer that this additional text might be too speculative** because the 9° shoe inclination is an extrapolation and the estimated possible change in walking speed is based on the literature equations for healthy walking. Therefore, **we have removed this text from the manuscript.**

Additional reference not in manuscript:

Pandolf KB, Givoni B, Goldman RF. Predicting energy expenditure with loads while standing or walking very slowly. Journal of Applied Physiology. 1977 Oct 1;43(4):577-81.

f) Thank you for this question. A study from **Corlett et al., [1972] shows that walking on stairs leads to a lower metabolic rate than walking on a ramp with an equivalent average slope** as that of the stairs (Table R5). Walking with shoes that offset the slope of a ramp could be considered somewhat similar to walking on stairs. Our statistical results confirm that walking with level shoes is only optimal when walking on a level treadmill grade.

=> **We agree with the reviewer that the comparison with stairs seems like a stretch in our discussion. Therefore, we have removed this text** from the discussion (also in response to question 20 from reviewer 3 (R3.Q20)).

Additional table (not in manuscript) from Corlett et al., [1972]. **Table R5: O₂ consumption measurements.** This table shows the ranking of the conditions from lowest to highest O₂ consumption for each participant. The conditions are labeled as "15" = stairs with a 15 inch riser, "10" = stairs with a 10 inch riser and R = ramp with an equivalent average slope as that of the stairs (i.e., 20 degrees) The results show that one of

the stairs conditions (i.e., the one with the 15 inch risers) always had the lowest O₂ consumption in all 8 of the participants.

SUBJT		Total O ₂ values rank, lowest first		
		2	3
SHORT SUBJECT	1	15	10	R
	15	10	R
	15	R	10
	15	10	R
TALL SUBJECT	5	15	10	R
	15	R	10
	15	10	R
	15	10	R

Ranking of and min total O₂
consumption 20° slope

Reference (already in the manuscript):

Corlett EN, Hutcheson C, DeLugan MA, Rogozenski J. Ramps or stairs: The choice using physiological and biomechanic criteria. Applied ergonomics. 1972 Dec 1;3(4):195-201.